# MAKING RL WITH PREFERENCE-BASED FEEDBACK EFFICIENT VIA RANDOMIZATION

**Runzhe Wu**
Department of Computer Science
Cornell University
rw646@cornell.edu

**Wen Sun**
Department of Computer Science
Cornell University
ws455@cornell.edu

## ABSTRACT

Reinforcement Learning algorithms that learn from human feedback (RLHF) need to be efficient in terms of *statistical complexity, computational complexity, and query complexity*. In this work, we consider the RLHF setting where the feedback is given in the format of preferences over pairs of trajectories. In the linear MDP model, using randomization in algorithm design, we present an algorithm that is sample efficient (i.e., has near-optimal worst-case regret bounds) and has polynomial running time (i.e., computational complexity is polynomial with respect to relevant parameters). Our algorithm further minimizes the query complexity through a novel randomized active learning procedure. In particular, our algorithm demonstrates a near-optimal tradeoff between the regret bound and the query complexity. To extend the results to more general nonlinear function approximation, we design a model-based randomized algorithm inspired by the idea of Thompson sampling. Our algorithm minimizes Bayesian regret bound and query complexity, again achieving a near-optimal tradeoff between these two quantities. Computation-wise, similar to the prior Thompson sampling algorithms under the regular RL setting, the main computation primitives of our algorithm are Bayesian supervised learning oracles which have been heavily investigated on the empirical side when applying Thompson sampling algorithms to RL benchmark problems.

## 1 INTRODUCTION

Reinforcement learning from human feedback (RLHF) has been widely used across various domains, including robotics (Jain et al., 2013; 2015) and natural language processing (Stiennon et al., 2020; Ouyang et al., 2022). Unlike standard RL, RLHF requires the agent to learn from feedback in the format of preferences between pairs of trajectories instead of per-step reward since assigning a dense reward function for each state is challenging in many tasks. For instance, in natural language generation, rating each generated token individually is challenging. Hence, it is more realistic to ask humans to compare two pieces of text and indicate their preference. Recent works have shown that, by integrating preference-based feedback into the training process, we can align models with human intention and enable high-quality human-machine interaction.

Despite the existing empirical applications of RLHF, its theoretical foundation remains far from satisfactory. Empirically, researchers first learn reward models from preference-based feedback and then optimize the reward models via policy gradient-based algorithms such as PPO (Schulman et al., 2017). Questions such as whether or not the learned reward model is accurate, whether PPO is sufficient for deep exploration, and how to strategically collect more feedback on the fly are often ignored. Theoretically, prior works study the regret bound for RL with preference-based feedback (Saha et al., 2023; Chen et al., 2022). Despite achieving sublinear worst-case regret, these algorithms are computationally intractable even for simplified models such as tabular Markov Decision Processes (MDPs). This means that we cannot easily leverage the algorithmic ideas in prior work to guide or improve how we perform RLHF in practice.

In addition to maximizing reward, another important metric in RLHF is the query complexity since human feedback is expensive to collect. To illustrate, we note that InstructGPT's training data comprises a mere 30K instances of human feedback (Ouyang et al., 2022), which is significantly fewer

than the internet-scale dataset for training the GPT-3 base model. This underscores the challenge of scaling up the size of human feedback datasets. Ross et al. (2013); Laskey et al. (2016) also pointed out that extensively querying for feedback puts too much burden on human experts. Empirically, Lightman et al. (2023) observes that active learning reduces query complexity and improves the learned reward model. In theory, query complexity is mostly studied in the settings of active learning, online learning, and bandits (Cesa-Bianchi et al., 2005; Dekel et al., 2012; Agarwal, 2013; Hanneke & Yang, 2021; Zhu & Nowak, 2022; Sekhari et al., 2023a;b), but overlooked in RL.

In this work, *we aim to design new RL algorithms that can learn from preference-based feedback and can be efficient in statistical complexity (i.e., regret), computational complexity, and query complexity*. In particular, we strike a near-optimal balance between regret minimization and query complexity minimization. To achieve this goal, our key idea is to use *randomization* in algorithm design. We summarize our new algorithmic ideas and key contributions as follows.

1. For MDPs with linear structure (i.e., linear MDP (Jin et al., 2020)), we propose the first RL algorithm that achieves sublinear worst-case regret and computational efficiency simultaneously with preference-based feedback. Even when reduced to tabular MDPs, it is still the first to achieve a no-regret guarantee and computational efficiency. Moreover, it has an active learning procedure and attains a near-optimal tradeoff between the regret and the query complexity. Our algorithm adds *random Gaussian noises* to the learned state-action-wise reward model and the least-squares value iteration (LSVI) procedure. Using random noise instead of the UCB-style technique (Azar et al., 2017) preserves the Markovian property in the reward model and allows one to use dynamic programming to achieve computation efficiency.

2. For function approximation beyond linear, we present a model-based Thompson-sampling (TS) algorithm that forms posterior distributions over the transitions and reward models. Assuming the transition and the reward model class both have small $\ell_1$-*norm eluder dimension* – a structural condition introduced in Liu et al. (2022a) that is more general than the common $\ell_2$-norm eluder dimension (Russo & Van Roy, 2013), we show that our algorithm again achieves a near-optimal tradeoff between the Bayesian regret and the Bayesian query complexity. Computation-wise, similar to previous TS algorithms for regular RL (e.g., Osband et al. (2013)), the primary computation primitives are Bayesian supervised learning oracles for transition and reward learning.

3. Our query conditions for both algorithms are based on variance-style uncertainty quantification of the preference induced by the randomness of the reward model. We query for preference feedback only when the uncertainty of the preference on a pair of trajectories is large. Approximately computing the uncertainty can be easily done using i.i.d. random reward models drawn from the reward model distribution, which makes the active query procedure computationally tractable.

Overall, while our main contribution is on the theoretical side, our theoretical investigation provides several new practical insights. For instance, for regret minimization, our algorithms propose to draw a pair of trajectories with one from the latest policy and the other from an older policy instead of drawing two trajectories from the same policy (e.g., Christiano et al. (2017)), avoiding the situation of drawing two similar trajectories when the policy becomes more and more deterministic. Our theory shows that drawing two trajectories from a combination of new and older policies balances exploration and exploitation better. Another practical insight is the variance-style uncertainty measure for designing the query condition. Compared to more standard active learning procedure that relies on constructing version space and confidence intervals (Dekel et al., 2012; Puchkin & Zhivotovskiy, 2021; Zhu & Nowak, 2022; Sekhari et al., 2023a;b), our new approach comes with strong theoretical guarantees and is more computationally tractable. It is also amenable to existing implementations of Thompson sampling RL algorithms (e.g., using bootstrapping to approximate the posterior sampling (Osband et al., 2016a; 2023)).

## 2 COMPARISON TO PRIOR WORK

**RL with preference-based feedback.** Many recent works have obtained statistically efficient algorithms but are computationally inefficient even for tabular MDPs due to intractable policy search and version space construction (Chen et al., 2022; Zhan et al., 2023a;b; Saha et al., 2023). For example, Zhan et al. (2023b); Saha et al. (2023) use the idea from optimal design and rely on the computation oracle: $\arg\max_{\pi,\pi'\in\Pi} \|\mathbb{E}_{s,a\sim\pi}\phi(s,a) - \mathbb{E}_{s,a\sim\pi'}\phi(s,a)\|_A$ with some positive definite matrix $A$.

Here $\|x\|_A^2 := x^\top A x$, and $\phi$ is some state-action feature.[1] It is unclear how to implement this oracle since standard planning approaches based on dynamic programming cannot be applied here. In addition, these methods also actively maintain a policy space by eliminating potentially sub-optimal policies. The policy class can be exponentially large even in tabular settings, so how to maintain it computationally efficiently is unclear. We provide a more detailed discussion on the challenges in achieving computational efficiency in RLHF in Appendix A.

While the work mentioned above is intractable even for tabular MDPs, there are some other works that could be computationally efficient but have weaker statistical results. For instance, very recently, Wang et al. (2023) proposed a reduction framework that can be computationally efficient (depending on the base algorithm used in the reduction). However, their algorithms have PAC bounds while we focus on regret minimization. Moreover, we achieve a near-optimal balance between regret and query complexity. Novoseller et al. (2020) proposed a posterior sampling algorithm for tabular MDP but their analysis is asymptotic (i.e., they do not address exploration, exploitation, and query complexity tradeoff). Xu et al. (2020) proposed efficient algorithms that do reward-free exploration. However, it is limited to tabular MDPs and PAC bounds.

In contrast to the above works, our algorithms aim to achieve efficiency in statistical, computational, and query complexities simultaneously. Our algorithms leverage *randomization* to balance exploration, exploitation, and feedback query. Randomization allows us to avoid non-standard computational oracles and only use standard Dynamic Programming (DP) based oracles (e.g., value iteration), which makes our algorithm computationally more tractable. Prior works that simultaneously achieve efficiency in all three aspects are often restricted in the bandit and imitation learning settings where the exploration problem is much easier (Sekhari et al., 2023a).

**RL via randomization.** There are two lines of work that study RL via randomization. The first injects random noise into the learning object to encourage exploration. A typical example is the randomized least-squares value iteration (RLSVI) (Osband et al., 2016b), which adds Gaussian noise into the least-squares estimation and achieves near-optimal worst-case regret (Zanette et al., 2020; Agrawal et al., 2021) for linear MDPs. The other line of work is Bayesian RL and uses Thompson sampling (TS) (Osband et al., 2013; Osband & Van Roy, 2014b;a; Gopalan & Mannor, 2015; Agrawal & Jia, 2017; Efroni et al., 2021; Zhong et al., 2022; Agarwal & Zhang, 2022). They achieve provable Bayesian regret upper bound by maintaining posterior distributions over models.

**Active learning.** Numerous studies have studied active learning across various settings (Cesa-Bianchi et al., 2005; Dekel et al., 2012; Agarwal, 2013; Hanneke & Yang, 2015; 2021; Zhu & Nowak, 2022; Sekhari et al., 2023b;a). However, most of them focus on the bandits and online learning settings, and their active learning procedures are usually computationally intractable due to computing version spaces or upper and lower confidence bounds. In contrast, we design a variance-style uncertainty quantification for our query condition, which can be easily estimated by random samples of reward model. This makes our active learning procedure more computationally tractable.

## 3 PRELIMINARY

**Notations.** For two real numbers $a$ and $b$, we denote $[a, b] := \{x : a \leq x \leq b\}$. For an integer $N$, we denote $[N] := \{1, 2, \ldots, N\}$. For a set $\mathcal{S}$, we denote $\Delta(\mathcal{S})$ as the set of distributions over $\mathcal{S}$. Let $d_{\mathrm{TV}}(\cdot, \cdot)$ denote the total variation distance.

We consider a finite-horizon Markov decision process (MDP), which is a tuple $M(\mathcal{S}, \mathcal{A}, r^\star, P^\star, H)$ where $\mathcal{S}$ is the state space, $\mathcal{A}$ is the action space, $P^\star : \mathcal{S} \times \mathcal{A} \to \Delta(\mathcal{S})$ is the transition kernel, $r^\star : \mathcal{S} \times \mathcal{A} \to [0, 1]$ is the reward function, and $H$ is the length of the episode. The interaction proceeds for $T$ rounds. At each round $t \in [T]$, we need to select two policies $\pi_t^0$ and $\pi_t^1$ and execute them separately, which generates two trajectories $\tau_t^0$ and $\tau_t^1$ where $\tau_t^i = (s_{t,1}^i, a_{t,1}^i, \ldots, s_{t,H}^i, a_{t,H}^i)$ for $i \in \{0, 1\}$. For the ease of notation, we assume a fixed initial state $s_1$. Then, we need to decide whether to make a query for the preference between $\tau_t^0$ and $\tau_t^1$. If making a query, we obtain a preference feedback $o_t \in \{0, 1\}$ that is sampled from the Bernoulli distribution:

$$\Pr(o_t = 1 \mid \tau_t^1, \tau_t^0, r^\star) = \Pr(\tau_t^1 \text{ is preferred to } \tau_t^0 \mid r^\star) = \Phi\big(r^\star(\tau_t^1) - r^\star(\tau_t^0)\big)$$

---

[1]These works typically assume trajectory-wise feature $\phi(\tau)$ for a trajectory $\tau$. However, even when specified to state-action-wise features, these algorithms are still computationally intractable, even in tabular MDPs.

where $r^\star(\tau_t^i) = \sum_{h=1}^H r^\star(s_{t,h}^i, a_{t,h}^i)$ for $i \in \{0,1\}$ is the trajectory reward, and $\Phi : \mathbb{R} \to [0,1]$ is a monotonically increasing link function. We note that, by symmetry, we have $\Phi(r^\star(\tau_t^0) - r^\star(\tau_t^1)) + \Phi(r^\star(\tau_t^1) - r^\star(\tau_t^0)) = 1$. If not making a query, we receive no feedback.

This feedback model is weaker than the standard RL where the per-step reward signal is revealed. We impose the following assumption on the link function $\Phi$, which has appeared in many existing works of RLHF (Saha et al., 2023; Zhu et al., 2023; Zhan et al., 2023a).

**Assumption 3.1.** *We assume $\Phi$ is differentiable and there exists constants $\kappa, \overline{\kappa} > 0$ such that $\kappa^{-1} \le \Phi'(x) \le \overline{\kappa}^{-1}$ for any $x \in [-H, H]$.*

The constants $\kappa$ and $\overline{\kappa}$ characterize the non-linearity of $\Phi$ and determine the difficulty of estimating the reward from preference feedback. It is noteworthy that, in the theoretical results of our algorithms, the bounds depend polynomially on $\kappa$ but logarithmically on $\overline{\kappa}$. Some typical examples of the link functions are provided below.

**Example 3.2** (Link functions). *It is common to have $\Phi(x) = 1/(1 + \exp(-x))$, which recovers the Bradley-Terry-Luce (BTL) model (Bradley & Terry, 1952), and we have $\kappa = 2 + \exp(-H) + \exp(H)$ and $\overline{\kappa} = 4$. Additionally, if the trajectory-wise reward is scaled within the interval of $[0,1]$, then the difference in reward will be within the range of $[-1,1]$. In this case, another common choice of the link function is $\Phi(x) = (x+1)/2$, which results in $\kappa = \overline{\kappa} = 2$.*

The goal is to minimize the worst-case regret and the query complexity simultaneously:

$$\mathrm{Regret}_T := \sum_{t=1}^T \left( 2V^\star(s_1) - V^{\pi_t^0}(s_1) - V^{\pi_t^1}(s_1) \right), \quad \mathrm{Queries}_T := \sum_{t=1}^T Z_t.$$

Here $V^\pi(s_1) := \mathbb{E}_\pi[\sum_{h=1}^H r^\star(s_h, a_h)]$ denotes the state-value function of policy $\pi$, and we define $V^\star(s_1) := V^{\pi^\star}(s_1)$ where $\pi^\star$ is the optimal policy that maximizes the state-value function. The variable $Z_t \in \{0,1\}$ indicates whether a query is made at round $t$. Note that the regret looks at the sum of the performance gaps between two pairs of policies: $(\pi^\star, \pi_t^0)$ and $(\pi^\star, \pi_t^1)$. This is standard in dueling bandits (Yue & Joachims, 2011; Yue et al., 2012; Dudík et al., 2015; Bengs et al., 2022; Wu et al., 2023b) and RL with preference-based feedback (Saha et al., 2023; Chen et al., 2022).

**Bayesian RL.** We also consider Bayesian RL in this work when learning with general function approximation. In the Bayesian setting, $P^\star$ and $r^\star$ are sampled from some known prior distributions $\rho_P$ and $\rho_r$. The goal is to minimize the Bayesian regret and the Bayesian query complexity:

$$\mathrm{BayesRegret}_T := \mathbb{E}\left[ \sum_{t=1}^T \left( 2V^\star(s_1) - V^{\pi_t^0}(s_1) - V^{\pi_t^1}(s_1) \right) \right], \quad \mathrm{BayesQueries}_T := \mathbb{E}\left[ \sum_{t=1}^T Z_t \right].$$

Here the expectation is taken with respect to the prior distribution over $P^\star$ and $r^\star$. We will use Bayesian supervised learning oracles to compute posteriors over the transition and reward model.

## 4 A MODEL-FREE RANDOMIZED ALGORITHM FOR LINEAR MDPS

In this section, we present a model-free algorithm for linear MDPs which is defined as follows.

**Assumption 4.1** (Linear MDP (Jin et al., 2020)). *We assume a known feature map $\phi : \mathcal{S} \times \mathcal{A} \to \mathbb{R}^d$, an unknown (signed) measure $\mu : \mathcal{S} \to \mathbb{R}^d$, and an unknown vector $\theta_r^\star$ such that for any $(s,a) \in \mathcal{S} \times \mathcal{A}$, we have $P^\star(s' \mid s, a) = \phi^\top(s,a) \cdot \mu(s')$ and $r^\star(s,a) = \phi^\top(s,a) \cdot \theta_r^\star$. We assume $\|\phi(s,a)\|_2 \le 1$ for all $(s,a) \in \mathcal{S} \times \mathcal{A}$, $\int_\mathcal{S} \|\mu(s)\|_2 \, \mathrm{d}s \le \sqrt{d}$, and $\|\theta_r^\star\|_2 \le B$ for some $B > 0$. For a trajectory $\tau = (s_1, a_1, \dots, s_H, a_H)$, we define $\phi(\tau) = \sum_{h=1}^H \phi(s_h, a_h)$ and assume $\|\phi(\tau)\|_2 \le 1$.*

Linear MDPs can capture tabular MDPs by setting $d = |\mathcal{S}||\mathcal{A}|$ and $\phi(s,a)$ to be the one-hot encoding of $(s,a)$. In this case, we have $\|\phi(\tau)\|_2 \le H$. However, we can scale it down to get $\|\phi(\tau)\|_2 \le 1$ at the expense of scaling $B$ up by $H$. We define $\Theta_B = \{\theta \in \mathbb{R}^d : \|\theta\|_2 \le B\}$, which contains $\theta_r^\star$.

### 4.1 ALGORITHM

The algorithm, called PR-LSVI, is presented in Algorithm 1. At the beginning of episode $k$, it first computes the maximum likelihood estimate $\widehat{\theta}_{r,t}$ (Line 3). Computation-wise, while the likelihood

objective is not guaranteed to be concave due to the generality of $\Phi$, efficient algorithms exist in certain common scenarios. For example, if $\Phi(x) = 1/(1 + \exp(-x))$, it recovers the BTL model (Example 3.2). In this case, the MLE objective is concave in $\theta$ and thus can be solved in polynomial running time. Moreover, we emphasize that the reward is learned under trajectory-wise features, which is different from the standard RL setting where it is learned under state-action features.

Given the MLE $\widehat{\theta}_{r,t}$, it next samples $\overline{\theta}_{r,t}$ from a Gaussian distribution centered at $\widehat{\theta}_{r,t}$ (Line 4). Note that the covariance matrix $\Sigma_{t-1}^{-1}$ uses trajectory-wise features (Line 16) which allows the randomized Gaussian vector to capture trajectory-wise uncertainty of the learned reward. The noise aims to encourage exploration. Then, it computes the least-squares estimate of the state-action value function $\widehat{\theta}_{P,t,h}$ for each $h \in [H]$ and samples $\overline{\theta}_{P,t,h}$ from a Gaussian distribution centered at $\widehat{\theta}_{P,t,h}$ (Lines 7-8). Similar to the reward model, the noise is added to the state-value function to encourage exploration. We then define the value function $\overline{Q}_{t,h}$ and $\overline{V}_{t,h}$ as

$$\overline{Q}_{t,h}(s,a) := \phi(s,a)^\top \overline{\theta}_{r,t} + \omega_{t,h}(s,a), \qquad \overline{V}_{t,h}(s) := \max_a \overline{Q}_{t,h}(s,a) \qquad (1)$$

and the function $\omega : \mathcal{S} \times \mathcal{A} \to \mathbb{R}$ is defined as

$$\omega_{t,h}(s,a) = \begin{cases} \phi(s,a)^\top \overline{\theta}_{P,t,h} & \text{if } \|\phi(s,a)\|_{\Sigma_{t-1,h}^{-1}} \leq \alpha_L \\ \rho(s,a)\left(\phi(s,a)^\top \overline{\theta}_{P,t,h}\right) + (1 - \rho(s,a))(H - h) & \text{if } \alpha_L < \|\phi(s,a)\|_{\Sigma_{t-1,h}^{-1}} \leq \alpha_U \\ H - h & \text{if } \|\phi(s,a)\|_{\Sigma_{t-1,h}^{-1}} > \alpha_U \end{cases}$$

where $\rho(s,a) = (\alpha_U - \|\phi(s,a)\|_{\Sigma_{t-1,h}^{-1}})/(\alpha_U - \alpha_L)$ interpolates between the two regimes to ensure continuity. This truncation trick is from Zanette et al. (2020) and is crucial. It controls the abnormally high value estimates. Specifically, when $\|\phi(s,a)\|_{\Sigma_{t-1,h}^{-1}}$ is large, the uncertainty in the direction of $\phi(s,a)$ is large, which makes the estimate $\phi(s,a)^\top \overline{\theta}_{P,t,h}$ abnormally large. In this case, we have to truncate it to $H - h$. Moreover, we note that the usual "value clipping" trick (i.e., simply constraining the value function within the range of $[0, H - h + 1]$ by clipping) cannot easily work here since it introduces bias to the random walk analysis, also pointed out by Zanette et al. (2020).

Then, the algorithm computes the greedy policy $\pi_t^0$ with respect to $\overline{Q}_{t,h}$. The comparator policy $\pi_t^1$ is simply set to the greedy policy from the previous episode, $\pi_{t-1}^0$. In other words, we are comparing the two most recent greedy policies. This is different from previous work, which compares the current greedy policy with a fixed comparator (Wang et al., 2023). Analytically, for our algorithm, the cumulative regret incurred by $\pi_t^1$ for all $t \in [T]$ is equivalent to that incurred by $\pi_t^0$ for all $t \in [T]$. Hence, it suffices to compute the regret for one of them and multiply it by two to get the total regret.

Given the trajectories $\tau_t^0$ and $\tau_t^1$ generated by $\pi_t^0$ and $\pi_t^1$, we compute the *expected absolute reward difference* between the trajectories under the same noisy distribution of the reward parameter:

$$\mathop{\mathbb{E}}_{\theta_0, \theta_1 \sim \mathcal{N}(\widehat{\theta}_{r,t}, \sigma_r^2 \Sigma_{t-1}^{-1})} \left[ \left| (\phi(\tau_t^0) - \phi(\tau_t^1))^\top (\theta_0 - \theta_1) \right| \right]. \qquad (2)$$

This represents the uncertainty of the preference between the two trajectories, and we make a query only when it is larger than a threshold $\epsilon$ (Line 13). Intuitively, we only make a query on two trajectories when we are uncertain about the preference (e.g., the expected disagreement between two randomly sampled reward models is large). Computationally, we can estimate this expectation by drawing polynomially many reward models from the distribution $\mathcal{N}(\widehat{\theta}_{r,t}, \sigma_r^2 \Sigma_{t-1}^{-1})$ and computing the empirical average. The deviation of the empirical average to the true mean can be easily bounded by standard concentration inequalities. We simply use expectation here for analytical simplicity. If the query condition is triggered, we make a query for feedback on $\tau_t^0, \tau_t^1$, and update the trajectory-wise feature covariance matrix accordingly.

## 4.2 ANALYSIS

The theoretical results of Algorithm 1 are stated in Theorem 4.2. The detailed assignment of hyper-parameters can be found in Table 1, and the proof is provided in Appendix B.

**Theorem 4.2.** *Define $\gamma = \sqrt{\kappa + B^2}$, which characterizes the difficulty of estimating the reward model. Set $\sigma_r = \widetilde{\Theta}(\gamma\sqrt{d})$, $\sigma_P = \widetilde{\Theta}(H^{3/2} d^2 \gamma)$, $\alpha_U = (d^{5/2} H^{3/2} \gamma)^{-1}$, $\alpha_L = \alpha_U/2$, and $\lambda = 1$.*

---

**Algorithm 1** Preference-based and Randomized Least-Squares Value Iteration (PR-LSVI)

---

**Require:** STD $\sigma_{\mathrm{r}}, \sigma_{\mathrm{P}}$, threshold $\epsilon$, value cutoff parameters $\alpha_{\mathrm{L}}, \alpha_{\mathrm{U}}$, and regularization parameter $\lambda$.

1: Let $\pi_0^0$ be an arbitrary policy, $\Sigma_0 \leftarrow \lambda I$, $\Sigma_{0,h} \leftarrow \lambda I$ $(\forall h \in [H])$.
2: **for** $t = 1, \ldots, T$ **do**
3:     $\widehat{\theta}_{\mathrm{r},t} \leftarrow \arg\max_{\theta \in \Theta_B} \sum_{s=1}^{t-1} Z_s \ln(o_s \Phi((\phi(\tau_s^1) - \phi(\tau_s^0))^\top \theta) + (1 - o_s)\Phi((\phi(\tau_s^0) - \phi(\tau_s^1))^\top \theta))$
4:     $\overline{\theta}_{\mathrm{r},t} \sim \mathcal{N}(\widehat{\theta}_{\mathrm{r},t}, \sigma_{\mathrm{r}}^2 \Sigma_{t-1}^{-1})$
5:     $\widehat{\theta}_{\mathrm{P},t,H} \leftarrow 0$, $\overline{\theta}_{\mathrm{P},t,H} \leftarrow 0$
6:     **for** $h = H-1, \ldots, 1$ **do**
7:         $\widehat{\theta}_{\mathrm{P},t,h} \leftarrow \Sigma_{t-1,h}^{-1}(\sum_{i=1}^{t-1} \phi(s_{i,h}^0, a_{i,h}^0) \overline{V}_{t,h+1}(s_{i,h+1}^0))$
8:         $\overline{\theta}_{\mathrm{P},t,h} \sim \mathcal{N}(\widehat{\theta}_{\mathrm{P},t,h}, \sigma_{\mathrm{P}}^2 \Sigma_{t-1,h}^{-1})$
9:         Define $\overline{Q}_{t,h}$ and $\overline{V}_{t,h}$ as in (1).
10:     **end for**
11:     Set $\pi_t^0 \leftarrow \{\pi_{t,h}^0 : \pi_{t,h}^0(s) = \arg\max_a \overline{Q}_{t,h}(s,a), \forall s \in \mathcal{S}, h \in [H]\}$ and $\pi_t^1 \leftarrow \pi_{t-1}^0$.
12:     Sample $\tau_t^0 \sim \pi_t^0$ and $\tau_t^1 \sim \pi_t^1$.
13:     $Z_t \leftarrow \mathbb{1}\{\mathbb{E}_{\theta_0,\theta_1 \sim \mathcal{N}(\widehat{\theta}_{\mathrm{r},t}, \sigma_{\mathrm{r}}^2 \Sigma_{t-1}^{-1})}[|(\phi(\tau_t^0) - \phi(\tau_t^1))^\top (\theta_0 - \theta_1)|] > \epsilon\}$
14:     **if** $Z_t = 1$ **then**
15:         Query preference feedback $o_t$ on $\{\tau_t^0, \tau_t^1\}$
16:         $\Sigma_t \leftarrow \Sigma_{t-1} + (\phi(\tau_t^0) - \phi(\tau_t^1))(\phi(\tau_t^0) - \phi(\tau_t^1))^\top$
17:     **else**
18:         $\Sigma_t \leftarrow \Sigma_{t-1}$
19:     **end if**
20:     $\Sigma_{t,h} \leftarrow \Sigma_{t-1,h} + \phi(s_{t,h}^0, a_{t,h}^0)\phi^\top(s_{t,h}^0, a_{t,h}^0)$ $(\forall h \in [H])$.
21: **end for**

---

*Then, PR-LSVI (Algorithm 1) guarantees the following with probability at least $1 - \delta$:*

$$\mathrm{Regret}_T = \widetilde{O}\left(\epsilon T d^{1/2} + \sqrt{T} \cdot d^3 H^{5/2}\gamma + d^{17/2}H^{11/2}\gamma^3\right), \quad \mathrm{Queries}_T = \widetilde{O}\left(d^4\gamma^4/\epsilon^2\right).$$

To further study the balance between the regret and the query complexity, we let $\epsilon = T^{-\beta}$ for some $\beta \leq 1/2$. Then, the upper bounds in Theorem 4.2 can be rewritten as

$$\mathrm{Regret}_T = \widetilde{O}(T^{1-\beta}), \quad \mathrm{Queries}_T = \widetilde{O}(T^{2\beta})$$

where we only focus on the dependence on $T$ and omit any other factors for simplicity. We see that there is a tradeoff in $T$ between the regret and the query complexity — the smaller regret we want, the more queries we need to make. For example, when $\beta = 0$, the regret is $\widetilde{O}(T)$, and the query complexity is $\widetilde{O}(1)$, meaning that we will incur linear regret if we don't make any query. If we increase $\beta$ to $1/2$, the regret decreases to $\widetilde{O}(\sqrt{T})$ while the query complexity increases to $\widetilde{O}(T)$, meaning that the regret bound is optimal in $T$ but we make queries every episode.

We emphasize that this tradeoff in $T$ is *optimal*, as evidenced by a lower bound result established by Sekhari et al. (2023a). Their lower bound was originally proposed for contextual dueling bandits, which is a special case of our setting. Their results are stated below.

**Theorem 4.3.** *(Sekhari et al., 2023a, Theorem 5) The following two claims hold: (1) For any algorithm, there exists an instance that leads to $\mathrm{Regret}_T = \Omega(\sqrt{T})$; (2) For any algorithm achieving an expected regret upper bound in the form of $\mathbb{E}[\mathrm{Regret}_T] = O(T^{1-\beta})$ for some $\beta > 0$, there exists an instance that results in $\mathbb{E}[\mathrm{Queries}_T] = \Omega(T^{2\beta})$.*

However, the dependence on other parameters (e.g., $d$ and $H$) can be loose, and further improvement may be possible. We leave further investigation of these factors as future work.

Although injecting random noise is inspired by RLSVI (Zanette et al., 2020), we highlight five key differences between ours and theirs: (1) Since the feedback is trajectory-wise, we need to design random noise that preserves the state-action-wise format (so that it can be used in DP) but captures the trajectory-wise uncertainty. We do this by maintaining $\Sigma_t$, which uses trajectory-wise feature differences; (2) Since the preference feedback is generated from some probabilistic model, we learn the

reward model via MLE and use MLE generalization bound (Geer, 2000) to capture the uncertainty in learning. This allows us to use a more general link function $\Phi$; (3) We design a new regret decomposition technique to accommodate preference-based feedback. Particularly, we decompose regret to characterizes the *reward difference* between $\pi_t^0$ and $\pi_t^1$: $\text{Regret}_T \lesssim \sum_{t=1}^T (\overline{V}_t - \widetilde{V}_t) - (V^{\pi_t^0} - V^{\pi_t^1})$ where $\overline{V}_t$ is an estimate of $V^{\pi_t^0}$, and $\widetilde{V}_t := \mathbb{E}_{\tau \sim \pi_t^1}[\sum_{h=1}^H \phi(s_h, a_h)^\top \overline{\theta}_{\text{r},t}]$ is an estimate of $V^{\pi_t^1}$ under the real transition and the learned reward model. This is different from standard RL (Zanette et al., 2020), and is necessary since we cannot guarantee the learned reward model will be accurate in a state-action-wise manner under the preference-based feedback. (4) Our algorithms have a new randomized active learning procedure for reducing the number of queries, and our analysis achieves a near-optimal tradeoff between regret and query complexity; (5) In every round $t$, we propose to draw a pair of trajectories where one is from the current greedy policy $\pi_t^0$ and the other is from the greedy policy of the previous round, $\pi_{t-1}^0$. This ensures $\pi_t^1$ is conditionally independent of the Gaussian noises at round $t$, which is the key to optimism (with a constant probability).

**Running time.** To assess the time complexity of Algorithm 1, assuming finite number of actions[2], all steps can be computed in polynomial running time (i.e., polynomial in $d, H, A$) except the MLE of the reward model (Line 3), which depends on the link function $\Phi$. For the popular BTL model where $\Phi(x) = 1/(1 + \exp(-x))$, the MLE objective is concave with respect to $\theta$ and $\theta$ belongs to a convex set $\Theta_B$. In this case, we can use any convex programming algorithms for the MLE procedure (e.g., projected gradient ascent).

## 5    A MODEL-BASED THOMPSON SAMPLING ALGORITHM

In this section, we aim to extend to nonlinear function approximation. We do so in a model-based framework with Thompson sampling (TS). The motivation is that TS is often considered a computationally more tractable alternative to UCB-style algorithms.

### 5.1    ALGORITHM

The algorithm, called PbTS, is presented in Algorithm 2. At the beginning of episode $k$, it computes the reward model posterior $\rho_{\text{r},t}$ and the transition model posterior $\rho_{\text{P},t}$ (Line 3). Then, it samples $P_t$ and $r_t$ from the posteriors and computes the optimal policy $\pi_t^0$ assuming the true reward function is $r_t$ and the true model is $P_t$ (Line 5). Here we denote $V_{r,P}^\pi$ as the state-value function of $\pi$ under reward function $r$ and model $P$. Note that this oracle is a standard planning oracle. The comparator policy $\pi_t^1$ is simply set to be the policy from the previous episode, $\pi_{t-1}^0$, as we did in Algorithm 1. The two policies then generate respective trajectories $\tau_t^0$ and $\tau_t^1$. To decide whether we should make a query, we compute the uncertainty quantity under the posterior distribution of the reward: $\mathbb{E}_{r,r' \sim \rho_{\text{r},t}}[|r(\tau_t^0) - r(\tau_t^1) - (r'(\tau_t^0) - r'(\tau_t^1))|]$, which is analogous to (2) in Algorithm 1. We make a query only when it is larger than a threshold $\epsilon$. Similar to Algorithm 1, we can approximate this expectation by sampling polynomial many pairs of $r$ and $r'$ and then compute the empirical average.

### 5.2    ANALYSIS

The theoretical results of Algorithm 2 should rely on the complexity of the reward and the transition model. In our analysis, we employ two complexity measures — eluder dimension and bracketing number. We start by introducing a generic notion of $\ell_p$-eluder dimension (Russo & Van Roy, 2013).

**Definition 5.1** ($\ell_p$-norm $\epsilon$-dependence). *Let $p > 0$. Let $\mathcal{X}$ and $\mathcal{Y}$ be two sets and $d(\cdot, \cdot)$ be a distance function on $\mathcal{Y}$. Let $\mathcal{F} \subseteq \mathcal{X} \to \mathcal{Y}$ be a function class. We say an element $x \in \mathcal{X}$ is $\ell_p$-norm $\epsilon$-dependent on $\{x_1, x_2, \ldots, x_n\} \subseteq \mathcal{X}$ with respect to $\mathcal{F}$ and $d$ if any pair of functions $f, f' \in \mathcal{F}$ satisfying $\sum_{i=1}^n d^p(f(x_i), f'(x_i)) \le \epsilon^p$ also satisfies $d(f(x), f'(x)) \le \epsilon$. Otherwise, we say $x$ is $\ell_p$-norm $\epsilon$-independent of $\{x_1, x_2, \ldots, x_n\}$.*

**Definition 5.2** ($\ell_p$-norm eluder dimension). *The $\ell_p$-norm $\epsilon$-eluder dimension of function class $\mathcal{F} \subseteq \mathcal{X} \to \mathcal{Y}$, denoted by $\dim_p(\mathcal{F}, \epsilon, d)$, is the length of the longest sequence of elements in $\mathcal{X}$ satisfying that there exists $\epsilon' \ge \epsilon$ such that every element in the sequence is $\ell_p$-norm $\epsilon'$-independent of its predecessors.*

---

[2]This is to ensure that $\arg\max_a Q(s, a)$ can be computed efficiently.

---

**Algorithm 2** Preference-based Thompson Sampling (PbTS)

---

**Require:** priors $\rho_P$ and $\rho_r$, threshold $\epsilon$.
1: Let $\pi_0^0$ be an arbitrary policy.
2: **for** $t = 1, \ldots, T$ **do**
3:     Compute posteriors:

$$\rho_{P,t}(P) \propto \rho_P(P) \prod_{i=1}^{t-1} \prod_{h=1}^{H} P(s_{i,h+1}^0 \,|\, s_{i,h}^0, a_{i,h}^0),$$

$$\rho_{r,t}(r) \propto \rho_r(r) \prod_{i=1}^{t-1} \Big( o_i \Phi\big(r(\tau_i^1) - r(\tau_i^0)\big) + (1 - o_i)\Phi\big(r(\tau_i^0) - r(\tau_i^1)\big) \Big)^{Z_i}.$$

4:     Sample $P_t \sim \rho_{P,t}$ and $r_t \sim \rho_{r,t}$.
5:     Compute $\pi_t^0 \leftarrow \arg\max_\pi V_{r_t, P_t}^\pi(s_1)$ and $\pi_t^1 \leftarrow \pi_{t-1}^0$.
6:     Sample $\tau_t^0 \sim \pi_t^0$ and $\tau_t^1 \sim \pi_t^1$.
7:     $Z_t \leftarrow \mathbb{1}\{\mathbb{E}_{r,r'\sim\rho_{r,t}}[|r(\tau_t^0) - r(\tau_t^1) - (r'(\tau_t^0) - r'(\tau_t^1))|] > \epsilon\}$
8:     **if** $Z_t = 1$ **then**
9:         Query preference feedback $o_t$ on $\{\tau_t^0, \tau_t^1\}$
10:     **end if**
11: **end for**

---

The eluder dimension is non-decreasing in $p$, i.e., $\dim_p(\mathcal{F}, \epsilon, d) \leq \dim_q(\mathcal{F}, \epsilon, d)$ for any $p \leq q$. In the analysis, we will focus on $\ell_1$- and $\ell_2$-norm eluder dimension, which have been used in nonlinear bandits and RL extensively (Wen & Van Roy, 2013; Osband & Van Roy, 2014a; Jain et al., 2015; Wang et al., 2020; Ayoub et al., 2020; Foster et al., 2021; Ishfaq et al., 2021; Chen et al., 2022; Liu et al., 2022a; Sekhari et al., 2023a;b). Examples where eluder dimension is small include linear functions, generalized linear models, and functions in Reproducing Kernel Hilbert Space (RKHS).

The other complexity measure we use is the bracketing number (Van de Geer, 2000).

**Definition 5.3** (Bracketing number). *Consider a function class $\mathcal{F} \subseteq \mathcal{X} \to \mathbb{R}$. Given two functions $l, u : \mathcal{X} \to \mathbb{R}$, the bracket $[l, u]$ is defined as the set of functions $f \in \mathcal{F}$ with $l(x) \leq f(x) \leq u(x)$ for all $x \in \mathcal{X}$. It is called an $\omega$-bracket if $\|l - u\| \leq \omega$. The bracketing number of $\mathcal{F}$ w.r.t. the metric $\|\cdot\|$, denoted by $N_{[]}(\omega, \mathcal{F}, \|\cdot\|)$, is the minimum number of $\omega$-brackets needed to cover $\mathcal{F}$.*

The logarithm of the bracketing number is small in many common scenarios, which has been extensively examined by previous studies (e.g., Van de Geer (2000)) for deriving MLE generalization bound (Agarwal et al., 2020; Uehara & Sun, 2021; Liu et al., 2022b; 2023). For example, when $\mathcal{F}$ is finite, the bracketing number is bounded by its size. When $\mathcal{F}$ is a $d$-dimensional linear function class, the logarithm of the bracketing number is upper bounded by $d$ up to logarithmic factors.

It is worth noting that while we will employ both measures to the model class $\mathcal{P}$, we can not similarly apply them to the reward class $\mathcal{R}$. Instead, we have to rely on the complexity of the following function class, which comprises functions mapping pairs of trajectories to reward differences:

$$\widetilde{\mathcal{R}} := \left\{ \widetilde{r} \,:\, \widetilde{r}(\tau^0, \tau^1) = \sum_{h=1}^{H} r(s_h^0, a_h^0) - r(s_h^1, a_h^1), \, \forall \tau^i = \{s_h^i, a_h^i\}_h, i \in \{0, 1\}, r \in \mathcal{R} \right\}. \quad (3)$$

We have to use $\widetilde{\mathcal{R}}$ instead of $\mathcal{R}$ because we only receive preference feedback, and thus we cannot guarantee that the learned reward model is accurate state-action-wise. Now we are ready to state our main results. The proofs are provided in Appendix C.

**Theorem 5.4.** *PbTS (Algorithm 2) guarantees that*

$$\text{BayesRegret}_T = \widetilde{O}\left( T\epsilon + H^2 \cdot \dim_1\big(\mathcal{P}, 1/T\big) \cdot \sqrt{T \cdot \iota_\mathcal{P}} + \kappa \cdot \dim_1\big(\widetilde{\mathcal{R}}, 1/T\big) \cdot \sqrt{T \cdot \iota_\mathcal{R}} \right),$$

$$\text{BayesQueries}_T = \widetilde{O}\left( \min\left\{ \frac{\kappa\sqrt{T \cdot \iota_\mathcal{R}}}{\epsilon} \cdot \dim_1\big(\widetilde{\mathcal{R}}, \epsilon/2\big), \frac{\kappa^2 \cdot \iota_\mathcal{R}}{\epsilon^2} \cdot \dim_2\big(\widetilde{\mathcal{R}}, \epsilon/2\big) \right\} \right)$$

*where we denote $\iota_\mathcal{P} := \log(N_{[]}((HT|\mathcal{S}|)^{-1}, \mathcal{P}, \|\cdot\|_\infty))$ and $\iota_\mathcal{R} := \log(N_{[]}(\overline{\kappa}(2T)^{-1}, \widetilde{\mathcal{R}}, \|\cdot\|_\infty))$.*

Similar to the analysis of Algorithm 1, we study the balance between the Bayesian regret and the query complexity by setting $\epsilon = T^{-\beta}$ for some $\beta \leq 1/2$. Then, we can simplify the bounds into $\text{BayesRegret}_T = \widetilde{O}(T^{1-\beta})$ and

$$\text{BayesQueries}_T = \widetilde{O}\left( \min \left\{ \underbrace{T^{\beta+\frac{1}{2}} \cdot \dim_1\left(\widetilde{\mathcal{R}}, \epsilon/2\right)}_{\text{(i)}}, \; \underbrace{T^{2\beta} \cdot \dim_2\left(\widetilde{\mathcal{R}}, \epsilon/2\right)}_{\text{(ii)}} \right\} \right)$$

where we have hidden factors except $T$ and the eluder dimension for brevity. We see that there is again a tradeoff in $T$ between the Bayesian regret and the query complexity, similar to the one in Theorem 4.2. Term (ii) demonstrates that the tradeoff in $T$ is again *optimal*, evidenced by the lower bound (Theorem 4.3). Moreover, term (i) further improves the dependence on the eluder dimension (recalling that $\ell_1$-norm version is smaller than the $\ell_2$-norm version). However, the $T$-dependence is worse. It is desired to derive a query complexity upper bound that scales as $\widetilde{O}(T^{2\beta} \cdot \dim_1(\widetilde{\mathcal{R}}, \epsilon/2))$, attaining the favorable dependence on both $T$ and the eluder dimension. We leave it as future work.

We emphasize that the Bayesian regret analysis in Theorem 5.4 is not a simple extension of previous TS works. We highlight four main differences: (1) The feedback is preference-based, which necessitates a new Bayesian regret decomposition:

$$\text{BayesRegret}_T = \underbrace{\sum_{t=0}^{T} \mathbb{E}\left[ V_{r_t, P_t}^{\pi_t^0} - V_{r_t, P^\star}^{\pi_t^0} \right]}_{\text{T}_{\text{model}}} + \underbrace{\sum_{t=0}^{T} \mathbb{E}\left[ \left( V_{r_t, P^\star}^{\pi_t^0} - V_{r_t, P^\star}^{\pi_t^1} \right) - \left( V_{r^\star, P^\star}^{\pi_t^0} - V_{r^\star, P^\star}^{\pi_t^1} \right) \right]}_{\text{T}_{\text{reward}}}.$$

Here $\text{T}_{\text{model}}$ and $\text{T}_{\text{reward}}$ are the respective regret incurred due to model and reward misspecification. We highlight that $\text{T}_{\text{reward}}$ characterizes the misspecification in terms of the *reward difference* between $\pi_t^0$ and $\pi_t^1$, which is different from the standard Bayesian RL. (2) Unlike prior works (Russo & Van Roy, 2014), we do not rely on upper confidence bounds (UCB) or optimism. Instead, we construct version spaces by classic MLE generalization bound. Taking the reward learning as an example, given the preference data $\{\tau_i^0, \tau_i^1, o_i\}_{i=1}^{t-1}$, we construct the version space at round $t$ as

$$\mathcal{V}_t = \left\{ r \in \mathcal{R} \; : \; \sum_{i=1}^{t-1} d_{\text{TV}}^2 \left( \Pr(\cdot \,|\, \tau_i^1, \tau_i^0, \widehat{r}_t), \, \Pr(\cdot \,|\, \tau_i^1, \tau_i^0, r) \right) \leq \beta \right\}$$

where $\widehat{r}_t := \arg\max_r \log \sum_{i=1}^{t-1} \Pr(o_i \,|\, \tau_i^1, \tau_i^0, r)$ is the MLE from the preference data and $\beta$ is tuned appropriately to ensure $r^\star \in \mathcal{V}_t$ with high probability. We then show the posterior probability of $r_t$ and $r^\star$ not belonging to $\mathcal{V}_t$ is small. (3) Our analysis uses the tighter $\ell_1$-norm eluder dimension, which is strictly better than the $\ell_2$-norm eluder dimension used in prior work. (4) We also equipped it with a randomized active learning procedure for query complexity minimization.

**Computation.** The computational bottleneck of Algorithm 2 lies in the computation of the posterior distribution (Line 3). Prior TS works have used Bootstrapping to approximate posterior sampling (Osband et al., 2016a; 2023) and achieved competitive performance in common RL benchmarks.

**Non-Markovian reward.** Algorithm 2 can also be applied to non-Markovian reward (i.e., reward model is trajectory-wise) without any change. Here we consider Markovian reward for the consistency with Algorithm 1 and for the purpose of using a standard planning oracle for computing an optimal policy from a reward and transition model. While non-Markovian reward is more general, it is unclear how to solve the planning problem efficiently even in tabular MDPs. This computational intractability makes non-Markovian rewards not easily applicable in practice.

**Extension to SEC.** In Appendix C.4, we extend the eluder dimension in Theorem 5.4 to the Sequential Extrapolation Coefficient (SEC) (Xie et al., 2022), which is more general.

## 6 CONCLUSION

We use randomization to design algorithms for RL with preference-based feedback. Randomization allows us to minimize regret and query complexity while at the same time maintaining computation efficiency. For linear models, our algorithms achieve a near-optimal balance between the worst-case reward regret and query complexity with computational efficiency. For models beyond linear, using eluder dimension, we present a TS-inspired algorithm that balances Bayesian regret and Bayesian query complexity nearly optimally.

ACKNOWLEDGEMENTS

Both RW and WS acknowledge support from NSF IIS-2154711, NSF CAREER 2339395, and Cornell Infosys Collaboration.

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

# A    COMPUTATIONAL CHALLENGES IN RL WITH PREFERENCE-BASED FEEDBACK

For RL with preference-based feedback, there is currently no algorithm that achieves sublinear worst-case regret and computational efficiency simultaneously, even for tabular MDPs. In this section, we discuss the challenges that hindered previous works from being computationally efficient. Specifically, the reasons are twofold:

*(1) Trajectory-wise information.* The feedback is trajectory-wise, meaning that we only receive information about cumulative rewards instead of per-step rewards. In this case, there is no longer a ground truth per-step reward signal. To see this, consider an MDP with two steps and a trajectory with a cumulative reward of 1. Then, we cannot decide the respective per-step reward for the two steps — it could be that the first step has reward 1 and the second has reward 0, or both have reward 0.5. The preference-based feedback is strictly harder than trajectory-wise reward feedback, and thus the same issue persists. This invalidates all algorithms relying on per-step reward information (e.g., UCBVI (Azar et al., 2017)) and necessitates leveraging feedback signal at a trajectory level. However, trajectory-level approaches typically entail maintaining a version space via trajectory constraints (Saha et al., 2023; Chen et al., 2022; Zhan et al., 2023a), and the computational complexity of searching within this version space is at least exponential in the length of the episode. Some works circumvent this computational obstacle by additional assumptions (e.g., explorability (Chatterji et al., 2021)), which are restrictive and do not generally hold even in tabular MDPs.

*(2) Preference-based information.* The feedback relies on a pair of policies. Standard algorithms based on a single policy become computationally intractable when adapting to this setting since optimizing over a pair of policies simultaneously is qualitatively different. For example, Zhan et al. (2023b); Saha et al. (2023) use the idea from optimal design and need the computation oracle: $\arg\max_{\pi,\pi'\in\Pi} \|\mathbb{E}_{s,a\sim\pi}\phi(s,a) - \mathbb{E}_{s,a\sim\pi'}\phi(s,a)\|_A$ for some positive definite matrix $A$. Here $\|x\|_A^2 := x^\top A x$, and $\phi$ is some state-action wise feature.[3] It is unclear how to implement this oracle since standard planning approaches relying on dynamic programming cannot be applied here. Additionally, these methods also require actively maintaining a policy space $\Pi$ by eliminating potentially sub-optimal policies. The policy class can be exponentially large even in tabular settings, so it is unclear how to maintain a valid policy space in a computationally tractable manner.

These challenges motivate us to devise a novel algorithm using a technique distinct from previous approaches. Our solution centers around the concept of *randomization*, which allows us to balance exploration and exploitation and thus enable standard efficient computation oracles (e.g., DP-style planning oracle like value iteration).

# B    PROOF OF THEOREM 4.2

## B.1    NOTATIONS

We define some symbols and their values in Table 1. We have categorized them into four classes for the ease of reference.

The concept of covering number is defined below, which will be used to bound the statistical error of our algorithm.

**Definition B.1** (Covering number). *Consider a function class $\mathcal{F} \subseteq \mathcal{X} \to \mathbb{R}$. The $\omega$-cover of a function $\widehat{f} \in \mathcal{F}$ is defined as the set of functions $f \in \mathcal{F}$ for which $\|f - \widehat{f}\| \leq \omega$. The covering number of $\mathcal{F}$ w.r.t. the metric $\|\cdot\|$ denoted by $N(\omega, \mathcal{F}, \|\cdot\|)$ is the minimum number of $\omega$-covers needed to cover $\mathcal{F}$.*

---

[3]These prior work typically assume trajectory wise feature $\phi(\tau)$ for a state-action wise trajectory $\tau$. However, even when specializing to state-action-wise features, these algorithms are still not computationally tractable even in tabular MDPs.

Table 1: Symbols and their respective values.

| Symbol | Value |
|---|---|
| *(1) Error components* | |
| $\xi_{\mathrm{r},t}$ | $\overline{\theta}_{\mathrm{r},t} - \widehat{\theta}_{\mathrm{r},t}$ |
| $\xi_{\mathrm{P},t,h}$ | $\overline{\theta}_{\mathrm{P},t,h} - \widehat{\theta}_{\mathrm{P},t,h}$ |
| $\lambda_{t,h}$ | $\lambda \Sigma_{t-1,h}^{-1} \int_{s'} \mu_h^\star(s') \overline{V}_{t,h+1}(s') \,\mathrm{d}s'$ |
| $\eta_{\mathrm{r},t}$ | $\widehat{\theta}_{\mathrm{r},t} - \theta_{\mathrm{r}}^\star$ |
| $\eta_{\mathrm{P},t,h}$ | $\Sigma_{t-1,h}^{-1} \left( \sum_{i=1}^{t-1} \phi(s_{i,h}, a_{i,h}) \left( \overline{V}_{t,h+1}(s_{i,h+1}) - \mathbb{E}_{s_{i,h+1}} \left[ \overline{V}_{t,h+1}(s_{i,h+1}) \,\middle|\, s_{i,h}, a_{i,h} \right] \right) \right)$ |
| $\theta_{\mathrm{P},t,h}^\star$ | $\int_{s'} \mu_h^\star(s') \left[ \overline{V}_{t,h+1}(s') \right] \,\mathrm{d}s'$ |
| *(2) Statistical upper bounds* | |
| $\epsilon_{\mathrm{r},\xi}$ | $\sigma_{\mathrm{r}} \sqrt{2d \log(2dT/\delta)}$ |
| $\epsilon_{\mathrm{r},\eta}$ | $\sqrt{80\kappa d \log\left(24BT^2/(\overline{\kappa}\delta)\right) + 168B^2 d \log(6BT^2/\delta) + 4\lambda B^2}$ |
| $V_{\max}$ | $H(2 + (\epsilon_{\mathrm{r},\xi} + \epsilon_{\mathrm{r},\eta})/\sqrt{\lambda})$ |
| $\epsilon_\lambda$ | $V_{\max} \sqrt{\lambda d}$ |
| $\epsilon_{\mathrm{P},\xi}$ | $\sigma_{\mathrm{P}} \sqrt{2d \log(2dHT/\delta)}$ |
| $\iota_\epsilon$ | $\log\left( 12HT^2(T+\lambda) V_{\max} \left( B + \frac{2V_{\max}\sqrt{dt} + \epsilon_{\mathrm{P},\xi} + \epsilon_{\mathrm{r},\xi}}{\sqrt{\lambda}} \right) \right)$ |
| $\chi$ | (defined in Lemma B.3) |
| $\epsilon_{\mathrm{P},\eta}'$ | $\frac{6}{\sqrt{\lambda}} + 16dV_{\max} \sqrt{\iota_\epsilon - \log\left((\alpha_{\mathrm{U}} - \alpha_{\mathrm{L}})\delta\lambda\right)}$ |
| $\epsilon_{\mathrm{P},\eta}$ | $\chi \cdot \left( \frac{6}{\sqrt{\lambda}} + 16dV_{\max} \sqrt{\iota_\epsilon - \log\left(\delta\lambda\right)} \right)$ |
| *(3) Value cutoff* | |
| $\boldsymbol{L}_{t,h}(s)$ | $\mathbb{1}\{ \|\phi(s, \pi_{t,h}(s))\|_{\Sigma_{t-1,h}^{-1}} \leq \alpha_{\mathrm{L}} \}$ |
| $\boldsymbol{L}_{t,h}^{\complement}(s)$ | $\mathbb{1}\{ \|\phi(s, \pi_{t,h}(s))\|_{\Sigma_{t-1,h}^{-1}} > \alpha_{\mathrm{L}} \}$ |
| $L_{\max}$ | $\frac{2dH}{\alpha_L^2} \cdot \log\left(\frac{\lambda+T}{\lambda}\right)$ |
| *(4) Hyperparameters* | |
| $\sigma_{\mathrm{r}}$ | $\epsilon_{\mathrm{r},\eta}$ |
| $\sigma_{\mathrm{P}}$ | $(\epsilon_{\mathrm{P},\eta} + \epsilon_\lambda)\sqrt{H}$ |
| $\alpha_{\mathrm{U}}$ | $(\epsilon_{\mathrm{P},\xi} + \epsilon_{\mathrm{P},\eta} + \epsilon_\lambda)^{-1}$ |
| $\alpha_{\mathrm{L}}$ | $(\epsilon_{\mathrm{P},\xi} + \epsilon_{\mathrm{P},\eta} + \epsilon_\lambda)^{-1}/2$ |
| $\lambda$ | $1$ |

## B.2 Supporting Lemmas

**Lemma B.2** (Covering number of Euclidean balls). *(Pollard, 1990) Let $\Theta_B := \{\theta \in \mathbb{R}^d : \|\theta\|_2 \leq B\}$. Then we have*
$$N(\omega, \Theta_B, \|\cdot\|_2) \leq (3B/\omega)^d.$$

**Lemma B.3.** *There exists $\chi = O(\log(\epsilon_{P,\xi} + \epsilon_{P,\eta} + \epsilon_\lambda))$ such that $\epsilon'_{P,\eta} \leq \epsilon_{P,\eta}$ (recalling that $\chi$ is in the definition of $\epsilon_{P,\eta}$).*

*Proof of Lemma B.3.* By definition, we have
$$\epsilon'_{P,\eta} = \frac{6}{\sqrt{\lambda}} + 16 d V_{\max} \sqrt{\iota_\epsilon - \log\left((\alpha_U - \alpha_L)\delta\lambda\right)}$$
$$= \frac{6}{\sqrt{\lambda}} + 16 d V_{\max} \sqrt{\iota_\epsilon - \log(\delta\lambda) - \log(\alpha_U - \alpha_L)}$$
$$= \frac{6}{\sqrt{\lambda}} + 16 d V_{\max} \sqrt{\iota_\epsilon - \log(\delta\lambda) + \log 2(\epsilon_{P,\xi} + \epsilon_{P,\eta} + \epsilon_\lambda)}$$
$$\leq \chi \left( \frac{6}{\sqrt{\lambda}} + 16 d V_{\max} \sqrt{\iota_\epsilon - \log(\delta\lambda)} \right)$$
$$= \epsilon_{P,\eta}.$$

Here the third equality is by the definition of $\alpha_L$ and $\alpha_U$. The inequality holds by setting $\chi = O(\log(\epsilon_{P,\xi} + \epsilon_{P,\eta} + \epsilon_\lambda))$ to be large enough. □

**Lemma B.4.** *It holds that $\eta_{P,t,h} + \lambda_{t,h} = \widehat{\theta}_{P,t,h} - \theta^\star_{P,t,h}$ for any $t \in [T]$ and $h \in [H]$.*

*Proof of Lemma B.4.* By definition, we have
$$\eta_{P,t,h} + \lambda_{t,h}$$
$$= \Sigma^{-1}_{t-1,h} \left( \sum_{i=1}^{t-1} \phi(s_{i,h}, a_{i,h}) \left( \overline{V}_{t,h+1}(s_{i,h+1}) - \mathbb{E}_{s_{h+1}} \left[ \overline{V}_{t,h+1}(s_{i,h+1}) \,\middle|\, s_{i,h}, a_{i,h} \right] \right) \right)$$
$$+ \lambda \Sigma^{-1}_{t-1,h} \int_{s'} \mu^\star_h(s') \overline{V}_{t,h+1}(s') \, ds'$$
$$= \widehat{\theta}_{P,t,h} - \Sigma^{-1}_{t-1,h} \left( \sum_{i=1}^{t-1} \phi(s_{i,h}, a_{i,h})\phi(s_{i,h}, a_{i,h}) + \lambda I \right)^\top \int_{s'} \mu^\star_h(s') \left[ \overline{V}_{t,h+1}(s') \right] \, ds'$$
$$= \widehat{\theta}_{P,t,h} - \int_{s'} \mu^\star_h(s') \left[ \overline{V}_{t,h+1}(s') \right] \, ds'$$
$$= \widehat{\theta}_{P,t,h} - \theta^\star_{P,t,h}.$$
□

The following lemma is adapted from Zanette et al. (2020, Lemma 1) to our setting.

**Lemma B.5** (One-step decomposition). *For any $t \in [T], h \in [H], s \in \mathcal{S}, a \in \mathcal{A}$, and policy $\pi$, we have*
$$\phi(s,a)^\top \left( \overline{\theta}_{r,t} + \overline{\theta}_{P,t,h} \right) - Q^\pi_h(s,a)$$
$$= \phi(s,a)^\top (\xi_{r,t} + \xi_{P,t,h} + \eta_{r,t} + \eta_{P,t,h} - \lambda_{t,h}) + \mathbb{E}_{s'} \left[ \overline{V}_{t,h+1}(s') - V^\pi_{h+1}(s') \,\middle|\, s,a \right].$$

*Proof.* We have
$$\phi(s,a)^\top \left( \overline{\theta}_{r,t} + \overline{\theta}_{P,t,h} \right) - Q^\pi_h(s,a)$$
$$= \phi(s,a)^\top \left( \overline{\theta}_{r,t} + \overline{\theta}_{P,t,h} \right) - \phi(s,a)^\top \theta^\star_r - \mathbb{E}_{s'}[V^\pi_{h+1}(s') \,|\, s,a]$$
$$= \phi(s,a)^\top (\overline{\theta}_{r,t} - \widehat{\theta}_{r,t}) + \phi(s,a)^\top (\overline{\theta}_{P,t,h} - \widehat{\theta}_{P,t,h}) + \phi(s,a)^\top (\widehat{\theta}_{r,t} - \theta^\star_r)$$
$$+ \phi(s,a)^\top \widehat{\theta}_{P,t,h} - \mathbb{E}_{s'}[V^\pi_{h+1}(s') \,|\, s,a].$$

For the last but one term, we note that

$$\phi(s,a)^\top \widehat{\theta}_{\mathrm{P},t,h}$$

$$= \phi(s,a)^\top \Sigma_{t-1,h}^{-1} \left( \sum_{i=1}^{t-1} \phi(s_{i,h}, a_{i,h}) \overline{V}_{t,h+1}(s_{i,h+1}) \right)$$

$$= \phi(s,a)^\top \Sigma_{t-1,h}^{-1} \left( \sum_{i=1}^{t-1} \phi(s_{i,h}, a_{i,h}) \mathop{\mathbb{E}}_{s_{h+1}} \left[ \overline{V}_{t,h+1}(s_{i,h+1}) \,\middle|\, s_{i,h}, a_{i,h} \right] \right) + \phi(s,a)^\top \eta_{\mathrm{P},t,h}$$

$$= \phi(s,a)^\top \Sigma_{t-1,h}^{-1} \left( \sum_{i=1}^{t-1} \phi(s_{i,h}, a_{i,h}) \phi(s_{i,h}, a_{i,h})^\top \int_{s'} \mu_h^\star(s') \overline{V}_{t,h+1}(s') \, \mathrm{d}s' \right) + \phi(s,a)^\top \eta_{\mathrm{P},t,h}$$

$$= \phi(s,a)^\top \int_{s'} \mu_h^\star(s') \overline{V}_{t,h+1}(s') \, \mathrm{d}s' - \lambda \phi(s,a)^\top \Sigma_{t-1,h}^{-1} \int_{s'} \mu_h^\star(s') \overline{V}_{t,h+1}(s') \, \mathrm{d}s' + \phi(s,a)^\top \eta_{\mathrm{P},t,h}$$

$$= \mathop{\mathbb{E}}_{s'} \left[ \overline{V}_{t,h+1}(s') \,\middle|\, s,a \right] - \phi(s,a)^\top \lambda_{t,h} + \phi(s,a)^\top \eta_{\mathrm{P},t,h}.$$

Plugging this back, we get

$$\phi(s,a)^\top \left( \overline{\theta}_{\mathrm{r},t} + \overline{\theta}_{\mathrm{P},t,h} \right) - Q_h^\pi(s,a)$$

$$= \phi(s,a)^\top (\overline{\theta}_{\mathrm{r},t} - \widehat{\theta}_{\mathrm{r},t}) + \phi(s,a)^\top (\overline{\theta}_{\mathrm{P},t,h} - \widehat{\theta}_{\mathrm{P},t,h}) + \phi(s,a)^\top (\widehat{\theta}_{\mathrm{r},t} - \theta_{\mathrm{r}}^\star)$$

$$\quad + \mathop{\mathbb{E}}_{s'} \left[ \overline{V}_{t,h+1}(s') - V_{h+1}^\pi(s') \,\middle|\, s,a \right] - \phi(s,a)^\top \lambda_{t,h} + \phi(s,a)^\top \eta_{\mathrm{P},t,h}$$

$$= \phi(s,a)^\top (\xi_{\mathrm{r},t} + \xi_{\mathrm{P},t,h} + \eta_{\mathrm{r},t} + \eta_{\mathrm{P},t,h} - \lambda_{t,h}) + \mathop{\mathbb{E}}_{s'} \left[ \overline{V}_{t,h+1}(s') - V_{h+1}^\pi(s') \,\middle|\, s,a \right].$$

This completes the proof. □

**Lemma B.6.** *For any $t \in [T]$, if $Z_t = 0$, then we have*

$$\|\phi(\tau_t^0) - \phi(\tau_t^1)\|_{\Sigma_{t-1}^{-1}} \le \epsilon \sqrt{\pi}/(2\sigma_{\mathrm{r}}).$$

*Proof.* By the definition of $Z_t$, we have

$$\epsilon \ge \mathop{\mathbb{E}}_{\theta_0, \theta_1 \sim \mathcal{N}(\widehat{\theta}_{\mathrm{r},t}, \sigma_{\mathrm{r}}^2 \Sigma_{t-1}^{-1})} |(\phi(\tau_t^0) - \phi(\tau_t^1))^\top (\theta_0 - \theta_1)|$$

$$= \mathop{\mathbb{E}}_{u_0, u_1 \sim \mathcal{N}(0, I_d)} |(\phi(\tau_t^0) - \phi(\tau_t^1))^\top \sigma_{\mathrm{r}} \Sigma_{t-1}^{-1/2} (u_0 - u_1)|$$

$$= \mathop{\mathbb{E}}_{u \sim \mathcal{N}(0, I_d)} |(\phi(\tau_t^0) - \phi(\tau_t^1))^\top \sqrt{2} \sigma_{\mathrm{r}} \Sigma_{t-1}^{-1/2} u|$$

$$= \sigma_{\mathrm{r}} \sqrt{2} \mathop{\mathbb{E}}_{u \sim \mathcal{N}(0, I_d)} \sqrt{u^\top \Sigma_{t-1}^{-1/2} (\phi(\tau_t^0) - \phi(\tau_t^1))(\phi(\tau_t^0) - \phi(\tau_t^1))^\top \Sigma_{t-1}^{-1/2} u}$$

$$= \sigma_{\mathrm{r}} \sqrt{2} \mathop{\mathbb{E}}_{u \sim \mathcal{N}(0, I_d)} \|u\|_{\Sigma_{t-1}^{-1/2}(\phi(\tau_t^0) - \phi(\tau_t^1))(\phi(\tau_t^0) - \phi(\tau_t^1))^\top \Sigma_{t-1}^{-1/2}}.$$

We then apply Lemma D.9 and obtain

$$\epsilon \ge \sigma_{\mathrm{r}} \sqrt{2} \cdot \sqrt{\frac{2 \operatorname{tr} \left( \Sigma_{t-1}^{-1/2}(\phi(\tau_t^0) - \phi(\tau_t^1))(\phi(\tau_t^0) - \phi(\tau_t^1))^\top \Sigma_{t-1}^{-1/2} \right)}{\pi}}$$

$$= 2\sigma_{\mathrm{r}} \cdot \sqrt{\frac{(\phi(\tau_t^0) - \phi(\tau_t^1))^\top \Sigma_{t-1}^{-1/2} \Sigma_{t-1}^{-1/2} (\phi(\tau_t^0) - \phi(\tau_t^1))}{\pi}}$$

$$= \frac{2\sigma_{\mathrm{r}}}{\sqrt{\pi}} \cdot \|\phi(\tau_t^0) - \phi(\tau_t^1)\|_{\Sigma_{t-1}^{-1}}.$$

Hence, we complete the proof. □

## B.3 STATISTICAL UPPER BOUNDS

**Lemma B.7** (Gaussian noise). *Each of the following holds with probability at least $1 - \delta$:*

1. $\|\xi_{\mathrm{r},t}\|_{\Sigma_{t-1}} \leq \epsilon_{\mathrm{r},\xi}$ *for all* $t \in [T]$,

2. $\|\xi_{\mathrm{P},t,h}\|_{\Sigma_{t-1,h}} \leq \epsilon_{\mathrm{P},\xi}$ *for all* $t \in [T]$ *and* $h \in [H]$.

*Proof of Lemma B.7.* It simply follows from Lemma D.3 and the union bound. □

**Lemma B.8.** *Fix* $t \in [T]$. *It holds with probability at least* $1 - \delta$ *that*

$$\sum_{s=1}^{t-1} \left| \left( \phi(\tau_s^0) - \phi(\tau_s^1) \right)^\top \left( \theta_{\mathrm{r}}^\star - \widehat{\theta}_{\mathrm{r},t} \right) \right|^2$$

$$\leq 8 \sum_{s=1}^{t-1} \mathbb{E}_{\tau_s^0, \tau_s^1} \left| \left( \phi(\tau_s^0) - \phi(\tau_s^1) \right)^\top \left( \theta_{\mathrm{r}}^\star - \widehat{\theta}_{\mathrm{r},t} \right) \right|^2 + 168 B^2 d \log(3TB/\delta).$$

*Here the expectation is taken over the randomness of sampling* $\tau_s^0$ *and* $\tau_s^1$ *from* $\pi_s^0$ *and* $\pi_s^1$, *respectively.*

*Proof of Lemma B.8.* Let $\widetilde{\Theta_B}$ denote an $\omega$-covering of $\Theta_B$ with respect to $\| \cdot \|_2$ for some $\omega > 0$. By Lemma B.2, we have $|\widetilde{\Theta_B}| \leq (3B/\omega)^d$. Moreover, for any $s \in [t-1]$, by Cauchy-Schwarz inequality, it holds that

$$\left| \left( \phi(\tau_s^0) - \phi(\tau_s^1) \right)^\top \left( \theta_{\mathrm{r}}^\star - \widetilde{\theta} \right) \right|^2 \leq \left\| \phi(\tau_s^0) - \phi(\tau_s^1) \right\|_2^2 \cdot \left\| \theta_{\mathrm{r}}^\star - \widetilde{\theta} \right\|_2^2 \leq 16 B^2.$$

Then, by Lemma D.4 and union bound over $\widetilde{\Theta_B}$, it holds that

$$\sum_{s=1}^{t-1} \left| \left( \phi(\tau_s^0) - \phi(\tau_s^1) \right)^\top \left( \theta_{\mathrm{r}}^\star - \widetilde{\theta} \right) \right|^2$$

$$\leq 2 \sum_{s=1}^{t-1} \mathbb{E}_{\tau_s^0, \tau_s^1} \left| \left( \phi(\tau_s^0) - \phi(\tau_s^1) \right)^\top \left( \theta_{\mathrm{r}}^\star - \widetilde{\theta} \right) \right|^2 + 64 B^2 d \log(3B/(\omega\delta)) \tag{4}$$

for all $\widetilde{\theta} \in \widetilde{\Theta_B}$ with probability at least $1 - \delta$. Now we consider an arbitrary $\theta \in \Theta_B$ and let $\widetilde{\theta} \in \widetilde{\Theta_B}$ be the closest to $\theta$. Then, by triangle inequality and Cauchy-Schwarz inequality, we have the following two ineuqalities:

$$\left| \left( \phi(\tau_s^0) - \phi(\tau_s^1) \right)^\top (\theta_{\mathrm{r}}^\star - \theta) \right|^2$$

$$\leq 2 \left| \left( \phi(\tau_s^0) - \phi(\tau_s^1) \right)^\top \left( \theta_{\mathrm{r}}^\star - \widetilde{\theta} \right) \right|^2 + 2 \| \phi(\tau_s^0) - \phi(\tau_s^1) \|_2^2 \| \theta - \widetilde{\theta} \|_2^2$$

$$\leq 2 \left| \left( \phi(\tau_s^0) - \phi(\tau_s^1) \right)^\top \left( \theta_{\mathrm{r}}^\star - \widetilde{\theta} \right) \right|^2 + 8\omega^2,$$

and

$$\left| \left( \phi(\tau_s^0) - \phi(\tau_s^1) \right)^\top \left( \theta_{\mathrm{r}}^\star - \widetilde{\theta} \right) \right|^2$$

$$\leq 2 \left| \left( \phi(\tau_s^0) - \phi(\tau_s^1) \right)^\top (\theta_{\mathrm{r}}^\star - \theta) \right|^2 + 2 \| \phi(\tau_s^0) - \phi(\tau_s^1) \|_2^2 \| \theta - \widetilde{\theta} \|_2^2$$

$$\leq 2 \left| \left( \phi(\tau_s^0) - \phi(\tau_s^1) \right)^\top (\theta_{\mathrm{r}}^\star - \theta) \right|^2 + 8\omega^2$$

Applying these inequalities and (4), we get

$$\sum_{s=1}^{t-1} \left| \left( \phi(\tau_s^0) - \phi(\tau_s^1) \right)^\top (\theta_{\mathrm{r}}^\star - \theta) \right|^2$$

$$\leq 2 \sum_{s=1}^{t-1} \left| \left( \phi(\tau_s^0) - \phi(\tau_s^1) \right)^\top \left( \theta_{\mathrm{r}}^\star - \widetilde{\theta} \right) \right|^2 + 8\omega^2 t$$

$$\leq 4 \sum_{s=1}^{t-1} \mathop{\mathbb{E}}_{\tau_s^0, \tau_s^1} \left| \left( \phi(\tau_s^0) - \phi(\tau_s^1) \right)^\top \left( \theta_{\mathrm{r}}^\star - \widetilde{\theta} \right) \right|^2 + 128 B^2 d \log(3B/(\omega\delta)) + 8\omega^2 t$$

$$\leq 8 \sum_{s=1}^{t-1} \mathop{\mathbb{E}}_{\tau_s^0, \tau_s^1} \left| \left( \phi(\tau_s^0) - \phi(\tau_s^1) \right)^\top (\theta_{\mathrm{r}}^\star - \theta) \right|^2 + 128 B^2 d \log(3B/(\omega\delta)) + 40\omega^2 t$$

$$\leq 8 \sum_{s=1}^{t-1} \mathop{\mathbb{E}}_{\tau_s^0, \tau_s^1} \left| \left( \phi(\tau_s^0) - \phi(\tau_s^1) \right)^\top (\theta_{\mathrm{r}}^\star - \theta) \right|^2 + 168 B^2 d \log(3TB/\delta)$$

where the last step holds by setting $\omega = 1/t$. $\qquad\square$

**Lemma B.9** (Reward estimation error). *It holds with probability at least $1 - \delta$ that $\|\eta_{\mathrm{r},t}\|_{\Sigma_{t-1}} \leq \epsilon_{\mathrm{r},\eta}$ for all $t \in [T]$.*

*Proof of Lemma B.9.* Since $\widehat{\theta}_{\mathrm{r},t}$ is the MLE, by Lemma D.10, we have

$$\sum_{s=1}^{t-1} \mathop{\mathbb{E}}_{\tau_s^0, \tau_s^1} d_{\mathrm{TV}}^2 \left( \Pr\left( \cdot \mid \tau_s^0, \tau_s^1, \theta_{\mathrm{r}}^\star \right), \Pr\left( \cdot \mid \tau_s^0, \tau_s^1, \widehat{\theta}_{\mathrm{r},t} \right) \right) \leq 10 \log \left( N_{[]}\left( (2T)^{-1}, \overline{\mathcal{R}}, \|\cdot\|_\infty \right)/\delta \right) \tag{5}$$

with probability at least $1 - \delta$, where $\overline{\mathcal{R}}$ denotes the set of probability distributions of preference feedback and $\Pr(\cdot)$ denotes the preference feedback generating probability. Now we upper bound the covering number of $\overline{\mathcal{R}}$ by the covering number of $\Theta_B$. We notice the following:

$$\sup_{o, \tau^0, \tau^1} \left| \Pr\left( o \mid \tau^0, \tau^1, \theta_{\mathrm{r}}^\star \right) - \Pr\left( o \mid \tau^0, \tau^1, \widehat{\theta}_{\mathrm{r},t} \right) \right|$$

$$= \sup_{\tau^0, \tau^1} \left| \Phi\left( \left( \phi(\tau^0) - \phi(\tau^1) \right)^\top \theta_{\mathrm{r}}^\star \right) - \Phi\left( \left( \phi(\tau^0) - \phi(\tau^1) \right)^\top \widehat{\theta}_{\mathrm{r},t} \right) \right|$$

$$\leq \overline{\kappa}^{-1} \sup_{\tau^0, \tau^1} \left| \left( \phi(\tau^0) - \phi(\tau^1) \right)^\top (\theta_{\mathrm{r}}^\star - \widehat{\theta}_{\mathrm{r},t}) \right|$$

$$\leq \overline{\kappa}^{-1} \sup_{\tau^0, \tau^1} \| \phi(\tau^0) - \phi(\tau^1) \|_2 \| \theta_{\mathrm{r}}^\star - \widehat{\theta}_{\mathrm{r},t} \|_2$$

$$\leq 2\overline{\kappa}^{-1} \| \theta_{\mathrm{r}}^\star - \widehat{\theta}_{\mathrm{r},t} \|_2$$

where the equality holds since the feedback $o$ is binary, the first inequality is Lemma D.1, and the last inequality is by the condition that $\|\phi(\tau)\|_2 \leq 1$ for any trajectory $\tau$. This means that an $\omega$-covering of the space of parameter $\theta$ implies a $2\omega\overline{\kappa}^{-1}$-covering of the space of the probability distribution of the preference feedbacks. Hence, (5) can be further upper bounded by

$$\sum_{s=1}^{t-1} \mathop{\mathbb{E}}_{\tau_s^0, \tau_s^1} d_{\mathrm{TV}}^2 \left( \Pr\left( \cdot \mid \tau_s^0, \tau_s^1, \theta_{\mathrm{r}}^\star \right), \Pr\left( \cdot \mid \tau_s^0, \tau_s^1, \widehat{\theta}_{\mathrm{r},t} \right) \right)$$

$$\leq 10 \log \left( N_{[]}\left( (2T)^{-1}, \overline{\mathcal{R}}, \|\cdot\|_\infty \right)/\delta \right)$$

$$\leq 10 \log \left( N\left( \overline{\kappa}/(4T), \Theta_B, \|\cdot\|_2 \right)/\delta \right)$$

$$\leq 10 d \log \left( 12 BT/(\overline{\kappa}\delta) \right) \tag{6}$$

where the last inequality is Lemma B.2. For the left side, we note that

$$\sum_{s=1}^{t-1} \mathop{\mathbb{E}}_{\tau_s^0, \tau_s^1} d_{\mathrm{TV}}^2 \left( \Pr \left( \cdot \,\big|\, \tau_s^0, \tau_s^1, \theta_{\mathrm{r}}^\star \right), \Pr \left( \cdot \,\big|\, \tau_s^0, \tau_s^1, \widehat{\theta}_{\mathrm{r},t} \right) \right)$$

$$= \sum_{s=1}^{t-1} \mathop{\mathbb{E}}_{\tau_s^0, \tau_s^1} \left| \Phi \left( \left( \phi(\tau_s^0) - \phi(\tau_s^1) \right)^\top \theta_{\mathrm{r}}^\star \right) - \Phi \left( \left( \phi(\tau_s^0) - \phi(\tau_s^1) \right)^\top \widehat{\theta}_{\mathrm{r},t} \right) \right|^2$$

$$\geq \kappa^{-1} \sum_{s=1}^{t-1} \mathop{\mathbb{E}}_{\tau_s^0, \tau_s^1} \left| \left( \phi(\tau_s^0) - \phi(\tau_s^1) \right)^\top \left( \theta_{\mathrm{r}}^\star - \widehat{\theta}_{\mathrm{r},t} \right) \right|^2$$

$$\geq \frac{1}{8} \kappa^{-1} \sum_{s=1}^{t-1} \left| \left( \phi(\tau_s^0) - \phi(\tau_s^1) \right)^\top \left( \theta_{\mathrm{r}}^\star - \widehat{\theta}_{\mathrm{r},t} \right) \right|^2 - 21 \kappa^{-1} B^2 d \log(3TB/\delta) \qquad (7)$$

with probability at least $1 - \delta$, where the first inequality is by Lemma D.1 and the last inequality is by Lemma B.8. Furthermore, we have

$$\sum_{s=1}^{t-1} \left| \left( \phi(\tau_s^0) - \phi(\tau_s^1) \right)^\top \left( \theta_{\mathrm{r}}^\star - \widehat{\theta}_{\mathrm{r},t} \right) \right|^2$$

$$= \left( \theta_{\mathrm{r}}^\star - \widehat{\theta}_{\mathrm{r},t} \right)^\top \sum_{s=1}^{t-1} \left( \phi(\tau_s^0) - \phi(\tau_s^1) \right) \left( \phi(\tau_s^0) - \phi(\tau_s^1) \right)^\top \left( \theta_{\mathrm{r}}^\star - \widehat{\theta}_{\mathrm{r},t} \right)$$

$$= \left( \theta_{\mathrm{r}}^\star - \widehat{\theta}_{\mathrm{r},t} \right)^\top \Sigma_{t-1} \left( \theta_{\mathrm{r}}^\star - \widehat{\theta}_{\mathrm{r},t} \right) - \left( \theta_{\mathrm{r}}^\star - \widehat{\theta}_{\mathrm{r},t} \right)^\top \lambda I \left( \theta_{\mathrm{r}}^\star - \widehat{\theta}_{\mathrm{r},t} \right) \geq \|\eta_{\mathrm{r},t}\|_{\Sigma_{t-1}}^2 - 4\lambda B^2 \qquad (8)$$

Putting (6), (7), and (8) together, we get

$$\|\eta_{\mathrm{r},t}\|_{\Sigma_{t-1}}^2 \leq 80 \kappa d \log \left( 12BT/(\overline{\kappa}\delta) \right) + 168 B^2 d \log(3TB/\delta) + 4\lambda B^2$$

with probability at least $1 - 2\delta$. Adjusting $\delta$ to $\delta/2$ and taking the union bound over $t \in [T]$ yields the desired result. □

**Lemma B.10** (One-step transition estimation error). *Assume the events defined in Lemma B.7 hold. Then, the following claim holds for all $t \in [T]$ and $h \in [H]$ with probability at least $1 - \delta$: if $\max_s |\overline{V}_{t,h+1}(s)| \leq V_{\max}$, it holds that $\|\eta_{\mathrm{P},t,h}\|_{\Sigma_{t-1,h}} \leq \epsilon_{\mathrm{P},\eta}$.*

*Proof of Lemma B.10.* By definition, we have

$$\|\eta_{\mathrm{P},t,h}\|_{\Sigma_{t-1,h}} = \left\| \sum_{i=1}^{t-1} \phi(s_{i,h}, a_{i,h}) \left( \overline{V}_{t,h+1}(s_{i,h+1}) - \mathop{\mathbb{E}}_{s_{i,h+1}} \left[ \overline{V}_{t,h+1}(s_{i,h+1}) \,\big|\, s_{i,h}, a_{i,h} \right] \right) \right\|_{\Sigma_{t-1,h}^{-1}}.$$

Towards an upper bound of the above, we will conduct some covering arguments. To that end, we first derive the upper bounds of norms of $\overline{\theta}_{\mathrm{r},t}$ and $\overline{\theta}_{\mathrm{P},t,h}$.

**Bounding the $\ell_2$-norms of $\overline{\theta}_{\mathrm{r},t}$ and $\overline{\theta}_{\mathrm{P},t,h}$.** For $\overline{\theta}_{\mathrm{r},t}$, applying triangle inequality, we have $\|\overline{\theta}_{\mathrm{r},t}\|_2 \leq \|\xi_{\mathrm{r},t}\|_2 + \|\widehat{\theta}_{\mathrm{r},t}\|_2 \leq \|\xi_{\mathrm{r},t}\|_2 + B$. For $\|\xi_{\mathrm{r},t}\|_2$, we have

$$\|\xi_{\mathrm{r},t}\|_2 = \sqrt{\xi_{\mathrm{r},t}^\top \Sigma_{t-1}^{-1/2} \Sigma_{t-1}^{1/2} \xi_{\mathrm{r},t}} \leq \sqrt{\|\xi_{\mathrm{r},t}\|_{\Sigma_{t-1}} \|\xi_{\mathrm{r},t}\|_{\Sigma_{t-1}^{-1}}} \leq \sqrt{\epsilon_{\mathrm{r},\xi} \cdot \|\xi_{\mathrm{r},t}\|_2/\sqrt{\lambda}}$$

where in the last step we applied Lemma B.7 and the fact that $\|\xi_{\mathrm{r},t}\|_{\Sigma_{t-1}^{-1}} \leq \|\xi_{\mathrm{r},t}\|_2 \|\Sigma_{t-1}^{-1/2}\|_2 \leq \|\xi_{\mathrm{r},t}\|_2/\sqrt{\lambda}$. This implies that $\|\xi_{\mathrm{r},t}\|_2 \leq \epsilon_{\mathrm{r},\xi}/\sqrt{\lambda}$, and thus, $\|\overline{\theta}_{\mathrm{r},t}\|_2 \leq \epsilon_{\mathrm{r},\xi}/\sqrt{\lambda} + B$.

For $\overline{\theta}_{\mathrm{P},t,h}$, we have $\|\overline{\theta}_{\mathrm{P},t,h}\|_2 \leq \|\xi_{\mathrm{P},t,h}\|_2 + \|\widehat{\theta}_{\mathrm{P},t,h}\|_2$. For $\|\xi_{\mathrm{P},t,h}\|_2$, following a similar argument as above, we have

$$\|\xi_{\mathrm{P},t,h}\|_2 \leq \sqrt{\|\xi_{\mathrm{P},t,h}\|_{\Sigma_{t-1,h}} \|\xi_{\mathrm{P},t,h}\|_{\Sigma_{t-1,h}^{-1}}} \leq \sqrt{\epsilon_{\mathrm{P},\xi} \cdot \|\xi_{\mathrm{P},t,h}\|_2/\sqrt{\lambda}},$$

which implies that $\|\xi_{\mathrm{P},t,h}\|_2 \le \epsilon_{\mathrm{P},\xi}/\sqrt{\lambda}$. For $\|\widehat{\theta}_{\mathrm{P},t,h}\|_2$, we have

$$
\begin{aligned}
\|\widehat{\theta}_{\mathrm{P},t,h}\|_2 &= \left\| \Sigma_{t-1,h}^{-1/2} \Sigma_{t-1,h}^{-1/2} \left( \sum_{i=1}^{t-1} \phi(s_{i,h}, a_{i,h}) \overline{V}_{t,h+1}(s_{i,h+1}) \right) \right\|_2 \\
&\le \|\Sigma_{t-1,h}^{-1/2}\|_2 \cdot \left\| \sum_{i=1}^{t-1} \phi(s_{i,h}, a_{i,h}) \overline{V}_{t,h+1}(s_{i,h+1}) \right\|_{\Sigma_{t-1,h}^{-1}} \\
&\le \sqrt{\lambda^{-1}} \cdot \sqrt{t} \cdot \sqrt{ \sum_{i=1}^{t-1} \left\| \phi(s_{i,h}, a_{i,h}) \overline{V}_{t,h+1}(s_{i,h+1}) \right\|_{\Sigma_{t-1,h}^{-1}}^2 } \\
&\le \sqrt{\lambda^{-1}} \cdot \sqrt{t} \cdot V_{\max} \cdot \sqrt{ \sum_{i=1}^{t-1} \left\| \phi(s_{i,h}, a_{i,h}) \right\|_{\Sigma_{t-1,h}^{-1}}^2 } \\
&\le \frac{V_{\max}\sqrt{dt}}{\sqrt{\lambda}}
\end{aligned}
$$

where the second inequality is by the Jensen's inequality, the third inequality is by the condition that $\max_s |\overline{V}_{t,h}(s)| \le V_{\max}$, and the last inequality is by Lemma D.7. Putting these together, we get $\|\overline{\theta}_{\mathrm{P},t,h}\|_2 \le \epsilon_{\mathrm{P},\xi}/\sqrt{\lambda} + V_{\max}\sqrt{dt}/\sqrt{\lambda}$.

**Covering construction.** With these bounds in hand, we can now proceed with a covering construction. First define

$$
\begin{aligned}
&Q_{\theta_{\mathrm{r}}, \theta_{\mathrm{P}}, \Sigma}(s,a) \\
&:= \phi(s,a)^\top \theta_{\mathrm{r}} + \begin{cases} \phi(s,a)^\top \theta_{\mathrm{P}} & \text{if } \|\phi(s,a)\|_\Sigma \le \alpha_{\mathrm{L}} \\ \rho(s,a)\big( \phi(s,a)^\top \theta_{\mathrm{P}} \big) + (1 - \rho(s,a))(H-h) & \text{if } \alpha_{\mathrm{L}} < \|\phi(s,a)\|_\Sigma \le \alpha_{\mathrm{U}} \\ H-h & \text{if } \|\phi(s,a)\|_\Sigma > \alpha_{\mathrm{U}} \end{cases}, \\
&V_{\theta_{\mathrm{r}}, \theta_{\mathrm{P}}, \Sigma}(s) := \max_a Q_{\theta_{\mathrm{r}}, \theta_{\mathrm{P}}, \Sigma}(s,a). \quad\quad (9)
\end{aligned}
$$

where $\rho(s,a) := \frac{\alpha_{\mathrm{U}} - \|\phi(s,a)\|_\Sigma}{\alpha_{\mathrm{U}} - \alpha_{\mathrm{L}}}$. This definition aims to mimic the behavior of Algorithm 1. Then, we define the space of parameters as follows

$$
\begin{aligned}
\mathcal{U} := \Big\{ (\theta_{\mathrm{r}}, \theta_{\mathrm{P}}, \Sigma) \ : & \|\theta_{\mathrm{r}}\|_2 \le B + \epsilon_{\mathrm{r},\xi}/\sqrt{\lambda}, \ \|\theta_{\mathrm{P}}\|_2 \le (V_{\max}\sqrt{dt} + \epsilon_{\mathrm{P},\xi})/\sqrt{\lambda}, \\
& \|\Sigma\|_{\mathrm{F}} \le \sqrt{d}/\lambda, \ \Sigma = \Sigma^\top, \ \Sigma \succeq 0, \ \|V_{\theta_{\mathrm{r}}, \theta_{\mathrm{P}}, \Sigma}\|_\infty \le V_{\max} \Big\}.
\end{aligned}
$$

Clearly, $\overline{\theta}_{\mathrm{r},t}$ and $\overline{\theta}_{\mathrm{P},t,h}$ satisfy the constraints of $\theta_{\mathrm{r}}$ and $\theta_{\mathrm{P}}$. Moreover, $\|\Sigma_{t-1}^{-1}\|_{\mathrm{F}} \le \sqrt{d}\|\Sigma_{t-1}^{-1}\|_2 \le \sqrt{d}/\lambda$, and thus $\Sigma_{t-1}^{-1}$ satisfies the constraint of $\Sigma$. Hence, we have $(\overline{\theta}_{\mathrm{r},t}, \overline{\theta}_{\mathrm{P},t,h}, \Sigma_{t-1}) \in \mathcal{U}$.

By Lemma B.2, the size of an $\omega$-covering of $\mathcal{U}$ by treating it as a subset of $\mathbb{R}^{2d+d^2}$ can be bounded by

$$
N(\omega, \mathcal{U}, \|\cdot\|_2) \le \left( \frac{3\Big( (B + \epsilon_{\mathrm{r},\xi}/\sqrt{\lambda}) + (V_{\max}\sqrt{dt} + \epsilon_{\mathrm{P},\xi})/\sqrt{\lambda} + \sqrt{d}/\lambda \Big)}{\omega} \right)^{2d+d^2}
$$

We denote the covering set by $\widetilde{\mathcal{U}}$. We will write $N := N(\omega, \mathcal{U}, \|\cdot\|_2)$ for simplicity when there is no ambiguity. We note that it is possible that not all points in $\widetilde{\mathcal{U}}$ belong to $\mathcal{U}$. However, we can project all points to $\mathcal{U}$ so it becomes a $2\omega$-covering of $\mathcal{U}$. We will omit this subtlety.

**Covering argument.** For any $(\theta_{\mathrm{r}}, \theta_{\mathrm{P}}, \Sigma) \in \mathcal{U}$, we define

$$
x_{i, \theta_{\mathrm{r}}, \theta_{\mathrm{P}}, \Sigma} := V_{\theta_{\mathrm{r}}, \theta_{\mathrm{P}}, \Sigma}(s_{i,h+1}) - \mathbb{E}_{s'}\left[ V_{\theta_{\mathrm{r}}, \theta_{\mathrm{P}}, \Sigma}(s') \,|\, s_{i,h}, a_{i,h} \right].
$$

By construction, we have $|x_{i,\theta_r,\theta_P,\Sigma}| \leq 2V_{\max}$, so it is $4V_{\max}$-subgaussian conditioning on $(s_{i,h}, a_{i,h})$. Hence, for all $(\theta_r, \theta_P, \Sigma) \in \widetilde{\mathcal{U}}$, we have

$$\left\| \sum_{i=1}^{t-1} \phi(s_{i,h}, a_{i,h}) x_{i,\theta_r,\theta_P,\Sigma} \right\|_{\Sigma_{t-1,h}^{-1}} \leq \sqrt{32 V_{\max}^2 \left( d \log \left( 1 + \frac{t}{\lambda} \right) + \log(N/\delta) \right)} \qquad (10)$$

with probability at least $1 - \delta$ by Lemma D.5 and the union bound over $\widetilde{\mathcal{U}}$. Now consider an arbitrary tuple $(\theta_r, \theta_P, \Sigma) \in \mathcal{U}$ and the nearest point $(\tilde{\theta}_r, \tilde{\theta}_P, \widetilde{\Sigma}) \in \widetilde{\mathcal{U}}$. Then, we have

$$\left\| \sum_{i=1}^{t-1} \phi(s_{i,h}, a_{i,h}) x_{i,\theta_r,\theta_P,\Sigma} \right\|_{\Sigma_{t-1,h}^{-1}}$$

$$\leq \left\| \sum_{i=1}^{t-1} \phi(s_{i,h}, a_{i,h}) x_{i,\tilde{\theta}_r,\tilde{\theta}_P,\widetilde{\Sigma}} \right\|_{\Sigma_{t-1,h}^{-1}} + \left\| \sum_{i=1}^{t-1} \phi(s_{i,h}, a_{i,h}) (x_{i,\theta_r,\theta_P,\Sigma} - x_{i,\tilde{\theta}_r,\tilde{\theta}_P,\widetilde{\Sigma}}) \right\|_{\Sigma_{t-1,h}^{-1}}$$

$$\leq \left\| \sum_{i=1}^{t-1} \phi(s_{i,h}, a_{i,h}) x_{i,\tilde{\theta}_r,\tilde{\theta}_P,\widetilde{\Sigma}} \right\|_{\Sigma_{t-1,h}^{-1}} + \sum_{i=1}^{t-1} \| \phi(s_{i,h}, a_{i,h}) \|_{\Sigma_{t-1,h}^{-1}} \cdot \max_i |x_{i,\theta_r,\theta_P,\Sigma} - x_{i,\tilde{\theta}_r,\tilde{\theta}_P,\widetilde{\Sigma}}|$$

$$\leq \left\| \sum_{i=1}^{t-1} \phi(s_{i,h}, a_{i,h}) x_{i,\tilde{\theta}_r,\tilde{\theta}_P,\widetilde{\Sigma}} \right\|_{\Sigma_{t-1,h}^{-1}} + \sqrt{\lambda^{-1}} \cdot t \cdot \max_i |x_{i,\theta_r,\theta_P,\Sigma} - x_{i,\tilde{\theta}_r,\tilde{\theta}_P,\widetilde{\Sigma}}|$$

where the last step is by the fact that $\| \phi(s_{i,h}, a_{i,h}) \|_{\Sigma_{t-1,h}^{-1}} \leq \sqrt{\lambda^{-1}}$. Observing the derived upper bound above, the first term is already bounded by (10) since $(\tilde{\theta}_r, \tilde{\theta}_P, \widetilde{\Sigma}) \in \widetilde{\mathcal{U}}$. To bound the second term, we note that, by the definiton of $x_{i,\theta_r,\theta_P,\Sigma}$ and $x_{i,\tilde{\theta}_r,\tilde{\theta}_P,\widetilde{\Sigma}}$,

$$|x_{i,\theta_r,\theta_P,\Sigma} - x_{i,\tilde{\theta}_r,\tilde{\theta}_P,\widetilde{\Sigma}}| \leq 2 \max_s \left| V_{\theta_r,\theta_P,\Sigma}(s) - V_{\tilde{\theta}_r,\tilde{\theta}_P,\widetilde{\Sigma}}(s) \right|$$

$$= 2 \max_s \left| \max_a Q_{\theta_r,\theta_P,\Sigma}(s,a) - \max_a Q_{\tilde{\theta}_r,\tilde{\theta}_P,\widetilde{\Sigma}}(s,a) \right|$$

$$\leq 2 \max_{s,a} \left| Q_{\theta_r,\theta_P,\Sigma}(s,a) - Q_{\tilde{\theta}_r,\tilde{\theta}_P,\widetilde{\Sigma}}(s,a) \right|$$

We assume $\omega \leq (\alpha_U - \alpha_L)^2$ (and this will be satisfied later when we specify the value of $\omega$). Recall that we have three cases in the definition of $Q_{\theta_r,\theta_P,\Sigma}(s,a)$ (see (9)), and we will refer to them as Case L, Case M, and Case U. There are in total 6 possible combinations of cases for the pair $Q_{\theta_r,\theta_P,\Sigma}(s,a), Q_{\tilde{\theta}_r,\tilde{\theta}_P,\widetilde{\Sigma}}(s,a)$, and we will discuss them one by one below. For the ease of notation, we denote $Q(s,a)$ and $\widetilde{Q}(s,a)$ as shorthand for $Q_{\theta_r,\theta_P,\Sigma}(s,a)$ and $Q_{\tilde{\theta}_r,\tilde{\theta}_P,\widetilde{\Sigma}}(s,a)$.

**(1) Case L + Case U.** This is impossible, because, in this case, we have $|\|\phi\|_\Sigma - \|\phi\|_{\widetilde{\Sigma}}| > \alpha_U - \alpha_L$, but we also have $|\|\phi\|_\Sigma - \|\phi\|_{\widetilde{\Sigma}}| = |\sqrt{\phi^\top \Sigma \phi} - \sqrt{\phi^\top \widetilde{\Sigma} \phi}| \leq \sqrt{|\phi^\top (\Sigma - \widetilde{\Sigma}) \phi|} \leq \sqrt{\|\phi\|_2 \|\Sigma - \widetilde{\Sigma}\|_F \|\phi\|_2} \leq \sqrt{\omega} < \alpha_U - \alpha_L$.

**(2) Both are Case L.** Then we immediately have

$$|Q(s,a) - \widetilde{Q}(s,a)| = \left| \phi(s,a)^\top (\theta_r + \theta_P - \tilde{\theta}_r - \tilde{\theta}_P) \right| \leq \|\phi(s,a)\|_2 \left( \|\theta_r - \tilde{\theta}_r\|_2 + \|\theta_P - \tilde{\theta}_P\|_2 \right) \leq 2\omega.$$

**(3) Both are Case U.** Then we immediately have

$$|Q(s,a) - \widetilde{Q}(s,a)| = \left| \phi(s,a)^\top (\theta_r - \tilde{\theta}_r) \right| \leq |\phi(s,a)\|_2 \cdot \|\theta_r - \tilde{\theta}_r\|_2 \leq \omega.$$

**(4) Case L + Case M.** Without loss of generality, we assume $Q(s,a)$ is in Case M and $\widetilde{Q}(s,a)$ is in Case L. Then we have

$$|Q(s,a) - \widetilde{Q}(s,a)| \leq \rho(s,a) \left|\phi(s,a)^\top(\theta_\mathrm{r} + \theta_\mathrm{P} - \tilde{\theta}_\mathrm{r} - \tilde{\theta}_\mathrm{P})\right|$$

$$+ (1 - \rho(s,a)) \left|\phi(s,a)^\top(\tilde{\theta}_\mathrm{r} + \tilde{\theta}_\mathrm{P}) - (H - h)\right|$$

$$\leq \rho(s,a) \cdot 2\omega + (1 - \rho(s,a)) \cdot 2V_{\max}$$

Recall that $\rho(s,a) := \frac{\alpha_\mathrm{U} - \|\phi(s,a)\|_\Sigma}{\alpha_\mathrm{U} - \alpha_\mathrm{L}}$, so $1 - \rho(s,a) := \frac{\|\phi(s,a)\|_\Sigma - \alpha_\mathrm{L}}{\alpha_\mathrm{U} - \alpha_\mathrm{L}}$. Futhermore,

$$1 - \rho(s,a) = \frac{\|\phi(s,a)\|_\Sigma - \alpha_\mathrm{L}}{\alpha_\mathrm{U} - \alpha_\mathrm{L}} = \frac{\|\phi(s,a)\|_\Sigma - \|\phi(s,a)\|_{\widetilde{\Sigma}} + \|\phi(s,a)\|_{\widetilde{\Sigma}} - \alpha_\mathrm{L}}{\alpha_\mathrm{U} - \alpha_\mathrm{L}}$$

$$\leq \frac{\|\phi(s,a)\|_\Sigma - \|\phi(s,a)\|_{\widetilde{\Sigma}}}{\alpha_\mathrm{U} - \alpha_\mathrm{L}} = \frac{\sqrt{\phi^\top \Sigma \phi} - \sqrt{\phi^\top \widetilde{\Sigma} \phi}}{\alpha_\mathrm{U} - \alpha_\mathrm{L}}$$

$$\leq \frac{\sqrt{|\phi^\top(\Sigma - \widetilde{\Sigma})\phi|}}{\alpha_\mathrm{U} - \alpha_\mathrm{L}} \leq \frac{\sqrt{\|\phi\|_2 \|\Sigma - \widetilde{\Sigma}\|_2 \|\phi\|_2}}{\alpha_\mathrm{U} - \alpha_\mathrm{L}} \leq \frac{\sqrt{\omega}}{\alpha_\mathrm{U} - \alpha_\mathrm{L}}$$

where the first inequality is by the condition that $\widetilde{Q}$ is in Case L. Inserting this back, we get $|Q(s,a) - \widetilde{Q}(s,a)| \leq 2\omega + \frac{2V_{\max}\sqrt{\omega}}{\alpha_\mathrm{U} - \alpha_\mathrm{L}}$.

**(5) Case M + Case U.** Without loss of generality, we assume $Q(s,a)$ is in Case M and $\widetilde{Q}(s,a)$ is in Case U. Then we have

$$|Q(s,a) - \widetilde{Q}(s,a)| \leq \rho(s,a) \left|\phi(s,a)^\top(\theta_\mathrm{r} + \theta_\mathrm{P} - (H - h))\right| + (1 - \rho(s,a)) |(H - h) - (H - h)|$$

$$= \rho(s,a) \left|\phi(s,a)^\top(\theta_\mathrm{r} + \theta_\mathrm{P} - (H - h))\right| \leq \rho(s,a) \cdot 2V_{\max} \leq \frac{2V_{\max}\sqrt{\omega}}{\alpha_\mathrm{U} - \alpha_\mathrm{L}}$$

where the last inequality is from

$$\rho(s,a) = \frac{\alpha_\mathrm{U} - \|\phi(s,a)\|_\Sigma}{\alpha_\mathrm{U} - \alpha_\mathrm{L}} = \frac{\alpha_\mathrm{U} - \|\phi(s,a)\|_\Sigma - \|\phi(s,a)\|_{\widetilde{\Sigma}} + \|\phi(s,a)\|_{\widetilde{\Sigma}}}{\alpha_\mathrm{U} - \alpha_\mathrm{L}}$$

$$\leq \frac{\|\phi(s,a)\|_{\widetilde{\Sigma}} - \|\phi(s,a)\|_\Sigma}{\alpha_\mathrm{U} - \alpha_\mathrm{L}} = \frac{\sqrt{\phi^\top \widetilde{\Sigma} \phi} - \sqrt{\phi^\top \Sigma \phi}}{\alpha_\mathrm{U} - \alpha_\mathrm{L}}$$

$$\leq \frac{\sqrt{|\phi^\top(\Sigma - \widetilde{\Sigma})\phi|}}{\alpha_\mathrm{U} - \alpha_\mathrm{L}} \leq \frac{\sqrt{\|\phi\|_2 \|\Sigma - \widetilde{\Sigma}\|_2 \|\phi\|_2}}{\alpha_\mathrm{U} - \alpha_\mathrm{L}} \leq \frac{\sqrt{\omega}}{\alpha_\mathrm{U} - \alpha_\mathrm{L}}.$$

where the first inequality is by the condition that $\widetilde{Q}$ is in Case U.

**(6) Both are Case M.** Denote $\rho(s,a) = \frac{\alpha_\mathrm{U} - \|\phi(s,a)\|_\Sigma}{\alpha_\mathrm{U} - \alpha_\mathrm{L}}$ and $\widetilde{\rho}(s,a) = \frac{\alpha_\mathrm{U} - \|\phi(s,a)\|_{\widetilde{\Sigma}}}{\alpha_\mathrm{U} - \alpha_\mathrm{L}}$. Then, we have

$$|\rho(s,a) - \widetilde{\rho}(s,a)| = \frac{\left|\|\phi(s,a)\|_{\widetilde{\Sigma}} - \|\phi(s,a)\|_\Sigma\right|}{\alpha_\mathrm{U} - \alpha_\mathrm{L}} \leq \frac{\sqrt{\|\phi\|_2 \|\widetilde{\Sigma} - \Sigma\|_\mathrm{F} \|\phi\|_2}}{\alpha_\mathrm{U} - \alpha_\mathrm{L}} \leq \frac{\sqrt{\omega}}{\alpha_\mathrm{U} - \alpha_\mathrm{L}},$$

which also implies $|(1 - \rho(s,a)) - (1 - \widetilde{\rho}(s,a))| \leq \frac{\sqrt{\omega}}{\alpha_\mathrm{U} - \alpha_\mathrm{L}}$. Hence, we have

$$|Q(s,a) - \widetilde{Q}(s,a)| \leq \left|\rho(s,a)\phi(s,a)^\top(\theta_\mathrm{r} + \theta_\mathrm{P}) - \widetilde{\rho}(s,a)\phi(s,a)^\top(\tilde{\theta}_\mathrm{r} + \tilde{\theta}_\mathrm{P})\right|$$

$$+ |(1 - \rho(s,a))(H - h) - (1 - \widetilde{\rho}(s,a))(H - h)|$$

$$\leq \left|\rho(s,a)\phi(s,a)^\top(\theta_\mathrm{r} + \theta_\mathrm{P}) - \rho(s,a)\phi(s,a)^\top(\tilde{\theta}_\mathrm{r} + \tilde{\theta}_\mathrm{P})\right|$$

$$+ \left|\rho(s,a)\phi(s,a)^\top(\tilde{\theta}_\mathrm{r} + \tilde{\theta}_\mathrm{P}) - \widetilde{\rho}(s,a)\phi(s,a)^\top(\tilde{\theta}_\mathrm{r} + \tilde{\theta}_\mathrm{P})\right|$$

$$+ |(1 - \rho(s,a))(H - h) - (1 - \widetilde{\rho}(s,a))(H - h)|$$

$$\leq 2\omega + \frac{V_{\max}\sqrt{\omega}}{\alpha_\mathrm{U} - \alpha_\mathrm{L}} + \frac{H\sqrt{\omega}}{\alpha_\mathrm{U} - \alpha_\mathrm{L}}$$

**Putting Everything Together.** Taking the maximum of the six cases, we conclude that

$$|Q(s,a) - \widetilde{Q}(s,a)| \leq 2\omega + \frac{2V_{\max}\sqrt{\omega}}{\alpha_{\mathrm{U}} - \alpha_{\mathrm{L}}}$$

We set $\omega = \left(\frac{\alpha_{\mathrm{U}} - \alpha_{\mathrm{L}}}{2tV_{\max}}\right)^2$ and obtain

$$|Q(s,a) - \widetilde{Q}(s,a)| \leq 2\left(\frac{\alpha_{\mathrm{U}} - \alpha_{\mathrm{L}}}{2tV_{\max}}\right)^2 + \frac{1}{t} \leq \frac{3}{t}$$

where we leverage the condition that $\alpha_{\mathrm{L}}, \alpha_{\mathrm{U}} \leq 1$. Thus, we have

$$|x_{i,\theta_{\mathrm{r}},\theta_{\mathrm{P}},\Sigma} - x_{i,\tilde{\theta}_{\mathrm{r}},\tilde{\theta}_{\mathrm{P}},\widetilde{\Sigma}}| \leq 2\max_{s,a}\left|Q(s,a) - \widetilde{Q}(s,a)\right| \leq \frac{6}{t}.$$

**Final Bound.** The covering argument is now complete, and we can use it to derive an upper bound for $\|\eta_{\mathrm{P},t,h}\|_{\Sigma_{t-1,h}}$. To that end, we note that, for $\eta_{\mathrm{P},t,h}$, there must exists $(\theta_{\mathrm{r}}, \theta_{\mathrm{P}}, \Sigma) \in \mathcal{U}$ and its closest element $(\tilde{\theta}_{\mathrm{r}}, \widetilde{\theta}_{\mathrm{P}}, \widetilde{\Sigma}) \in \widetilde{\mathcal{U}}$ such that

$$\|\eta_{\mathrm{P},t,h}\|_{\Sigma_{t-1,h}}$$
$$= \left\|\sum_{i=1}^{t-1} \phi(s_{i,h}, a_{i,h})\left(\overline{V}_{t,h+1}(s_{i,h+1}) - \mathbb{E}_{s_{i,h+1}}\left[\overline{V}_{t,h+1}(s_{i,h+1}) \mid s_{i,h}, a_{i,h}\right]\right)\right\|_{\Sigma_{t-1,h}^{-1}}$$
$$= \left\|\sum_{i=1}^{t-1} \phi(s_{i,h}, a_{i,h})x_{i,\theta_{\mathrm{r}},\theta_{\mathrm{P}},\Sigma}\right\|_{\Sigma_{t-1,h}^{-1}}$$
$$= \left\|\sum_{i=1}^{t-1} \phi(s_{i,h}, a_{i,h})x_{i,\tilde{\theta}_{\mathrm{r}},\tilde{\theta}_{\mathrm{P}},\Sigma}\right\|_{\Sigma_{t-1,h}^{-1}} + \left\|\sum_{i=1}^{t-1} \phi(s_{i,h}, a_{i,h})\left(x_{i,\theta_{\mathrm{r}},\theta_{\mathrm{P}},\Sigma} - x_{i,\tilde{\theta}_{\mathrm{r}},\tilde{\theta}_{\mathrm{P}},\Sigma}\right)\right\|_{\Sigma_{t-1,h}^{-1}}$$

where the second term is bounded by

$$\left\|\sum_{i=1}^{t-1} \phi(s_{i,h}, a_{i,h})\left(x_{i,\theta_{\mathrm{r}},\theta_{\mathrm{P}},\Sigma} - x_{i,\tilde{\theta}_{\mathrm{r}},\tilde{\theta}_{\mathrm{P}},\Sigma}\right)\right\|_{\Sigma_{t-1,h}^{-1}}$$
$$\leq \sum_{i=1}^{t-1} \left\|\phi(s_{i,h}, a_{i,h})\left(x_{i,\theta_{\mathrm{r}},\theta_{\mathrm{P}},\Sigma} - x_{i,\tilde{\theta}_{\mathrm{r}},\tilde{\theta}_{\mathrm{P}},\Sigma}\right)\right\|_{\Sigma_{t-1,h}^{-1}}$$
$$\leq \sqrt{\lambda^{-1}} \cdot t \cdot \max_i |x_{i,\theta_{\mathrm{r}},\theta_{\mathrm{P}},\Sigma} - x_{i,\tilde{\theta}_{\mathrm{r}},\tilde{\theta}_{\mathrm{P}},\Sigma}|$$
$$\leq 6/\sqrt{\lambda},$$

and, applying (10), the first term is bounded by

$$\left\|\sum_{i=1}^{t-1} \phi(s_{i,h}, a_{i,h})x_{i,\tilde{\theta}_{\mathrm{r}},\tilde{\theta}_{\mathrm{P}},\Sigma}\right\|_{\Sigma_{t-1,h}^{-1}}$$
$$\leq \sqrt{32V_{\max}^2\left(d\log\left(1 + \frac{t}{\lambda}\right) + \log(N/\delta)\right)}$$
$$\leq \sqrt{32V_{\max}^2\left(d\log\left(1 + \frac{t}{\lambda}\right) + (2d + d^2)\log\left(\frac{3\left((B + \epsilon_{\mathrm{r},\xi}/\sqrt{\lambda}) + (V_{\max}\sqrt{d}t + \epsilon_{\mathrm{P},\xi})/\sqrt{\lambda} + \sqrt{d}/\lambda\right)}{\omega\delta}\right)\right)}$$
$$= V_{\max}\sqrt{32d\log\left(1 + \frac{t}{\lambda}\right) + 32(2d + d^2)\log\left(\frac{12V_{\max}^2t^2\left((B + \epsilon_{\mathrm{r},\xi}/\sqrt{\lambda}) + (V_{\max}\sqrt{d}t + \epsilon_{\mathrm{P},\xi})/\sqrt{\lambda} + \sqrt{d}/\lambda\right)}{(\alpha_{\mathrm{U}} - \alpha_{\mathrm{L}})^2\delta}\right)}$$

where the second inequality is by inserting the upper bound of the covering number $N$, and the equality is by inserting the value of $\omega$. The last line looks complicated and we can further simplify it and get

$$\left\|\sum_{i=1}^{t-1} \phi(s_{i,h}, a_{i,h}) x_{i,\tilde{\theta}_{\mathrm{r}}, \tilde{\theta}_{\mathrm{P}}, \Sigma}\right\|_{\Sigma_{t-1,h}^{-1}}$$

$$\leq 16 dV_{\max} \sqrt{\log\left(\frac{12V_{\max}t(t+\lambda)\left((B+\epsilon_{\mathrm{r},\xi}/\sqrt{\lambda}) + (V_{\max}\sqrt{dt} + \epsilon_{\mathrm{P},\xi})/\sqrt{\lambda} + \sqrt{d}/\lambda\right)}{(\alpha_{\mathrm{U}} - \alpha_{\mathrm{L}})\delta\lambda}\right)}.$$

Plugging these upper bounds back, we obtain

$$\|\eta_{\mathrm{P},t,h}\|_{\Sigma_{t-1,h}}$$

$$\leq \frac{6}{\sqrt{\lambda}} + 16 dV_{\max} \sqrt{\log\left(\frac{12V_{\max}t(t+\lambda)\left((B+\epsilon_{\mathrm{r},\xi}/\sqrt{\lambda}) + (V_{\max}\sqrt{dt} + \epsilon_{\mathrm{P},\xi})/\sqrt{\lambda} + \sqrt{d}/\lambda\right)}{(\alpha_{\mathrm{U}} - \alpha_{\mathrm{L}})\delta\lambda}\right)}.$$

Applying union bound over all $t \in [T]$ and $h \in [H]$, the upper bound exactly becomes $\epsilon'_{\mathrm{P},\eta}$. Further invoking Lemma B.3 finishes the proof. $\square$

**Lemma B.11** (Boundness of value functions and transition estimation Error). *Assume the events define in Lemmas B.7 and B.9 hold. Then, the following holds with probability at least $1 - \delta$:*

1. $\max_{s,a} |\overline{Q}_{t,h}(s,a)| \leq V_{\max}$ *for all $t \in [T]$ and $h \in [H]$.*

2. $\|\eta_{\mathrm{P},t,h}\|_{\Sigma_{t-1,h}} \leq \epsilon_{\mathrm{P},\eta}$ *for all $t \in [T]$ and $h \in [H]$.*

3. $\|\lambda_{t,h}\|_{\Sigma_{t-1,h}} \leq \epsilon_\lambda$ *for all $t \in [T]$ and $h \in [H]$.*

*The first statement implies $\max_s |\overline{V}_{t,h}(s)| \leq V_{\max}$ for all $t \in [T]$ and $h \in [H]$.*

*Proof.* We prove the three statements together by induction.

For the first statement, we define $V_{\max,h} := (H - h + 1)(1 + (\epsilon_{\mathrm{r},\xi} + \epsilon_{\mathrm{r},\eta})/\sqrt{\lambda})$, and we will actually show that $\max_{s,a} |\overline{Q}_{t,h}(s,a) - Q_h^\star(s,a)| \leq V_{\max,h}$ in the induction, which immediately leads to $\max_{s,a} |\overline{Q}_{t,h}(s,a)| \leq V_{\max,h} + (H - h + 1) \leq V_{\max}$.

**Base.** For $h = H$, we can directly apply Lemma B.10 without the transition argument, which leads to the desired upper bound of $\|\eta_{\mathrm{P},t,H}\|_{\Sigma_{t-1,h}}$. Moreover, the upper bound of $\|\lambda_{t,h}\|_{\Sigma_{t-1,h}}$ also trivially holds. For an upper bound on the value function, we immediately have

$$\left|\overline{Q}_{t,H}(s,a) - Q_H^\star(s,a)\right| = \left|\phi(s,a)^\top(\overline{\theta}_{\mathrm{r},t} - \theta_{\mathrm{r}}^\star)\right| \leq \|\phi(s,a)\|_{\Sigma_{t-1}^{-1}}\|\overline{\theta}_{\mathrm{r},t} - \theta_{\mathrm{r}}^\star\|_{\Sigma_{t-1}}$$

by Cauchy-Schwarz inequality. We note that $\|\phi(s,a)\|_{\Sigma_{t-1}^{-1}} \leq \sqrt{\lambda^{-1}}$ and $\|\overline{\theta}_{\mathrm{r},t} - \theta_{\mathrm{r}}^\star\|_{\Sigma_{t-1}} \leq \|\xi_{\mathrm{r},t} + \eta_{\mathrm{r},t}\|_{\Sigma_{t-1}} \leq \|\xi_{\mathrm{r},t}\|_{\Sigma_{t-1}} + \|\eta_{\mathrm{r},t}\|_{\Sigma_{t-1}} \leq \epsilon_{\mathrm{r},\xi} + \epsilon_{\mathrm{r},\eta}$. Hence, we have

$$\left|\overline{Q}_{t,H}(s,a) - Q_H^\star(s,a)\right| \leq (\epsilon_{\mathrm{r},\xi} + \epsilon_{\mathrm{r},\eta})/\sqrt{\lambda} \leq V_{\max,H}.$$

**Inductive Steps.** Now consider $h < H$. By induction hypothesis, we have $\max_{t,s} |\overline{V}_{t,h+1}(s)| \leq V_{\max}$. Thus, we can apply Lemma B.10 and get $\|\eta_{\mathrm{P},t,h}\|_{\Sigma_{t-1,h}} \leq \epsilon_{\mathrm{P},\eta}$. For the upper bound of $\|\lambda_{t,h}\|_{\Sigma_{t-1,h}}$, by definition, we have

$$\|\lambda_{t,h}\|_{\Sigma_{t-1,h}} = \left\|\lambda \int_{s'} \mu_h^\star(s') \overline{V}_{t,h+1}(s') \, ds'\right\|_{\Sigma_{t-1,h}^{-1}}$$

$$\leq \sqrt{\lambda} \left\|\int_{s'} \mu_h^\star(s') \overline{V}_{t,h+1}(s') \, ds'\right\|_2$$

$$\leq V_{\max}\sqrt{\lambda d} = \epsilon_\lambda$$

where the second inequality is by $\|\Sigma_{t-1,h}^{-1/2}\|_2 \le \sqrt{\lambda}$, and the last inequality is by Assumption 4.1.

For the upper bound on the value function, consider the following three cases.

**Case 1:** $\|\phi(s,a)\|_{\Sigma_{t-1,h}^{-1}} \le \alpha_{\mathrm{L}}.$   We apply Lemma B.5 and get

$$
\begin{aligned}
&|\phi(s,a)^\top \left(\overline{\theta}_{\mathrm{r},t} + \overline{\theta}_{\mathrm{P},t,h}\right) - Q_h^\star(s,a)| \\
&= \left|\phi(s,a)^\top (\xi_{\mathrm{r},t} + \xi_{\mathrm{P},t,h} + \eta_{\mathrm{r},t} + \eta_{\mathrm{P},t,h} - \lambda_{t,h}) + \underset{s'}{\mathbb{E}}\left[\overline{V}_{t,h+1}(s') - V_{h+1}^\star(s') \mid s,a\right]\right| \\
&\le \left|\phi(s,a)^\top (\xi_{\mathrm{r},t} + \eta_{\mathrm{r},t})\right| + \left|\phi(s,a)^\top (\xi_{\mathrm{P},t,h} + \eta_{\mathrm{P},t,h} - \lambda_{t,h})\right| + \left|\underset{s'}{\mathbb{E}}\left[\overline{V}_{t,h+1}(s') - V_{h+1}^\star(s') \mid s,a\right]\right| \\
&\le (\epsilon_{\mathrm{r},\xi} + \epsilon_{\mathrm{r},\eta})/\sqrt{\lambda} + \|\phi(s,a)\|_{\Sigma_{t-1,h}^{-1}}(\epsilon_{\mathrm{P},\xi} + \epsilon_{\mathrm{P},\eta} + \epsilon_\lambda) + \left((H-h)(1 + (\epsilon_{\mathrm{r},\xi} + \epsilon_{\mathrm{r},\eta})/\sqrt{\lambda})\right) \\
&\le (\epsilon_{\mathrm{r},\xi} + \epsilon_{\mathrm{r},\eta})/\sqrt{\lambda} + 1 + \left((H-h)(1 + (\epsilon_{\mathrm{r},\xi} + \epsilon_{\mathrm{r},\eta})/\sqrt{\lambda})\right) \\
&= (H-h+1)(1 + (\epsilon_{\mathrm{r},\xi} + \epsilon_{\mathrm{r},\eta})/\sqrt{\lambda}) \\
&= V_{\max,h}
\end{aligned}
$$

where the second inequality is by Cauchy-Schwarz inequality, Lemmas B.7 and B.9, and the induction hypothesis. The third inequality is by the condition that $\|\phi(s,a)\|_{\Sigma_{t-1,h}^{-1}} \le \alpha_{\mathrm{L}}$ and the definition of $\alpha_{\mathrm{L}}$.

**Case 2:** $\|\phi(s,a)\|_{\Sigma_{t-1,h}^{-1}} > \alpha_{\mathrm{U}}.$   We have

$$
\begin{aligned}
\left|\overline{Q}_{t,h}(s,a) - Q_h^\star(s,a)\right| &\le |\phi(s,a)^\top \overline{\theta}_{\mathrm{r},t} + H - h - Q_h^\star(s,a)| \\
&= |\phi(s,a)^\top (\xi_{\mathrm{r},t} + \eta_{\mathrm{r},t}) + H - h + \phi(s,a)^\top \theta_{\mathrm{r}}^\star - Q_h^\star(s,a)| \\
&\le (\epsilon_{\mathrm{r},\xi} + \epsilon_{\mathrm{r},\eta})/\sqrt{\lambda} + (H-h+1) \\
&\le V_{\max,h}
\end{aligned}
$$

where we used the fact that $0 \le \phi(s,a)^\top \theta_{\mathrm{r}}^\star \le 1$ and $0 \le Q_h^\star(s,a) \le H - h + 1$, which leads to $-(H-h+1) \le \phi(s,a)^\top \theta_{\mathrm{r}}^\star - Q_h^\star(s,a) \le 1$.

**Case 3:** $\alpha_{\mathrm{L}} < \|\phi(s,a)\|_{\Sigma_{t-1,h}^{-1}} \le \alpha_{\mathrm{U}}.$   Denoting $\rho := \rho(s,a)$ for simplicity, we have

$$
\begin{aligned}
&\left|\overline{Q}_{t,h}(s,a) - Q_h^\star(s,a)\right| \\
&\le \rho \left|\phi(s,a)^\top \left(\overline{\theta}_{\mathrm{r},t} + \overline{\theta}_{\mathrm{P},t,h}\right) - Q_h^\star(s,a)\right| + (1-\rho)\left|\phi(s,a)^\top \overline{\theta}_{\mathrm{r},t} + H - h - Q_h^\star(s,a)\right| \\
&\le \rho(H-h+1)\left(1 + (\epsilon_{\mathrm{r},\xi} + \epsilon_{\mathrm{r},\eta})/\sqrt{\lambda}\right) + (1-\rho)\left|\phi(s,a)^\top (\xi_{\mathrm{r},t} + \eta_{\mathrm{r},t}) + H - h + \phi(s,a)^\top \theta_{\mathrm{r}}^\star - Q_h^\star(s,a)\right| \\
&\le \rho(H-h+1)\left(1 + (\epsilon_{\mathrm{r},\xi} + \epsilon_{\mathrm{r},\eta})/\sqrt{\lambda}\right) + (1-\rho)\left((\epsilon_{\mathrm{r},\xi} + \epsilon_{\mathrm{r},\eta})/\sqrt{\lambda} + (H-h+1)\right) \\
&\le V_{\max,h}
\end{aligned}
$$

where we have used the similar arguments from Case 1 and 2.

Taking the maximum over the three cases, we conclude that

$$
|\overline{Q}_{t,h}(s,a) - Q_h^\star(s,a)| \le V_{\max,h}
$$

which implies $\max_{s,a} |\overline{Q}_{t,h}(s,a)| \le V_{\max}$. $\qquad\square$

**Lemma B.12** (High probability bounds – summary). *All of the following events hold simultaneously with probability at least $1 - \delta$:*

1.  $\|\xi_{\mathrm{r},t}\|_{\Sigma_{t-1}} \le \epsilon_{\mathrm{r},\xi}$ *for all $t \in [T]$*

2.  $\|\xi_{\mathrm{P},t,h}\|_{\Sigma_{t-1,h}} \le \epsilon_{\mathrm{P},\xi}$ *for all $t \in [T]$ and $h \in [H]$.*

3. $\|\eta_{\mathrm{r},t}\|_{\Sigma_{t-1}} \leq \epsilon_{\mathrm{r},\eta}$ *for all $t \in [T]$.*

4. $\|\eta_{\mathrm{P},t,h}\|_{\Sigma_{t-1,h}} \leq \epsilon_{\mathrm{P},\eta}$ *for all $t \in [T]$ and $h \in [H]$.*

5. $\|\lambda_{t,h}\|_{\Sigma_{t-1,h}} \leq \epsilon_{\lambda}$ *for all $t \in [T]$ and $h \in [H]$.*

6. $|\overline{Q}_{t,h}(s,a)| \leq V_{\max}$ *and $|\overline{V}_{t,h}(s)| \leq V_{\max}$ for all $(s,a) \in \mathcal{S} \times \mathcal{A}$, $t \in [T]$, and $h \in [H]$.*

*Proof of Lemma B.12.* The first and second statements are by Lemma B.7. The third is by Lemma B.9. The last three are by Lemma B.11. Then we take a union bound over all of them. $\square$

## B.4 BOUNDING REGRET

We define

$$\widetilde{V}_t = \mathop{\mathbb{E}}_{\tau \sim \pi_t^1} \left[ \sum_{h=1}^{H} \phi(s_h, a_h)^\top \overline{\theta}_{\mathrm{r},t} \right].$$

**Lemma B.13.** *Assume all events listed in Lemma B.12 hold. Fix $t \in [T]$. Then, we have $(V^\star - \overline{V}_t) + (\widetilde{V} - V^{\pi_t^1}) \leq 0$ with probability at least $\mathrm{F}^2(-1)$ where $\mathrm{F}$ denotes the CDF of a standard normal distribution (i.e., $\mathcal{N}(0,1)$).*

*Proof.* Define the indicators

$$\boldsymbol{L}_{t,h}^\star(s) := \mathbb{1}\left\{ \|\phi(s, \pi^\star(s))\|_{\Sigma_{t-1,h}^{-1}} \leq \alpha_{\mathrm{L}} \right\},$$

$$\boldsymbol{M}_{t,h}^\star(s) := \mathbb{1}\left\{ \alpha_{\mathrm{L}} < \|\phi(s, \pi^\star(s))\|_{\Sigma_{t-1,h}^{-1}} \leq \alpha_{\mathrm{U}} \right\},$$

$$\boldsymbol{U}_{t,h}^\star(s) := \mathbb{1}\left\{ \|\phi(s, \pi^\star(s))\|_{\Sigma_{t-1,h}^{-1}} > \alpha_{\mathrm{U}} \right\}.$$

We note that they are independent of the random noises at episode $t$ and only depends on the information up to episode $t - 1$. Then, we can decompose $V^\star - \overline{V}_t$:

$$
\begin{aligned}
V^\star - \overline{V}_t &= \mathop{\mathbb{E}}_{s_1}\left[ V_1^\star(s_1) - \overline{V}_{t,1}(s_1) \right] = \mathop{\mathbb{E}}_{s_1}\left[ Q_1^\star(s_1, \pi^\star(s_1)) - \overline{Q}_{t,1}(s_1, \pi_{t,1}^0(s_1)) \right] \\
&\leq \mathop{\mathbb{E}}_{s_1}\left[ Q_1^\star(s_1, \pi^\star(s_1)) - \overline{Q}_{t,1}(s_1, \pi^\star(s_1)) \right] \\
&= \mathop{\mathbb{E}}_{s_1}\left[ \boldsymbol{L}_{t,1}^\star(s_1) \left( Q_1^\star(s_1, \pi^\star(s_1)) - \overline{Q}_{t,1}(s_1, \pi^\star(s_1)) \right) \right] \\
&\quad + \mathop{\mathbb{E}}_{s_1}\left[ \boldsymbol{M}_{t,1}^\star(s_1) \left( Q_1^\star(s_1, \pi^\star(s_1)) - \overline{Q}_{t,1}(s_1, \pi^\star(s_1)) \right) \right] \\
&\quad + \mathop{\mathbb{E}}_{s_1}\left[ \boldsymbol{U}_{t,1}^\star(s_1) \left( Q_1^\star(s_1, \pi^\star(s_1)) - \overline{Q}_{t,1}(s_1, \pi^\star(s_1)) \right) \right] \\
&=: \mathrm{T_L} + \mathrm{T_M}, + \mathrm{T_U}.
\end{aligned}
$$

Now we establish upper bounds for each term. For $\mathrm{T_U}$, we have

$$
\begin{aligned}
& \mathop{\mathbb{E}}_{s_1}\left[ \boldsymbol{U}_{t,1}^\star(s_1) \left( Q_1^\star(s_1, \pi^\star(s_1)) - \overline{Q}_{t,1}(s_1, \pi^\star(s_1)) \right) \right] \\
&= \mathop{\mathbb{E}}_{s_1}\left[ \boldsymbol{U}_{t,1}^\star(s_1) \left( Q_1^\star(s_1, \pi^\star(s_1)) - \phi(s_1, \pi^\star(s_1))^\top \overline{\theta}_{\mathrm{r},t} - (H-1) \right) \right] \\
&= \mathop{\mathbb{E}}_{s_1}\left[ \boldsymbol{U}_{t,1}^\star(s_1) \left( Q_1^\star(s_1, \pi^\star(s_1)) - \phi(s_1, \pi^\star(s_1))^\top \theta_{\mathrm{r}}^\star - \phi(s_1, \pi^\star(s_1))^\top (\eta_{\mathrm{r},t} + \xi_{\mathrm{r},t}) - (H-1) \right) \right] \\
&\leq \mathop{\mathbb{E}}_{s_1}\left[ \left( -\boldsymbol{U}_{t,1}^\star(s_1) \phi(s_1, \pi^\star(s_1)) \right)^\top (\eta_{\mathrm{r},t} + \xi_{\mathrm{r},t}) \right]
\end{aligned}
$$

where the inequality holds since

$$
\begin{aligned}
& Q_1^\star(s_1, \pi^\star(s_1)) - \phi(s_1, \pi^\star(s_1))^\top \theta_{\mathrm{r}}^\star - (H-1) \\
&= \phi(s_1, \pi^\star(s_1))^\top \theta_{\mathrm{r}}^\star + \mathop{\mathbb{E}}_{s_2}[V_2^\star(s_2) \mid s_1, \pi^\star(s_1)] - \phi(s_1, \pi^\star(s_1))^\top \theta_{\mathrm{r}}^\star - (H-1) \leq 0.
\end{aligned}
$$

For $T_L$, we have

$$
\mathbb{E}_{s_1}\left[\boldsymbol{L}^\star_{t,1}(s_1)\left(Q^\star_1(s_1,\pi^\star(s_1)) - \overline{Q}_{t,1}(s_1,\pi^\star(s_1))\right)\right]
$$
$$
= \mathbb{E}_{s_1}\left[\boldsymbol{L}^\star_{t,1}(s_1)\left(Q^\star_1(s_1,\pi^\star(s_1)) - \phi(s_1,\pi^\star(s_1))^\top(\overline{\theta}_{r,t} + \overline{\theta}_{P,t,h})\right)\right]
$$
$$
= \mathbb{E}_{s_1}\left[\boldsymbol{L}^\star_{t,1}(s_1)\left(-\phi(s_1,\pi^\star(s_1))^\top(\xi_{r,t} + \xi_{P,t,1} + \eta_{r,t} + \eta_{P,t,1} - \lambda_{t,1}) + \mathbb{E}_{s_2}\left[V^\star_2(s_2) - \overline{V}_{t,2}(s')\,\big|\,s_1,\pi^\star(s_1)\right]\right)\right]
$$

where the last equality is by Lemma B.5.

For $T_M$, we have

$$
\mathbb{E}_{s_1}\left[\boldsymbol{M}^\star_{t,1}(s_1)\left(Q^\star_1(s_1,\pi^\star(s_1)) - \overline{Q}_{t,1}(s_1,\pi^\star(s_1))\right)\right]
$$
$$
= \mathbb{E}_{s_1}\left[\boldsymbol{M}^\star_{t,1}(s_1)\left(\rho(s_1,\pi^\star(s_1))\left(Q^\star_1(s_1,\pi^\star(s_1)) - \phi(s_1,\pi^\star(s_1))^\top(\overline{\theta}_{r,t} + \overline{\theta}_{P,t,1})\right)\right.\right.
$$
$$
\left.\left. + \left(1 - \rho(s_1,\pi^\star(s_1))\right)\left(Q^\star_1(s_1,\pi^\star(s_1)) - \phi(s_1,\pi^\star(s_1))^\top\overline{\theta}_{r,t} - (H - h)\right)\right)\right]
$$
$$
\leq \mathbb{E}_{s_1}\left[\boldsymbol{M}^\star_{t,1}(s_1)\left(\rho(s_1,\pi^\star(s_1))\left(-\phi(s_1,\pi^\star(s_1))^\top(\xi_{r,t} + \xi_{P,t,1} + \eta_{r,t} + \eta_{P,t,1} - \lambda_{t,1})\right.\right.\right.
$$
$$
\left.\left.\left. + \mathbb{E}_{s_2}\left[V^\star_2(s_2) - \overline{V}_{t,2}(s')\,\big|\,s_1,\pi^\star(s_1)\right]\right) + \left(1 - \rho(s_1,\pi^\star(s_1))\right)\left(-\phi(s_1,\pi^\star(s_1))\right)^\top(\eta_{r,t} + \xi_{r,t})\right)\right]
$$

where the inequality is by similar arguments as in the case of $T_U$ and $T_L$.

So putting $T_L, T_M, T_U$ together, we have

$$
V^\star - \overline{V}_t
$$
$$
\leq \mathbb{E}_{s_1}\left[\underbrace{\left(\boldsymbol{U}^\star_{t,1}(s_1) + \boldsymbol{M}^\star_{t,1}(s_1)(1 - \rho(s_1,\pi^\star(s_1)))\right)}_{=:\boldsymbol{UM}^\star_{t,1}(s_1)}(-\phi(s_1,\pi^\star(s_1)))^\top(\eta_{r,t} + \xi_{r,t})\right.
$$
$$
+ \underbrace{\left(\boldsymbol{L}^\star_{t,1}(s_1) + \boldsymbol{M}^\star_{t,1}(s_1)\rho(s_1,\pi^\star(s_1))\right)}_{=:\boldsymbol{LM}^\star_{t,1}(s_1)}\left(-\phi(s_1,\pi^\star(s_1))^\top(\xi_{r,t} + \xi_{P,t,1} + \eta_{r,t} + \eta_{P,t,1} - \lambda_{t,1})\right.
$$
$$
\left.\left. + \mathbb{E}_{s_2}\left[V^\star_2(s_2) - \overline{V}_{t,2}(s')\,\big|\,s_1,\pi^\star(s_1)\right]\right)\right]
$$

Keeping expanding the last term, we arrive at

$$
V^\star - \overline{V}_t \leq \mathbb{E}_{\tau \sim \pi^\star}\left[\sum_{h=1}^{H}\left(\boldsymbol{UM}^\star_{t,h}(s_h)\prod_{i=1}^{h-1}\boldsymbol{LM}^\star_{t,i}(s_i)\right)\left(-\phi(s_h,a_h)^\top(\xi_{r,t} + \eta_{r,t})\right)\right.
$$
$$
\left. + \sum_{h=1}^{H}\left(\prod_{i=1}^{h}\boldsymbol{LM}^\star_{t,i}(s_i)\right)\left(-\phi(s_h,a_h)^\top(\xi_{r,t} + \xi_{P,t,h} + \eta_{r,t} + \eta_{P,t,h} - \lambda_{t,h})\right)\right]
$$
$$
= \mathbb{E}_{\tau \sim \pi^\star}\left[\sum_{h=1}^{H}\left(\boldsymbol{UM}^\star_{t,h}(s_h)\prod_{i=1}^{h-1}\boldsymbol{LM}^\star_{t,i}(s_i) + \prod_{i=1}^{h}\boldsymbol{LM}^\star_{t,i}(s_i)\right)\left(-\phi(s_h,a_h)^\top(\xi_{r,t} + \eta_{r,t})\right)\right.
$$
$$
\left. + \sum_{h=1}^{H}\left(\prod_{i=1}^{h}\boldsymbol{LM}^\star_{t,i}(s_i)\right)\left(-\phi(s_h,a_h)^\top(\xi_{P,t,h} + \eta_{P,t,h} - \lambda_{t,h})\right)\right]
$$
$$
=: (-\phi^\star_r)^\top(\xi_{r,t} + \eta_{r,t}) + \sum_{h=1}^{H}(-\phi^\star_{P,h})^\top(\xi_{P,t,h} + \eta_{P,t,h} - \lambda_{t,h})
$$

where in the last step we defined $\phi_{\mathrm{r}}^{\star}$ and $\phi_{\mathrm{P},h}^{\star}$ as

$$\phi_{\mathrm{r}}^{\star} := \mathop{\mathbb{E}}_{\tau \sim \pi^{\star}} \left[ \sum_{h=1}^{H} \left( \boldsymbol{U}\boldsymbol{M}_{t,h}^{\star}(s_h) \prod_{i=1}^{h-1} \boldsymbol{L}\boldsymbol{M}_{t,i}^{\star}(s_i) + \prod_{i=1}^{h} \boldsymbol{L}\boldsymbol{M}_{t,i}^{\star}(s_i) \right) \phi(s_h, a_h) \right],$$

$$\phi_{\mathrm{P},h}^{\star} := \mathop{\mathbb{E}}_{\tau \sim \pi^{\star}} \left[ \left( \prod_{i=1}^{h} \boldsymbol{L}\boldsymbol{M}_{t,i}^{\star}(s_i) \right) \phi(s_h, a_h) \right].$$

Now we also decompose $\widetilde{V}_t - V^{\pi_t^1}$ and get

$$\widetilde{V}_t - V^{\pi_t^1} = \mathop{\mathbb{E}}_{\tau \sim \pi_t^1} \left[ \sum_{h=1}^{H} \phi(s_h, a_h) \right]^{\top} \left( \overline{\theta}_{\mathrm{r},t} - \theta_{\mathrm{r}}^{\star} \right)$$

$$= \mathop{\mathbb{E}}_{\tau \sim \pi_t^1} \left[ \sum_{h=1}^{H} \phi(s_h, a_h) \right]^{\top} (\xi_{\mathrm{r},t} + \eta_{\mathrm{r},t})$$

$$=: (\widetilde{\phi}_t)^{\top} (\xi_{\mathrm{r},t} + \eta_{\mathrm{r},t}).$$

Combining the decomposition together, we obtain

$$(V^{\star} - \overline{V}_t) + (\widetilde{V} - V^{\pi_t^1})$$

$$\leq \left( (-\phi_{\mathrm{r}}^{\star})^{\top} (\xi_{\mathrm{r},t} + \eta_{\mathrm{r},t}) + \sum_{h=1}^{H} (-\phi_{\mathrm{P},h}^{\star})^{\top} (\xi_{\mathrm{P},t,h} + \eta_{\mathrm{P},t,h} - \lambda_{t,h}) \right) + \widetilde{\phi}_t^{\top} (\xi_{\mathrm{r},t} + \eta_{\mathrm{r},t})$$

$$= \left( \widetilde{\phi}_t - \phi^{\star} \right)^{\top} (\xi_{\mathrm{r},t} + \eta_{\mathrm{r},t}) + \sum_{h=1}^{H} (-\phi_{\mathrm{P},h}^{\star})^{\top} (\xi_{\mathrm{P},t,h} + \eta_{\mathrm{P},t,h} - \lambda_{t,h})$$

$$\leq \| \widetilde{\phi}_t - \phi^{\star} \|_{\Sigma_{t-1}^{-1}} \| \eta_{\mathrm{r},t} \|_{\Sigma_{t-1}} + (\widetilde{\phi}_t - \phi^{\star})^{\top} \xi_{\mathrm{r},t}$$

$$\quad + \sum_{h=1}^{H} \| \phi_{\mathrm{P},h}^{\star} \|_{\Sigma_{t-1,h}^{-1}} \left( \| \eta_{\mathrm{P},t,h} \|_{\Sigma_{t-1,h}} + \| \lambda_{t,h} \|_{\Sigma_{t-1,h}} \right) - \sum_{h=1}^{H} \phi_{\mathrm{P},h}^{\star \top} \xi_{\mathrm{P},t,h}$$

$$\leq \| \widetilde{\phi}_t - \phi^{\star} \|_{\Sigma_{t-1}^{-1}} \epsilon_{\mathrm{r},\eta} - (\widetilde{\phi}_t - \phi^{\star})^{\top} \xi_{\mathrm{r},t} + \sqrt{\sum_{h=1}^{H} \| \phi_{\mathrm{P},h}^{\star} \|_{\Sigma_{t-1,h}^{-1}}^2} \cdot \sqrt{H(\epsilon_{\mathrm{P},\eta} + \epsilon_\lambda)^2} - \sum_{h=1}^{H} \phi_{\mathrm{P},h}^{\star \top} \xi_{\mathrm{P},t,h}$$

$$= \underbrace{\| \widetilde{\phi}_t - \phi^{\star} \|_{\Sigma_{t-1}^{-1}} \epsilon_{\mathrm{r},\eta} - (\widetilde{\phi}_t - \phi^{\star})^{\top} \xi_{\mathrm{r},t}}_{(a)}$$

$$+ \underbrace{\sqrt{\sum_{h=1}^{H} \min\left\{ 1, \| \phi_{\mathrm{P},h}^{\star} \|_{\Sigma_{t-1,h}^{-1}}^2 \right\}} \cdot \sqrt{H(\epsilon_{\mathrm{P},\eta} + \epsilon_\lambda)^2} - \sum_{h=1}^{H} \phi_{\mathrm{P},h}^{\star \top} \xi_{\mathrm{P},t,h}}_{(b)}$$

where the second inequality is by Cauchy-Schwarz inequality, the third inequality is Lemma B.12 and Cauchy-Schwarz inequality again, and the last step is by the fact that $\| \phi_{\mathrm{P},h}^{\star} \|_{\Sigma_{t-1,h}^{-1}}^2 \leq 1/\lambda \leq 1$.

**For (a),** we note that $(\widetilde{\phi}_t - \phi^{\star})^{\top} \xi_{\mathrm{r},t} \sim \mathcal{N}(0, \sigma_{\mathrm{r}}^2 \| \widetilde{\phi}_t - \phi^{\star} \|_{\Sigma_{t-1}^{-1}}^2)$. So by setting $\sigma_{\mathrm{r}} \geq \epsilon_{\mathrm{r},\eta}$, we have $(a) \leq 0$ with probability at least $\mathrm{F}(-1)$.

**For (b),** similarly, we note that $\sum_{h=1}^{H} \phi_{\mathrm{P},h}^{\star \top} \xi_{\mathrm{P},t,h} \sim \mathcal{N}(0, \sigma_{\mathrm{P}}^2 \sum_{h=1}^{H} \| \phi_{\mathrm{P},h}^{\star} \|_{\Sigma_{t-1,h}^{-1}}^2)$. So by setting $\sigma_{\mathrm{P}} \geq \sqrt{H(\epsilon_{\mathrm{P},\eta} + \epsilon_\lambda)^2}$, we have $(b) \leq 0$ with probability at least $\mathrm{F}(-1)$.

**Conclusion.** Finally, we note that (a) and (b) are independent, so the probability that both (a) and (b) hold is at least $\mathrm{F}^2(-1)$. $\qquad\square$

**Lemma B.14.** *It holds that*

$$\sum_{t=1}^{T}\sum_{h=1}^{H} \boldsymbol{L}_{t,h}^{\mathsf{C}}(s_{t,h}) \leq L_{\max}.$$

*Proof.* Denote $a_{t,h} = \pi_t(s_{t,h})$. Then, we have

$$
\begin{aligned}
\sum_{t=1}^{T}\sum_{h=1}^{H} \boldsymbol{L}_{t,h}^{\mathsf{C}}(s_{t,h}) &= \sum_{t=1}^{T}\sum_{h=1}^{H} \mathbb{1}\left\{\|\phi(s_{t,h},a_{t,h})\|_{\Sigma_{t-1,h}^{-1}}^2 > \alpha_L^2\right\} \\
&\leq \sum_{t=1}^{T}\sum_{h=1}^{H} \min\left\{\frac{\|\phi(s_{t,h},a_{t,h})\|_{\Sigma_{t-1,h}^{-1}}^2}{\alpha_L^2}, 1\right\} \\
&\leq \frac{1}{\alpha_L^2}\sum_{h=1}^{H}\sum_{t=1}^{T} \min\left\{\|\phi(s_{t,h},a_{t,h})\|_{\Sigma_{t-1,h}^{-1}}^2, 1\right\} \\
&\leq \frac{H}{\alpha_L^2}\cdot 2d\log\left(\frac{\lambda+T}{\lambda}\right) = L_{\max}
\end{aligned}
$$

where the second inequality uses the fact that $\alpha_{\mathrm{L}} < 1$, and the last inequality is by Lemma D.6. $\square$

**Lemma B.15.** *Assume all events listed in Lemma B.12 hold. Then, the following holds*

$$
\begin{aligned}
\left|\sum_t\left((\overline{V}_t - V^{\pi_t^0}) - (\widetilde{V}_t - V^{\pi_t^1})\right)\right| = \widetilde{O}\bigg( & V_{\max}L_{\max} + V_{\max}\sqrt{HT} \\
&+ (\epsilon_{\mathrm{r},\xi} + \epsilon_{\mathrm{r},\eta})\left(\sqrt{dT} + L_{\max} + \sqrt{HT} + \frac{\epsilon T\sqrt{\pi}}{2\sigma_{\mathrm{r}}}\right) \\
&+ (\epsilon_{\mathrm{P},\xi} + \epsilon_{\mathrm{P},\eta} + \epsilon_\lambda)\left(H\sqrt{dT} + L_{\max} + \sqrt{HT}\right).\bigg)
\end{aligned}
$$

*Proof.* By definition, we have

$$
\begin{aligned}
&\left|\sum_{t=1}^{T}\left((\overline{V}_t - V^{\pi_t^0}) - (\widetilde{V}_t - V^{\pi_t^1})\right)\right| \\
&= \left|\sum_{t=1}^{T} \mathop{\mathbb{E}}_{s_1,a_1\sim\pi_t^0}\left[\overline{Q}_{t,1}(s_1,a_1) - Q_1^{\pi_t^0}(s_1,a_1)\right] - \sum_{t=1}^{T}\sum_{h=1}^{H}\mathop{\mathbb{E}}_{s_h,a_h\sim\pi_t^1}\phi(s_h,a_h)\left(\overline{\theta}_{\mathrm{r},t} - \theta_{\mathrm{r}}^\star\right)\right|
\end{aligned}
$$

By triangle inequality, we have

$$
\begin{aligned}
&\leq \left|\sum_{t=1}^{T} \mathop{\mathbb{E}}_{s_1,a_1\sim\pi_t^0}\left[\boldsymbol{L}_{t,1}(s_{t,1})\left(\overline{Q}_{t,1}(s_1,a_1) - Q_1^{\pi_t^0}(s_1,a_1)\right)\right] - \sum_{t=1}^{T}\sum_{h=1}^{H}\mathop{\mathbb{E}}_{s_h,a_h\sim\pi_t^1}\phi(s_h,a_h)\left(\xi_{\mathrm{r},t} + \eta_{\mathrm{r},t}\right)\right| \\
&\quad + 2V_{\max}\sum_{t=1}^{T}\mathop{\mathbb{E}}_{s_1,a_1\sim\pi_t^0}\left[\boldsymbol{L}_{t,1}^{\mathsf{C}}(s_{t,1})\right]
\end{aligned}
$$

For the first term, conditioning on $\boldsymbol{L}_{t,1}(s_{t,1})$, we have

$$
\begin{aligned}
&= \left|\sum_{t=1}^{T}\mathop{\mathbb{E}}_{s_1,a_1\sim\pi_t^0}\left[\boldsymbol{L}_{t,1}(s_{t,1})\left(\phi(s_1,a_1)^\top(\overline{\theta}_{\mathrm{r},t} + \overline{\theta}_{\mathrm{P},t,1}) - Q_1^{\pi_t^0}(s_1,a_1)\right)\right]\right. \\
&\quad \left.- \sum_{t=1}^{T}\sum_{h=1}^{H}\mathop{\mathbb{E}}_{s_h,a_h\sim\pi_t^1}\phi(s_h,a_h)\left(\xi_{\mathrm{r},t} + \eta_{\mathrm{r},t}\right)\right| + 2V_{\max}\sum_{t=1}^{T}\mathop{\mathbb{E}}_{s_1,a_1\sim\pi_t^0}\left[\boldsymbol{L}_{t,1}^{\mathsf{C}}(s_{t,1})\right]
\end{aligned}
$$

Applying triangle inequality again, we get

$$\leq \left| \sum_{t=1}^{T} \mathop{\mathbb{E}}_{s_1,a_1\sim\pi_t^0} \left[ \left( \phi(s_1,a_1)^\top(\overline{\theta}_{\mathrm{r},t} + \overline{\theta}_{\mathrm{P},t,1}) - Q_1^{\pi_t^0}(s_1,a_1) \right) \right] - \sum_{t=1}^{T}\sum_{h=1}^{H} \mathop{\mathbb{E}}_{s_h,a_h\sim\pi_t^1} \phi(s_h,a_h)(\xi_{\mathrm{r},t} + \eta_{\mathrm{r},t}) \right|$$

$$+ 2V_{\max} \sum_{t=1}^{T} \mathop{\mathbb{E}}_{s_1,a_1\sim\pi_t^0} \left[ \boldsymbol{L}_{t,1}^{\texttt{C}}(s_{t,1}) \right]$$

$$+ \left| \sum_{t=1}^{T} \mathop{\mathbb{E}}_{s_1,a_1\sim\pi_t^0} \left[ \boldsymbol{L}_{t,1}^{\texttt{C}}(s_{t,1}) \left( \phi(s_1,a_1)^\top(\overline{\theta}_{\mathrm{r},t} + \overline{\theta}_{\mathrm{P},t,1}) - Q_1^{\pi_t^0}(s_1,a_1) \right) \right] \right|$$

Applying Lemma B.5 and the triangle inequality, we get

$$\leq \left| \sum_{t=1}^{T} \mathop{\mathbb{E}}_{s_2\sim\pi_t^0} \left[ \overline{V}_{t,2}(s_2) - V_2^{\pi_t^0}(s_2) \right] - \sum_{t=1}^{T}\sum_{h=2}^{H} \mathop{\mathbb{E}}_{s_h,a_h\sim\pi_t^1} \phi(s_h,a_h)(\xi_{\mathrm{r},t} + \eta_{\mathrm{r},t}) \right.$$

$$\left. + \sum_{t=1}^{T} \left( \mathop{\mathbb{E}}_{s_1,a_1\sim\pi_t^0} \phi^\top(s_1,a_1)(\xi_{\mathrm{r},t} + \xi_{\mathrm{P},t,1} + \eta_{\mathrm{r},t} + \eta_{\mathrm{P},t,1} - \lambda_{t,1}) - \mathop{\mathbb{E}}_{s_1,a_1\sim\pi_t^1} \phi^\top(s_1,a_1)(\xi_{\mathrm{r},t} + \eta_{\mathrm{r},t}) \right) \right|$$

$$+ 2V_{\max} \sum_{t=1}^{T} \mathop{\mathbb{E}}_{s_1,a_1\sim\pi_t^0} \left[ \boldsymbol{L}_{t,1}^{\texttt{C}}(s_{t,1}) \right]$$

$$+ \left| \sum_{t=1}^{T} \mathop{\mathbb{E}}_{s_1,a_1\sim\pi_t^0} \left[ \boldsymbol{L}_{t,1}^{\texttt{C}}(s_{t,1}) \left( \phi(s_1,a_1)^\top(\overline{\theta}_{\mathrm{r},t} + \overline{\theta}_{\mathrm{P},t,1}) - Q_1^{\pi_t^0}(s_1,a_1) \right) \right] \right|$$

Keep expanding the first term, we get

$$\left| \sum_{t=1}^{T} \left( (\overline{V}_t - V^{\pi_t^0}) - (\widetilde{V}_t - V^{\pi_t^1}) \right) \right|$$

$$\leq 2V_{\max} \sum_{h=1}^{H}\sum_{t=1}^{T} \mathop{\mathbb{E}}_{s_h,a_h\sim\pi_t^0} \left[ \boldsymbol{L}_{t,h}^{\texttt{C}}(s_{t,h}) \right]$$

$$+ \sum_{h=1}^{H} \left| \sum_{t=1}^{T} \mathop{\mathbb{E}}_{s_h,a_h\sim\pi_t^0} \left[ \boldsymbol{L}_{t,h}^{\texttt{C}}(s_{t,h}) \left( \phi(s_h,a_h)^\top(\overline{\theta}_{\mathrm{r},t} + \overline{\theta}_{\mathrm{P},t,h}) - Q_h^{\pi_t^0}(s_h,a_h) \right) \right] \right|$$

$$+ \left| \sum_{t=1}^{T}\sum_{h=1}^{H} \left( \mathop{\mathbb{E}}_{s_h,a_h\sim\pi_t^0} \mathop{\mathbb{E}}_{\widetilde{s}_h,\widetilde{a}_h\sim\pi_t^1} \left( \phi(s_h,a_h) - \phi(\widetilde{s}_h,\widetilde{a}_h) \right)^\top (\xi_{\mathrm{r},t} + \eta_{\mathrm{r},t}) \right) \right|$$

$$+ \left| \sum_{t=1}^{T}\sum_{h=1}^{H} \mathop{\mathbb{E}}_{s_h,a_h\sim\pi_t^0} \phi^\top(s_h,a_h)\left( \xi_{\mathrm{P},t,h} + \eta_{\mathrm{P},t,h} - \lambda_{t,h} \right) \right|$$

$$=: \texttt{T}_1 + \texttt{T}_2 + \texttt{T}_3 + \texttt{T}_4.$$

We bound each term separately.

**Bounding $\texttt{T}_1$.** By Hoeffding's inequality and Lemma B.14, we have

$$\texttt{T}_1 \leq 2V_{\max} \sum_{h=1}^{H}\sum_{t=1}^{T} \left[ \boldsymbol{L}_{t,h}^{\texttt{C}}(s_{t,h}) \right] + 2V_{\max}\sqrt{\frac{HT}{2}\log(1/\delta)}$$

$$\leq 2V_{\max} \left( L_{\max} + \sqrt{\frac{HT}{2}\log(1/\delta)} \right).$$

with probability at least $1 - \delta$.

**Bounding** $\mathrm{T}_2$**.** By definition and Lemma B.4, we have

$$\mathrm{T}_2 = \sum_{h=1}^{H} \left| \sum_{t=1}^{T} \mathbb{E}_{s_h,a_h \sim \pi_t^0} \left[ \boldsymbol{L}_{t,h}^{\mathsf{C}}(s_{t,h}) \Big( \phi(s_h,a_h)^\top (\xi_{\mathrm{r},t} + \eta_{\mathrm{r},t} + \theta_\mathrm{r}^\star + \xi_{\mathrm{P},t,h} + \eta_{\mathrm{P},t,h} + \lambda_{t,h} + \theta_{\mathrm{P},t,h}^\star ) \right. \right.$$

$$\left. \left. - Q_h^{\pi_t^0}(s_h,a_h) \Big) \right] \right|$$

$$\leq \sum_{h=1}^{H} \left| \sum_{t=1}^{T} \mathbb{E}_{s_h,a_h \sim \pi_t^0} \left[ \boldsymbol{L}_{t,h}^{\mathsf{C}}(s_{t,h}) \Big( \phi(s_h,a_h)^\top (\theta_\mathrm{r}^\star + \theta_{\mathrm{P},t,h}^\star) - Q_h^{\pi_t^0}(s_h,a_h) \right. \right.$$

$$\left. \left. + \phi^\top(s_h,a_h)(\xi_{\mathrm{r},t} + \eta_{\mathrm{r},t}) + \phi^\top(s_h,a_h)(\xi_{\mathrm{P},t,h} + \eta_{\mathrm{P},t,h} + \lambda_{t,h}) \Big) \right] \right|$$

$$\leq \sum_{h=1}^{H} \left| \sum_{t=1}^{T} \mathbb{E}_{s_h,a_h \sim \pi_t^0} \left[ \boldsymbol{L}_{t,h}^{\mathsf{C}}(s_{t,h}) \Big( \phi(s_h,a_h)^\top (\theta_\mathrm{r}^\star + \theta_{\mathrm{P},t,h}^\star) - Q_h^{\pi_t^0}(s_h,a_h) \right. \right.$$

$$+ \|\phi(s_h,a_h)\|_{\Sigma_{t-1}^{-1}} (\|\xi_{\mathrm{r},t}\|_{\Sigma_{t-1}} + \|\eta_{\mathrm{r},t}\|_{\Sigma_{t-1}})$$

$$\left. \left. + \|\phi(s_h,a_h)\|_{\Sigma_{t-1,h}^{-1}} (\|\xi_{\mathrm{P},t,h}\|_{\Sigma_{t-1,h}} + \|\eta_{\mathrm{P},t,h}\|_{\Sigma_{t-1,h}} + \|\lambda_{t,h}\|_{\Sigma_{t-1,h}}) \Big) \right] \right|.$$

We note that $|\phi(s_h,a_h)^\top (\theta_\mathrm{r}^\star + \theta_{\mathrm{P},t,h}^\star)| \leq V_{\max}$ and $|Q_h^{\pi_t^0}(s_h,a_h)| \leq V_{\max}$. Moreover, we have $\|\phi(s_h,a_h)\|_{\Sigma_{t-1}^{-1}} \leq 1/\sqrt{\lambda}$ and $\|\phi(s_h,a_h)\|_{\Sigma_{t-1,h}^{-1}} \leq 1/\sqrt{\lambda}$. Apply triangle inequality and inserting these upper bounds back, we obtain

$$\mathrm{T}_2 \leq \sum_{h=1}^{H} \left| \sum_{t=1}^{T} \mathbb{E}_{s_h,a_h \sim \pi_t^0} \left[ \boldsymbol{L}_{t,h}^{\mathsf{C}}(s_{t,h}) \Big( \phi(s_h,a_h)^\top (\theta_\mathrm{r}^\star + \theta_{\mathrm{P},t,h}^\star) - Q_h^{\pi_t^0}(s_h,a_h) \Big) \right] \right|$$

$$+ \sum_{h=1}^{H} \sum_{t=1}^{T} \mathbb{E}_{s_h,a_h \sim \pi_t^0} \left[ \boldsymbol{L}_{t,h}^{\mathsf{C}}(s_{t,h}) \|\phi(s_h,a_h)\|_{\Sigma_{t-1}^{-1}} (\|\xi_{\mathrm{r},t}\|_{\Sigma_{t-1}} + \|\eta_{\mathrm{r},t}\|_{\Sigma_{t-1}}) \right]$$

$$+ \sum_{h=1}^{H} \sum_{t=1}^{T} \mathbb{E}_{s_h,a_h \sim \pi_t^0} \left[ \boldsymbol{L}_{t,h}^{\mathsf{C}}(s_{t,h}) \|\phi(s_h,a_h)\|_{\Sigma_{t-1,h}^{-1}} (\|\xi_{\mathrm{P},t,h}\|_{\Sigma_{t-1,h}} + \|\eta_{\mathrm{P},t,h}\|_{\Sigma_{t-1,h}} + \|\lambda_{t,h}\|_{\Sigma_{t-1,h}}) \right]$$

$$\leq 2V_{\max} \sum_{h=1}^{H} \sum_{t=1}^{T} \mathbb{E}_{s_h,a_h \sim \pi_t^0} \left[ \boldsymbol{L}_{t,h}^{\mathsf{C}}(s_{t,h}) \right]$$

$$+ \sqrt{\lambda^{-1}} \sum_{h=1}^{H} \sum_{t=1}^{T} \mathbb{E}_{s_h,a_h \sim \pi_t^0} \left[ \boldsymbol{L}_{t,h}^{\mathsf{C}}(s_{t,h})(\epsilon_{\mathrm{r},\xi} + \epsilon_{\mathrm{r},\eta} + \epsilon_{\mathrm{P},\xi} + \epsilon_{\mathrm{P},\eta} + \epsilon_\lambda) \right]$$

$$= \underbrace{\sum_{h=1}^{H} \sum_{t=1}^{T} \mathbb{E}_{s_h,a_h \sim \pi_t^0} \left[ \boldsymbol{L}_{t,h}^{\mathsf{C}}(s_{t,h}) \right]}_{(*)} \left( 2V_{\max} + \frac{\epsilon_{\mathrm{r},\xi} + \epsilon_{\mathrm{r},\eta} + \epsilon_{\mathrm{P},\xi} + \epsilon_{\mathrm{P},\eta} + \epsilon_\lambda}{\sqrt{\lambda}} \right).$$

We notice that $(*)$ can be bounded by Hoeffding's inequality and Lemma B.14, as we did similarly for $\mathrm{T}_1$:

$$(*) \leq L_{\max} + \sqrt{\frac{HT}{2} \log(1/\delta)}$$

Hence, we have

$$\mathrm{T}_2 \leq \left( L_{\max} + \sqrt{\frac{HT}{2} \log(1/\delta)} \right) \left( 2V_{\max} + \frac{\epsilon_{\mathrm{r},\xi} + \epsilon_{\mathrm{r},\eta} + \epsilon_{\mathrm{P},\xi} + \epsilon_{\mathrm{P},\eta} + \epsilon_\lambda}{\sqrt{\lambda}} \right)$$

with probability at least $1 - \delta$.

**Bounding $\mathrm{T}_3$.** By definition, we have

$$\mathrm{T}_3 = \left| \sum_{t=1}^{T} \mathop{\mathbb{E}}_{\tau \sim \pi_t^0} \mathop{\mathbb{E}}_{\widetilde{\tau} \sim \pi_t^1} \left( \phi(\tau) - \phi(\widetilde{\tau}) \right) (\xi_{\mathrm{r},t} + \eta_{\mathrm{r},t}) \right|$$

By Cauchy-Schwarz inequality, we have

$$\leq \sum_{t=1}^{T} \mathop{\mathbb{E}}_{\tau \sim \pi_t^0} \mathop{\mathbb{E}}_{\widetilde{\tau} \sim \pi_t^1} \left\| \phi(\tau) - \phi(\widetilde{\tau}) \right\|_{\Sigma_{t-1}^{-1}} \left( \| \xi_{\mathrm{r},t} \|_{\Sigma_{t-1}} + \| \eta_{\mathrm{r},t} \|_{\Sigma_{t-1}} \right)$$

$$\leq \left( \epsilon_{\mathrm{r},\xi} + \epsilon_{\mathrm{r},\eta} \right) \sum_{t=1}^{T} \mathop{\mathbb{E}}_{\tau \sim \pi_t^0} \mathop{\mathbb{E}}_{\widetilde{\tau} \sim \pi_t^1} \left\| \phi(\tau) - \phi(\widetilde{\tau}) \right\|_{\Sigma_{t-1}^{-1}}$$

Since $\| \phi(\tau) - \phi(\widetilde{\tau}) \|_{\Sigma_{t-1}^{-1}} \leq 2/\sqrt{\lambda}$, by Hoeffding's inequality, we have

$$\leq \left( \epsilon_{\mathrm{r},\xi} + \epsilon_{\mathrm{r},\eta} \right) \left( \sum_{t=1}^{T} \left\| \phi(\tau) - \phi(\widetilde{\tau}) \right\|_{\Sigma_{t-1}^{-1}} + \sqrt{\frac{2T \log(1/\delta)}{\lambda}} \right)$$

$$= \left( \epsilon_{\mathrm{r},\xi} + \epsilon_{\mathrm{r},\eta} \right) \Bigg( \underbrace{\sum_{t=1}^{T} Z_t \left\| \phi(\tau) - \phi(\widetilde{\tau}) \right\|_{\Sigma_{t-1}^{-1}}}_{(\mathrm{i})} + \underbrace{\sum_{t=1}^{T} (1 - Z_t) \left\| \phi(\tau) - \phi(\widetilde{\tau}) \right\|_{\Sigma_{t-1}^{-1}}}_{(\mathrm{ii})} + \sqrt{\frac{2T \log(1/\delta)}{\lambda}} \Bigg).$$

To bound (i), we have

$$(\mathrm{i}) = \sum_{t=1}^{T} Z_t \min \left\{ 2/\sqrt{\lambda}, \left\| \phi(\tau) - \phi(\widetilde{\tau}) \right\|_{\Sigma_{t-1}^{-1}} \right\}$$

$$\leq \sqrt{T \sum_{t=1}^{T} Z_t \min \left\{ 4/\lambda, \left\| \phi(\tau) - \phi(\widetilde{\tau}) \right\|_{\Sigma_{t-1}^{-1}}^2 \right\}}$$

$$\leq 2\sqrt{T \sum_{t=1}^{T} Z_t \min \left\{ 1, \left\| \phi(\tau) - \phi(\widetilde{\tau}) \right\|_{\Sigma_{t-1}^{-1}}^2 \right\}}$$

$$\leq 2\sqrt{T \cdot 2d \log \left( \frac{\lambda + 4T}{\lambda} \right)}$$

where the last inequality is Lemma D.6. To bound (ii), we have

$$(\mathrm{ii}) = \sum_{t=1}^{T} (1 - Z_t) \left\| \phi(\tau) - \phi(\widetilde{\tau}) \right\|_{\Sigma_{t-1}^{-1}}$$

$$\leq \sum_{t=1}^{T} (1 - Z_t) \cdot \epsilon \sqrt{\pi} / (2\sigma_{\mathrm{r}})$$

$$\leq \epsilon T \sqrt{\pi} / (2\sigma_{\mathrm{r}})$$

where the first inequality is by Lemma B.6. Putting the two upper bounds together, we have

$$\mathrm{T}_3 \leq \left( \epsilon_{\mathrm{r},\xi} + \epsilon_{\mathrm{r},\eta} \right) \left( 2\sqrt{T \cdot 2d \log \left( \frac{\lambda + 4T}{\lambda} \right)} + \epsilon T \sqrt{\pi} / (2\sigma_{\mathrm{r}}) + \sqrt{\frac{2T \log(1/\delta)}{\lambda}} \right).$$

**Bounding $\mathrm{T}_4$.** By Cauchy-Schwarz inequality, we have

$$\mathrm{T}_4 \leq \left| \sum_{t=1}^{T} \sum_{h=1}^{H} \mathop{\mathbb{E}}_{s_h, a_h \sim \pi_t^0} \| \phi(s_h, a_h) \|_{\Sigma_{t-1,h}^{-1}} \left( \| \xi_{\mathrm{P},t,h} \|_{\Sigma_{t-1,h}} + \| \eta_{\mathrm{P},t,h} \|_{\Sigma_{t-1,h}} + \| \lambda_{t,h} \|_{\Sigma_{t-1,h}} \right) \right|$$

$$\leq \left( \epsilon_{\mathrm{P},\xi} + \epsilon_{\mathrm{P},\eta} + \epsilon_{\lambda} \right) \sum_{h=1}^{H} \sum_{t=1}^{T} \mathop{\mathbb{E}}_{s_h, a_h \sim \pi_t^0} \| \phi(s_h, a_h) \|_{\Sigma_{t-1,h}^{-1}}$$

Since $\|\phi(s_h, a_h)\|_{\Sigma_{t-1,h}^{-1}} \leq 1/\sqrt{\lambda}$, by Hoeffding's inequality, we have

$$\leq \left( \epsilon_{\mathrm{P},\xi} + \epsilon_{\mathrm{P},\eta} + \epsilon_\lambda \right) \left( \sum_{h=1}^{H} \sum_{t=1}^{T} \|\phi(s_h, a_h)\|_{\Sigma_{t-1,h}^{-1}} + \sqrt{\frac{TH \log(1/\delta)}{2\lambda}} \right)$$

$$= \left( \epsilon_{\mathrm{P},\xi} + \epsilon_{\mathrm{P},\eta} + \epsilon_\lambda \right) \left( \sum_{h=1}^{H} \sum_{t=1}^{T} \min \left\{ 1/\sqrt{\lambda}, \|\phi(s_h, a_h)\|_{\Sigma_{t-1,h}^{-1}} \right\} + \sqrt{\frac{TH \log(1/\delta)}{2\lambda}} \right)$$

$$\leq \left( \epsilon_{\mathrm{P},\xi} + \epsilon_{\mathrm{P},\eta} + \epsilon_\lambda \right) \left( \sum_{h=1}^{H} \sqrt{T \sum_{t=1}^{T} \min \left\{ 1/\lambda, \|\phi(s_h, a_h)\|_{\Sigma_{t-1,h}^{-1}}^2 \right\}} + \sqrt{\frac{TH \log(1/\delta)}{2\lambda}} \right)$$

Since $\lambda \geq 1$, we have

$$\leq \left( \epsilon_{\mathrm{P},\xi} + \epsilon_{\mathrm{P},\eta} + \epsilon_\lambda \right) \left( \sum_{h=1}^{H} \sqrt{T \sum_{t=1}^{T} \min \left\{ 1, \|\phi(s_h, a_h)\|_{\Sigma_{t-1,h}^{-1}}^2 \right\}} + \sqrt{\frac{TH \log(1/\delta)}{2\lambda}} \right)$$

$$\leq \left( \epsilon_{\mathrm{P},\xi} + \epsilon_{\mathrm{P},\eta} + \epsilon_\lambda \right) \left( H \sqrt{T \cdot 2d \log \left( \frac{\lambda + T}{\lambda} \right)} + \sqrt{\frac{TH \log(1/\delta)}{2\lambda}} \right)$$

where the last inequality is Lemma D.6.

**Conclusion.** Combining the above bounds, we have

$$\left| \sum_{t=1}^{T} \left( (\overline{V}_t - V^{\pi_t^0}) - (\widetilde{V}_t - V^{\pi_t^1}) \right) \right|$$

$$\leq 2V_{\max} \left( L_{\max} + \sqrt{\frac{HT}{2} \log(1/\delta)} \right)$$

$$+ \left( L_{\max} + \sqrt{\frac{HT}{2} \log(1/\delta)} \right) \left( 2V_{\max} + \frac{\epsilon_{\mathrm{r},\xi} + \epsilon_{\mathrm{r},\eta} + \epsilon_{\mathrm{P},\xi} + \epsilon_{\mathrm{P},\eta} + \epsilon_\lambda}{\sqrt{\lambda}} \right)$$

$$+ \left( \epsilon_{\mathrm{r},\xi} + \epsilon_{\mathrm{r},\eta} \right) \left( 2\sqrt{T \cdot 2d \log \left( \frac{\lambda + 4T}{\lambda} \right)} + \frac{\epsilon T \sqrt{\pi}}{2\sigma_{\mathrm{r}}} + \sqrt{\frac{2T \log(1/\delta)}{\lambda}} \right)$$

$$+ \left( \epsilon_{\mathrm{P},\xi} + \epsilon_{\mathrm{P},\eta} + \epsilon_\lambda \right) \left( H \sqrt{T \cdot 2d \log \left( \frac{\lambda + T}{\lambda} \right)} + \sqrt{\frac{TH \log(1/\delta)}{2\lambda}} \right)$$

$$\leq 4V_{\max} \left( L_{\max} + \sqrt{\frac{HT}{2} \log(1/\delta)} \right)$$

$$+ \left( \epsilon_{\mathrm{r},\xi} + \epsilon_{\mathrm{r},\eta} \right) \left( 2\sqrt{T \cdot 2d \log \left( \frac{\lambda + 4T}{\lambda} \right)} + \frac{\epsilon T \sqrt{\pi}}{2\sigma_{\mathrm{r}}} + \frac{L_{\max} + 2\sqrt{TH \log(1/\delta)/2}}{\sqrt{\lambda}} \right)$$

$$+ \left( \epsilon_{\mathrm{P},\xi} + \epsilon_{\mathrm{P},\eta} + \epsilon_\lambda \right) \left( H \sqrt{T \cdot 2d \log \left( \frac{\lambda + T}{\lambda} \right)} + \frac{L_{\max} + 2\sqrt{TH \log(1/\delta)/2}}{\sqrt{\lambda}} \right)$$

$$= \widetilde{O} \Big( V_{\max} L_{\max} + V_{\max} \sqrt{HT}$$

$$+ (\epsilon_{\mathrm{r},\xi} + \epsilon_{\mathrm{r},\eta}) \left( \sqrt{dT} + L_{\max} + \sqrt{HT} + \frac{\epsilon T \sqrt{\pi}}{2\sigma_{\mathrm{r}}} \right)$$

$$+ (\epsilon_{\mathrm{P},\xi} + \epsilon_{\mathrm{P},\eta} + \epsilon_\lambda) \left( H \sqrt{dT} + L_{\max} + \sqrt{HT} \right) \Big).$$

$$\square$$

**Lemma B.16** (Regret decomposition). *Assume all events listed in Lemma B.12 hold. Then, we have*

$$\sum_{t=1}^{T}(V^{\star} - \overline{V}_t) + (\widetilde{V}_t - V^{\pi_t^1})$$

$$\leq \frac{1}{\mathrm{F}^2(-1)} \sum_{t=1}^{T} \left( (\overline{V}_t - V^{\pi_t^0}) - (\widetilde{V}_t - V^{\pi_t^1}) + (V^{\pi_t^0} - \overline{V}_t^-) - (V^{\pi_t^1} - \widetilde{V}_t^-) \right)$$

$$+ \frac{2V_{\max}}{\mathrm{F}^2(-1)} \left( \sqrt{\frac{T \log(1/\delta)}{2}} + 4\delta \right)$$

*with probability at least $1 - \delta$.*

*Proof.* We note that, at any round $t \in [T]$, conditioning on all information collected up to round $t-1$, the randomness of $\overline{V}_t$ and $\widetilde{V}_t$ only comes from the randomness of Gaussian noise variables $\xi_{\mathrm{r},t}, \xi_{\mathrm{P},t,1}, \ldots, \xi_{\mathrm{P},t,H}$. In other words, the values of $\overline{V}_t$ and $\widetilde{V}_t$ are determined once given these Gaussian noise variables. In light of this, we write out the dependence on the Gaussian noise variables explicitly: we treat $\overline{V}_t$ and $\widetilde{V}_t$ as functions of $\xi_{\mathrm{r},t}, \xi_{\mathrm{P},t,1}, \cdots, \xi_{\mathrm{P},t,H}$ and define $\overline{V}_t[\xi_{\mathrm{r},t}, \xi_{\mathrm{P},t,1}, \cdots, \xi_{\mathrm{P},t,H}]$ and $\widetilde{V}_t[\xi_{\mathrm{r},t}^-, \xi_{\mathrm{P},t,1}^-, \cdots, \xi_{\mathrm{P},t,H}^-]$ as the values of $\overline{V}_t$ and $\widetilde{V}_t$ obtained at round $t$ with the Gaussian noise variables $\xi_{\mathrm{r},t}, \xi_{\mathrm{P},1}, \cdots, \xi_{\mathrm{P},H}$. Then, we define a notion of "worst-case" Gaussian noise variables as follows:

$$\xi_{\mathrm{r},t}^-, \xi_{\mathrm{P},t,1}^-, \cdots, \xi_{\mathrm{P},t,H}^- := \underset{\xi_{\mathrm{r},t}^-, \xi_{\mathrm{P},t,1}^-, \cdots, \xi_{\mathrm{P},t,H}^-}{\arg\min} \overline{V}_t[\xi_{\mathrm{r},t}^-, \xi_{\mathrm{P},t,1}^-, \cdots, \xi_{\mathrm{P},t,H}^-] - \widetilde{V}_t[\xi_{\mathrm{r},t}^-, \xi_{\mathrm{P},t,1}^-, \cdots, \xi_{\mathrm{P},t,H}^-]$$

$$\text{s.t.} \quad \|\xi_{\mathrm{r},t}^-\|_{\Sigma_{t-1}} \leq \epsilon_{\mathrm{r},\xi} \quad \text{and} \quad \forall h \in [H] : \|\xi_{\mathrm{P},t,h}^-\|_{\Sigma_{t-1,h}} \leq \epsilon_{\mathrm{P},\xi}.$$

And we denote $\overline{V}_t^-$ and $\widetilde{V}_t^-$ as the value functions specified by $\xi_{\mathrm{r},t}^-, \xi_{\mathrm{P},t,1}^-, \cdots, \xi_{\mathrm{P},t,H}^-$:

$$\overline{V}_t^- := \overline{V}_t[\xi_{\mathrm{r},t}^-, \xi_{\mathrm{P},t,1}^-, \cdots, \xi_{\mathrm{P},t,H}^-], \qquad \widetilde{V}_t^- := \widetilde{V}_t[\xi_{\mathrm{r},t}^-, \xi_{\mathrm{P},t,1}^-, \cdots, \xi_{\mathrm{P},t,H}^-].$$

In other words, $\overline{V}_t^-$ and $\widetilde{V}_t^-$ are counterparts of $\overline{V}_t$ and $\widetilde{V}_t$ that attain the smallest difference, $\overline{V}_t - \widetilde{V}_t$, while the noise variables still satisfy the high probability bounds. By Lemma B.12, we immediately have

$$\Pr(\mathcal{E}_{\mathrm{low}}) := \Pr\left( \overline{V}_t^- - \widetilde{V}_t^- \leq \overline{V}_t - \widetilde{V}_t \right) \geq 1 - \delta/T. \tag{11}$$

We note that here we have $\delta/T$ instead of $\delta$ because we are considering a fixed $t$ and the results of Lemma B.12 are derived by the union bound over all $t \in [T]$ — thus, the probability of the event $\mathcal{E}_{\mathrm{low}}$ is at least $1 - \delta/T$ for a single $t$.

Then, we denote $\mathcal{E}_{\mathrm{opt}}$ as the event that $\overline{V}_t - \widetilde{V}_t \geq V^{\star} - V^{\pi_t^1}$. Thus, by Lemma B.13, we have $\Pr(\mathcal{E}_{\mathrm{opt}}) \geq \mathrm{F}^2(-1)$. Moreover, we denote $p_{\mathrm{alg}}$ as the randomness of the algorithm. Then, we define the joint distribution $p_{\mathrm{opt}}$ of $\overline{V}_t$ and $\widetilde{V}_t$ by restricting $p_{\mathrm{alg}}$ to the event $\mathcal{E}_{\mathrm{opt}}$. Specifically, it is defined as

$$p_{\mathrm{opt}}(\overline{V}_t, \widetilde{V}_t) := \begin{cases} p_{\mathrm{alg}}(\overline{V}_t, \widetilde{V}_t)/\Pr(\mathcal{E}_{\mathrm{opt}}) & \text{if } \mathcal{E}_{\mathrm{opt}} \\ 0 & \text{otherwise} \end{cases}$$

Let $z := \overline{V}_t - \widetilde{V}_t$. Then, we have

$$(V^{\star} - \overline{V}_t) + (\widetilde{V}_t - V^{\pi_t^1}) \leq V^{\star} - \overline{V}_t^- + \widetilde{V}_t^- - V^{\pi_t^1}$$

$$= \underset{z \sim p_{\mathrm{opt}}}{\mathbb{E}} \left[ V^{\star} - \overline{V}_t^- - z + z + \widetilde{V}_t^- - V^{\pi_t^1} \right]$$

$$\leq \underset{z \sim p_{\mathrm{opt}}}{\mathbb{E}} \left[ z - \overline{V}_t^- + \widetilde{V}_t^- \right]$$

$$= \int_z \frac{\mathbb{1}\{\mathcal{E}_{\mathrm{opt}}\} \cdot p_{\mathrm{alg}}(z)}{\Pr(\mathcal{E}_{\mathrm{opt}})} \cdot (z - \overline{V}_t^- + \widetilde{V}_t^-) \, \mathrm{d}z$$

$$= \underset{z \sim p_{\mathrm{alg}}}{\mathbb{E}} \left[ \mathbb{1}\{\mathcal{E}_{\mathrm{opt}}\} \left( z - \overline{V}_t^- + \widetilde{V}_t^- \right) \right] / \Pr(\mathcal{E}_{\mathrm{opt}})$$

where the second inequality is by the definition of $p_{\mathrm{opt}}$, which rules out the case that $\mathcal{E}_{\mathrm{opt}}$ does not hold. Considering the event $\mathcal{E}_{\mathrm{low}}$, we have

$$(V^\star - \overline{V}_t) + (\widetilde{V}_t - V^{\pi_t^1})$$

$$\leq \left( \mathop{\mathbb{E}}_{z \sim p_{\mathrm{alg}}} \left[ \mathbb{1}\{\mathcal{E}_{\mathrm{opt}}\} \mathbb{1}\{\mathcal{E}_{\mathrm{low}}\} \left( z - \overline{V}_t^- + \widetilde{V}_t^- \right) \right] + \mathop{\mathbb{E}}_{z \sim p_{\mathrm{alg}}} \left[ \mathbb{1}\{\mathcal{E}_{\mathrm{opt}}\} \mathbb{1}\{\mathcal{E}_{\mathrm{low}}^{\complement}\} \left( z - \overline{V}_t^- + \widetilde{V}_t^- \right) \right] \right) / \Pr(\mathcal{E}_{\mathrm{opt}})$$

$$\leq \left( \mathop{\mathbb{E}}_{z \sim p_{\mathrm{alg}}} \left[ \mathbb{1}\{\mathcal{E}_{\mathrm{opt}}\} \mathbb{1}\{\mathcal{E}_{\mathrm{low}}\} \left( z - \overline{V}_t^- + \widetilde{V}_t^- \right) \right] + \mathop{\mathbb{E}}_{z \sim p_{\mathrm{alg}}} \left[ \mathbb{1}\{\mathcal{E}_{\mathrm{opt}}\} \mathbb{1}\{\mathcal{E}_{\mathrm{low}}^{\complement}\} 4 V_{\max} \right] \right) / \Pr(\mathcal{E}_{\mathrm{opt}})$$

$$\leq \left( \mathop{\mathbb{E}}_{z \sim p_{\mathrm{alg}}} \left[ \mathbb{1}\{\mathcal{E}_{\mathrm{opt}}\} \mathbb{1}\{\mathcal{E}_{\mathrm{low}}\} \left( z - \overline{V}_t^- + \widetilde{V}_t^- \right) \right] + 4 V_{\max} \delta / T \right) / \Pr(\mathcal{E}_{\mathrm{opt}})$$

where the last inequality is by (11). For the first term, we have

$$\mathop{\mathbb{E}}_{z \sim p_{\mathrm{alg}}} \left[ \mathbb{1}\{\mathcal{E}_{\mathrm{opt}}\} \mathbb{1}\{\mathcal{E}_{\mathrm{low}}\} \left( z - \overline{V}_t^- + \widetilde{V}_t^- \right) \right]$$

$$\leq \mathop{\mathbb{E}}_{z \sim p_{\mathrm{alg}}} \left[ \mathbb{1}\{\mathcal{E}_{\mathrm{low}}\} \left( z - \overline{V}_t^- + \widetilde{V}_t^- \right) \right]$$

$$= \mathop{\mathbb{E}}_{z \sim p_{\mathrm{alg}}} \left[ \left( z - \overline{V}_t^- + \widetilde{V}_t^- \right) \right] + \mathop{\mathbb{E}}_{z \sim p_{\mathrm{alg}}} \left[ \mathbb{1}\{\mathcal{E}_{\mathrm{low}}^{\complement}\} \left( z - \overline{V}_t^- + \widetilde{V}_t^- \right) \right]$$

$$\leq \mathop{\mathbb{E}}_{z \sim p_{\mathrm{alg}}} \left[ \left( z - \overline{V}_t^- + \widetilde{V}_t^- \right) \right] + 4 V_{\max} \delta / T$$

$$= \mathbb{E} \left[ \overline{V}_t - \widetilde{V}_t - \overline{V}_t^- + \widetilde{V}_t^- \right] + 4 V_{\max} \delta / T$$

$$= \mathbb{E} \left[ (\overline{V}_t - V^{\pi_t^0}) - (\widetilde{V}_t - V^{\pi_t^1}) \right] + \mathbb{E} \left[ (V^{\pi_t^0} - \overline{V}_t^-) - (V^{\pi_t^1} - \widetilde{V}_t^-) \right] + 4 V_{\max} \delta / T.$$

where the first inequality holds since $z - \overline{V}_t^- + \widetilde{V}_t^- \geq 0$ conditioning on $\mathcal{E}_{\mathrm{low}}$, and the second inequality is by (11) again.

Recall that $\Pr(\mathcal{E}_{\mathrm{opt}}) \geq \mathrm{F}^2(-1)$. Inserting all of these back, we obtain

$$\sum_{t=1}^{T} (V^\star - \overline{V}_t) + (\widetilde{V}_t - V^{\pi_t^1})$$

$$\leq \mathrm{F}^{-2}(-1) \left( 8 V_{\max} \delta + \sum_{t=1}^{T} \left( \mathbb{E} \left[ (\overline{V}_t - V^{\pi_t^0}) - (\widetilde{V}_t - V^{\pi_t^1}) \right] + \mathbb{E} \left[ (V^{\pi_t^0} - \overline{V}_t^-) - (V^{\pi_t^1} - \widetilde{V}_t^-) \right] \right) \right)$$

$$\leq \mathrm{F}^{-2}(-1) \left( 8 V_{\max} \delta + \sum_{t=1}^{T} \left( (\overline{V}_t - V^{\pi_t^0}) - (\widetilde{V}_t - V^{\pi_t^1}) + (V^{\pi_t^0} - \overline{V}_t^-) - (V^{\pi_t^1} - \widetilde{V}_t^-) \right) \right.$$

$$\left. + 2 V_{\max} \sqrt{\frac{T \log(1/\delta)}{2}} \right).$$

where the last inequality is the Hoeffding's inequality. $\qquad \square$

Given all these lemmas, we are ready to establish an upper bound of regret. We first note that, since $\pi_t^1 = \pi_{t-1}^0$ for all $t$, the regret incurred by $\pi_t^1$ for all $t$ is equivalent to that incurred by $\pi_t^0$ for all $t$. Hence, it suffices to compute the regret incurred by $\pi_t^0$ for $t \in [T]$ and multiply it by two to get the total regret.

We start with the following regret decomposition:

$$\mathrm{Regret}_T \leq \sum_{t=1}^{T} \left( V^\star - V^{\pi_t^0} \right) = \underbrace{\sum_{t=1}^{T} \left( V^\star - \overline{V}_t + \widetilde{V}_t - V^{\pi_t^1} \right)}_{(*)} + \sum_{t=1}^{T} \left( \overline{V}_t - V^{\pi_t^0} + V^{\pi_t^1} - \widetilde{V}_t \right).$$

By Lemma B.16, we can further decompose $(*)$ and obtain

$$\text{Regret}_T \leq \frac{1}{\text{F}^2(-1)} \underbrace{\sum_{t=1}^{T} \left( V^{\pi_t^0} - \overline{V}_t^- + \widetilde{V}_t^- - V_t^{\pi_t^1} \right)}_{\text{(i)}} + \left( 1 + \frac{1}{\text{F}^2(-1)} \right) \underbrace{\sum_{t=1}^{T} \left( \overline{V}_t - V^{\pi_t^0} + V^{\pi_t^1} - \widetilde{V}_t \right)}_{\text{(ii)}}$$

$$+ \frac{2V_{\max}}{\text{F}^2(-1)} \left( \sqrt{\frac{T\log(1/\delta)}{2}} + 4\delta \right).$$

We note that both (i) and (ii) can be bounded by Lemma B.15:

$$\text{(i), (ii)} \leq \widetilde{O}\Bigg( V_{\max}L_{\max} + V_{\max}\sqrt{HT}$$

$$+ (\epsilon_{\text{r},\xi} + \epsilon_{\text{r},\eta}) \left( \sqrt{dT} + L_{\max} + \sqrt{HT} + \frac{\epsilon T\sqrt{\pi}}{2\sigma_{\text{r}}} \right)$$

$$+ (\epsilon_{\text{P},\xi} + \epsilon_{\text{P},\eta} + \epsilon_\lambda) \left( H\sqrt{dT} + L_{\max} + \sqrt{HT} \right) \Bigg)$$

Inserting this back, we obtain

$$\text{Regret}_T = \widetilde{O}\Bigg( V_{\max}L_{\max} + V_{\max}\sqrt{HT}$$

$$+ (\epsilon_{\text{r},\xi} + \epsilon_{\text{r},\eta}) \left( \sqrt{dT} + L_{\max} + \sqrt{HT} + \frac{\epsilon T\sqrt{\pi}}{2\sigma_{\text{r}}} \right)$$

$$+ (\epsilon_{\text{P},\xi} + \epsilon_{\text{P},\eta} + \epsilon_\lambda) \left( H\sqrt{dT} + L_{\max} + \sqrt{HT} \right)$$

$$+ V_{\max}\sqrt{T} \Bigg)$$

To compute this quantity, we note the following asymptotic rate:

- $\epsilon_{\text{r},\xi} + \epsilon_{\text{r},\eta} = \widetilde{O}\left( d\sqrt{\kappa + B^2} \right)$
- $\epsilon_{\text{P},\xi} + \epsilon_{\text{P},\eta} + \epsilon_\lambda = \widetilde{O}\left( d^{5/2}H^{3/2}\sqrt{\kappa + B^2} \right)$
- $L_{\max} = \widetilde{O}\left( d^6 H^4(\kappa + B^2) \right)$

Hence, we have

$$\text{Regret}_T = \widetilde{O}\left( \epsilon T\sqrt{d} + \sqrt{T} \cdot d^3 H^{5/2}\sqrt{\kappa + B^2} + d^{17/2}H^{11/2}(\kappa + B^2)^{3/2} \right).$$

## B.5 BOUNDING NUMBER OF QUERIES

Recall that the number of queries are computed via

$$\text{Queries}_T = \sum_{t=1}^{T} Z_t = \sum_{t=1}^{T} Z_t \mathbb{1}\left\{ \mathbb{E}_{\theta_0,\theta_1 \sim \mathcal{N}(\hat{\theta}_{\text{r},t}, \sigma_{\text{r}}^2\Sigma_{t-1}^{-1})} \left[ \left| (\phi(\tau_t^0) - \phi(\tau_t^1))^\top (\theta_0 - \theta_1) \right| \right] > \epsilon \right\}.$$

Notice that Cauchy-Schwarz inequality implies

$$\mathbb{E}_{\theta_0,\theta_1} \left| (\phi(\tau_t^0) - \phi(\tau_t^1))^\top (\theta_0 - \theta_1) \right| \leq \|\phi(\tau_t^0) - \phi(\tau_t^1)\|_{\Sigma_{t-1}^{-1}} \mathbb{E}_{\theta_0,\theta_1} \|\theta_0 - \theta_1\|_{\Sigma_{t-1}}$$

$$\leq \frac{1}{\sqrt{\lambda}}\|\phi(\tau_t^0) - \phi(\tau_t^1)\|_2 \mathbb{E}_{\theta_0,\theta_1} \|\theta_0 - \theta_1\|_{\Sigma_{t-1}} \leq \frac{2}{\sqrt{\lambda}} \mathbb{E}_{\theta_0,\theta_1} \|\theta_0 - \theta_1\|_{\Sigma_{t-1}} \leq \frac{2\sqrt{2\sigma_{\text{r}}^2 d}}{\sqrt{\lambda}}$$

where the second inequality is by $\|\Sigma_{t-1}^{-1}\|_2 = 1/\lambda$, and the last step is by the fact that $\theta_0 - \theta_1$ follows $\mathcal{N}(0, 2\sigma_{\text{r}}^2\Sigma_{t-1}^{-1})$ and Lemma D.3. We denote $\zeta := 2\sigma_{\text{r}}\sqrt{2d/\lambda}$ for the ease of notation. Then, we

have

$$\text{Queries}_T = \sum_{t=1}^{T} Z_t \mathbb{1}\left\{\min\left\{\zeta, \mathop{\mathbb{E}}_{\theta_0,\theta_1}\left[\left|(\phi(\tau_t^0) - \phi(\tau_t^1))^\top(\theta_0 - \theta_1)\right|\right]\right\} > \epsilon\right\}$$

$$\leq \sum_{t=1}^{T} Z_t \mathbb{1}\left\{\min\left\{1, \mathop{\mathbb{E}}_{\theta_0,\theta_1}\left[\left|(\phi(\tau_t^0) - \phi(\tau_t^1))^\top(\theta_0 - \theta_1)\right|\right]\right\} > \epsilon/\zeta\right\}$$

Applying Cauchy-Schwarz inequality, we have

$$\text{Queries}_T \leq \sum_{t=1}^{T} Z_t \mathbb{1}\left\{\min\left\{1, \mathop{\mathbb{E}}_{\theta_0,\theta_1}\left[\left((\phi(\tau_t^0) - \phi(\tau_t^1))^\top(\theta_0 - \theta_1)\right)^2\right]\right\} > \epsilon^2/\zeta^2\right\}.$$

Let $u_0, u_1$ denotes two independent standard Gaussian variables with zero mean and identity covariance matrix. Then, $\theta_0 - \theta_1$ has the same joint distribution as $\sigma_r \Sigma_{t-1}^{-1/2}(u_0 - u_1)$. Hence, we can rewrite the expectation in the indicator as

$$\mathop{\mathbb{E}}_{\theta_0,\theta_1}\left[\left((\phi(\tau_t^0) - \phi(\tau_t^1))^\top(\theta_0 - \theta_1)\right)^2\right] = \mathop{\mathbb{E}}_{u_0,u_1}\left[\left((\phi(\tau_t^0) - \phi(\tau_t^1))^\top \sigma_r \Sigma_{t-1}^{-1/2}(u_0 - u_1)\right)^2\right]$$

Furthermore, we have

$$\mathop{\mathbb{E}}_{u_0,u_1}\left[\left((\phi(\tau_t^0) - \phi(\tau_t^1))^\top \sigma_r \Sigma_{t-1}^{-1/2}(u_0 - u_1)\right)^2\right]$$

$$= \mathop{\mathbb{E}}_{u_0,u_1}\left[(\phi(\tau_t^0) - \phi(\tau_t^1))^\top \sigma_r \Sigma_{t-1}^{-1/2}(u_0 - u_1)(u_0 - u_1)^\top \sigma_r \Sigma_{t-1}^{-1/2}(\phi(\tau_t^0) - \phi(\tau_t^1))\right]$$

$$= (\phi(\tau_t^0) - \phi(\tau_t^1))^\top \sigma_r \Sigma_{t-1}^{-1/2} \mathop{\mathbb{E}}_{u_0,u_1}\left[(u_0 - u_1)(u_0 - u_1)^\top\right] \sigma_r \Sigma_{t-1}^{-1/2}(\phi(\tau_t^0) - \phi(\tau_t^1)).$$

For the expectation in the middle, we have

$$\mathop{\mathbb{E}}_{u_0,u_1}\left[(u_0 - u_1)(u_0 - u_1)^\top\right] = \mathbb{E}[u_0 u_0^\top] + \mathbb{E}[u_1 u_1^\top] = 2I$$

where we have used the fact that $\mathbb{E}[u_0 u_1^\top] = 0$ by independence. Therefore, we have

$$\mathop{\mathbb{E}}_{\theta_0,\theta_1}\left[\left((\phi(\tau_t^0) - \phi(\tau_t^1))^\top(\theta_0 - \theta_1)\right)^2\right]$$

$$= (\phi(\tau_t^0) - \phi(\tau_t^1))^\top \sigma_r \Sigma_{t-1}^{-1/2}(2I)\sigma_r \Sigma_{t-1}^{-1/2}(\phi(\tau_t^0) - \phi(\tau_t^1))$$

$$= 2\sigma_r^2 \left\|\phi(\tau_t^0) - \phi(\tau_t^1)\right\|_{\Sigma_{t-1}^{-1}}^2.$$

Inserting this back, we obtain

$$\text{Queries}_T \leq \sum_{t=1}^{T} Z_t \mathbb{1}\left\{\min\left\{1, 2\sigma_r^2 \left\|\phi(\tau_t^0) - \phi(\tau_t^1)\right\|_{\Sigma_{t-1}^{-1}}^2\right\} > \epsilon^2/\zeta^2\right\}$$

$$\leq \sum_{t=1}^{T} Z_t \mathbb{1}\left\{\min\left\{1, \left\|\phi(\tau_t^0) - \phi(\tau_t^1)\right\|_{\Sigma_{t-1}^{-1}}^2\right\} > \frac{\epsilon^2}{2\zeta^2 \sigma_r^2}\right\}$$

$$\leq \frac{2\zeta^2 \sigma_r^2}{\epsilon^2} \sum_{t=1}^{T} Z_t \min\left\{1, \left\|\phi(\tau_t^0) - \phi(\tau_t^1)\right\|_{\Sigma_{t-1}^{-1}}^2\right\}$$

$$\leq \frac{2\zeta^2 \sigma_r^2}{\epsilon^2} \cdot 2d \log\left(\frac{\lambda + 4T}{\lambda}\right)$$

where the last step is Lemma D.6. Plugging the value of $\zeta$ and all other variables, we obtain

$$\text{Queries}_T \leq \frac{32\sigma_r^4 d^2}{\lambda \epsilon^2} \log\left(\frac{\lambda + 4T}{\lambda}\right) = \widetilde{O}\left(\frac{d^4(\kappa + B^2)^2}{\epsilon^2}\right).$$

## C  PROOF OF THEOREM 5.4

We first present some supporting results in Appendix C.1. Then, we prove the upper bound of Bayesian regret in Appendix C.2 and the number of queries in Appendix C.3.

### C.1  SUPPORTING LEMMAS

In Appendix C.1.1, we establish some supporting results for a probability estimation problem in the frequentist setting. In Appendix C.1.2, we adapt these results to the Bayesian setting. The Bayesian results will be heavily used later in the proof of Theorem 5.4 in Appendices C.2 and C.3.

#### C.1.1  SUPPORTING RESULTS FROM FREQUENTIST SETTING

Consider a conditional probability estimation problem. Let $\mathcal{X}$ and $\mathcal{Y}$ be the instance space and the target space, respectively. Let $\mathcal{F} : (\mathcal{X} \times \mathcal{Y}) \to \mathbb{R}$ be a function class. We are given a dataset $D := \{(x_i, y_i)\}_{i=1}^{n}$ where $x_i \sim \mathcal{D}_i$ and $y_i \sim f^\star(x, \cdot)$. We assume $f^\star \in \mathcal{F}$. Regarding the data generation process, we assume the data distribution $\mathcal{D}_i$ is history-dependent, i.e., $x_i$ can depend on the previous samples: $x_1, y_1, \ldots, x_{i-1}, y_{i-1}$ for any $i \in [n]$. Our goal is to estimate the true conditional probability $f^\star$ using the dataset $D$.

At a high level, this problem is designed to capture both the reward learning and the model learning problems in the RL setting. Specifically, in the reward learning problem, we will instantiate $\mathcal{X} = \mathcal{S} \times \mathcal{A}$ and $\mathcal{Y} = \{0, 1\}$, where we recall that $\mathcal{S}$ and $\mathcal{A}$ are the state space and the action space, respectively, and the preference feedback is binary. In the model learning problem, we can instantiate $\mathcal{X} = \mathcal{S} \times \mathcal{A}$ and $\mathcal{Y} = \mathcal{S}$. We abstract these two problems into this conditional probability estimation problem to make the analysis more concise. However, one caveat is that all we derived in this section are *frequentist* results, while we are considering *Bayesian* RL. Thus, the results are not directly applicable. In Appendix C.1.2, we will adapt these frequentist results to the Bayesian setting so that they can be applied.

Now we establish some important results for this problem. First, we have the following lemma, which is a consequence of Lemma D.10.

**Lemma C.1** (MLE generalization bound). *Fix $\delta \in (0, 1)$. Follow the setting stated above. Let $\widehat{f}$ be the maximum likelihood estimator:*

$$\widehat{f} = \arg\max_{f \in \mathcal{F}} \sum_{i=1}^{n} \log f(x_i, y_i).$$

*Define the version space:*

$$\mathcal{V}^{\mathcal{F}} = \left\{ f \in \mathcal{F} \ : \ \sum_{i=1}^{n} d_{\mathrm{TV}}^2 \left( \widehat{f}(x_i, \cdot), f(x_i, \cdot) \right) \leq \beta_{\mathcal{F}}(n) \right\}$$

*where $\beta_{\mathcal{F}}(n) := 98 \log(2 N_{[]}((n|\mathcal{Y}|)^{-1}, \mathcal{F}, \|\cdot\|_\infty)/\delta)$. Then, the following holds*

*(i) $f^\star \in \mathcal{V}^{\mathcal{F}}$ with probability at least $1 - \delta$,*

*(ii) for any $f, f' \in \mathcal{V}^{\mathcal{F}}$, it holds that*

$$\sum_{i=1}^{n} d_{\mathrm{TV}}^2 \left( f(x_i, \cdot), f'(x_i, \cdot) \right) \leq 4\beta_{\mathcal{F}}(n).$$

*Proof of Lemma C.1.* We first construct an auxiliary version space $\widetilde{\mathcal{V}}$ as follows

$$\widetilde{\mathcal{V}}^{\mathcal{F}} = \left\{ f \in \mathcal{F} \ : \ \sum_{i=1}^{n} \mathbb{E}_{x \sim \mathcal{D}_i} d_{\mathrm{TV}}^2 \left( \widehat{f}(x, \cdot), f(x, \cdot) \right) \leq 10 \log \left( N_{[]} \left( (n|\mathcal{Y}|)^{-1}, \mathcal{F}, \|\cdot\|_\infty \right)/\delta \right) \right\}$$

By Lemma D.10, $f^\star \in \widetilde{\mathcal{V}}^{\mathcal{F}}$ with probability at least $1 - \delta$. To prove (i), we will show that whenever $f^\star \in \widetilde{\mathcal{V}}^{\mathcal{F}}$, we have $f^\star \in \mathcal{V}^{\mathcal{F}}$ as well with high probability. Let $\mathcal{F}_{[]}$ denote an $(n|\mathcal{Y}|)^{-1}$-bracket of $\mathcal{F}$

with respect to $\| \cdot \|_\infty$. Then for all $f_{[]} \in \mathcal{F}_{[]}$, the following holds with probability at least $1 - \delta$ by Lemma D.4 and the union bound on $\mathcal{F}_{[]}$,

$$\sum_{i=1}^n d_{\mathrm{TV}}^2 \left( f_{[]}(x_i, \cdot), f^\star(x_i, \cdot) \right)$$

$$\leq 2 \sum_{i=1}^n \mathbb{E}_{x \sim \mathcal{D}_i} d_{\mathrm{TV}}^2 \left( f_{[]}(x, \cdot), f^\star(x, \cdot) \right) + 4 \log \left( N_{[]} \left( (n|\mathcal{Y}|)^{-1}, \mathcal{F}, \| \cdot \|_\infty \right) / \delta \right). \quad (12)$$

Now for any $f \in \mathcal{F}$, there must exist $f_{[]} \in \mathcal{F}_{[]}$ such that $\|f - f_{[]}\|_\infty \leq (n|\mathcal{Y}|)^{-1}$, which impies the following

$$\sum_{i=1}^n d_{\mathrm{TV}}^2 \left( f(x_i, \cdot), f_{[]}(x_i, \cdot) \right) \leq \sum_{i=1}^n |\mathcal{Y}|^2 \|f - f_{[]}\|_\infty^2 \leq \sum_{i=1}^n n^{-2} = 1/n, \quad (13)$$

$$\sum_{i=1}^n \mathbb{E}_{x \sim \mathcal{D}_i} d_{\mathrm{TV}}^2 \left( f(x, \cdot), f_{[]}(x, \cdot) \right) \leq \sum_{i=1}^n |\mathcal{Y}|^2 \|f - f_{[]}\|_\infty^2 \leq \sum_{i=1}^n n^{-2} = 1/n \quad (14)$$

Hence, for all $f \in \mathcal{F}$, we have

$$\sum_{i=1}^n d_{\mathrm{TV}}^2 \left( f(x_i, \cdot), f^\star(x_i, \cdot) \right)$$

$$\leq 2 \sum_{i=1}^n d_{\mathrm{TV}}^2 \left( f_{[]}(x_i, \cdot), f^\star(x_i, \cdot) \right) + 2 \sum_{i=1}^n d_{\mathrm{TV}}^2 \left( f(x_i, \cdot), f_{[]}(x_i, \cdot) \right)$$

$$\leq 2 \sum_{i=1}^n d_{\mathrm{TV}}^2 \left( f_{[]}(x_i, \cdot), f^\star(x_i, \cdot) \right) + 2/n$$

$$\leq 4 \sum_{i=1}^n \mathbb{E}_{x \sim \mathcal{D}_i} d_{\mathrm{TV}}^2 \left( f_{[]}(x, \cdot), f^\star(x, \cdot) \right) + 8 \log \left( N_{[]} \left( (n|\mathcal{Y}|)^{-1}, \mathcal{F}, \| \cdot \|_\infty \right) / \delta \right) + 2/n$$

$$\leq 8 \sum_{i=1}^n \mathbb{E}_{x \sim \mathcal{D}_i} d_{\mathrm{TV}}^2 \left( f(x, \cdot), f^\star(x, \cdot) \right) + 8 \log \left( N_{[]} \left( (n|\mathcal{Y}|)^{-1}, \mathcal{F}, \| \cdot \|_\infty \right) / \delta \right) + 10/n.$$

with probability at least $1 - \delta$. Here the second inequality uses (13), the third uses (12), and the last uses (14). Therefore, for any possible value of the estimator $\widehat{f}$, conditioning on $f^\star \in \widetilde{\mathcal{V}}^{\mathcal{F}}$, we have

$$\sum_{i=1}^n d_{\mathrm{TV}}^2 \left( \widehat{f}(x_i, \cdot), f^\star(x_i, \cdot) \right)$$

$$\leq 8 \sum_{i=1}^n \mathbb{E}_{x \sim \mathcal{D}_i} d_{\mathrm{TV}}^2 \left( \widehat{f}(x, \cdot), f^\star(x, \cdot) \right) + 8 \log \left( N_{[]} \left( (n|\mathcal{Y}|)^{-1}, \mathcal{F}, \| \cdot \|_\infty \right) / \delta \right) + 10/n$$

$$\leq 88 \log \left( N_{[]} \left( (n|\mathcal{Y}|)^{-1}, \mathcal{F}, \| \cdot \|_\infty \right) / \delta \right) + 10/n$$

$$\leq 98 \log \left( N_{[]} \left( (n|\mathcal{Y}|)^{-1}, \mathcal{F}, \| \cdot \|_\infty \right) / \delta \right)$$

with probability at least $1 - \delta$. Here the second inequality is by the definition of $\widetilde{\mathcal{V}}^{\mathcal{F}}$. The last inequality holds since it is reasonable to assume that the bracketing number is at least some constant so that $\log(N_{[]}((n|\mathcal{Y}|)^{-1}, \mathcal{F}, \| \cdot \|_\infty)/\delta) > 1 \geq 1/n$.

The above means whenever $f^\star \in \widetilde{\mathcal{V}}^{\mathcal{F}}$, we have $f^\star \in \mathcal{V}^{\mathcal{F}}$ as well with probability at least $1 - \delta$. Since $f^\star \in \widetilde{\mathcal{V}}^{\mathcal{F}}$ with probability at least $1 - \delta$, we have $f^\star \in \mathcal{V}^{\mathcal{F}}$ with probability at least $1 - 2\delta$. Adjusting $\delta$ completes the proof of (i).

For (ii), we have

$$\sum_{i=1}^{n} d_{\mathrm{TV}}^2\Big(f(x,\cdot), f'(x,\cdot)\Big) \le \sum_{i=1}^{n} \Big(d_{\mathrm{TV}}\Big(f(x,\cdot), \widehat{f}(x,\cdot)\Big) + d_{\mathrm{TV}}\Big(\widehat{f}(x,\cdot), f'(x,\cdot)\Big)\Big)^2$$

$$\le \sum_{i=1}^{n} 2d_{\mathrm{TV}}^2\Big(f(x,\cdot), \widehat{f}(x,\cdot)\Big) + \sum_{i=1}^{n} 2d_{\mathrm{TV}}^2\Big(\widehat{f}(x,\cdot), f'(x,\cdot)\Big)$$

$$\le 4\beta_{\mathcal{F}}(n)$$

where the second inequality is by the fact that $(a+b)^2 \le 2a^2 + 2b^2$ for any $a$ and $b$, and the third inequality is by the definition of $\mathcal{V}^{\mathcal{F}}$. $\qquad\square$

Next, we further assume that the function class $\mathcal{F}$ is parameterized by a function class $\mathcal{G}$ via a link function $\Phi$. The reason for this assumption is that we want to capture the structure of the reward learning problem, where the feedback generating distribution is parameterized by the reward function via a link function $\Phi$.

**Assumption C.2** (Binary label and function parameterization.). *Assume $\mathcal{Y} = \{0,1\}$ is binary, and there is a function class $\mathcal{G} \subseteq \mathcal{X} \to [0, G]$ that parameterizes $\mathcal{F}$ via a link function $\Phi$. Specifically, we assume*

$$\mathcal{F} = \Big\{ f(x,0) = \Phi\big(g(x)\big),\ f(x,1) = 1 - \Phi\big(g(x)\big)\ :\ g \in \mathcal{G} \Big\},$$

*where we further assume $\Phi$ satisfies Assumption 3.1. For any $f \in \mathcal{F}$, let $g_f$ denote the function $g$ that parameterizes $f$. We define $\widehat{g} := g_{\widehat{f}}$ and $g^\star := g_{f^\star}$.*

As a preliminary note, the function class $\mathcal{G}$ will actually correspond to the function class $\widetilde{\mathcal{R}}$ (3) later in the proof.

We should emphasize again that, although the function class $\mathcal{G}$ aims to capture the structure of the reward learning problem, the function class $\mathcal{F}$ will be used to capture both the reward learning and the model learning problems. When it is applied to the model learning problem, we can simply ignore the function class $\mathcal{G}$ and any results that are related to $\mathcal{G}$. On the other hand, when it is applied to the reward learning problem, such a function class $\mathcal{G}$ will be helpful to derive some results in the reward function itself.

Before diving into the analysis of the function class $\mathcal{G}$, we first introduce the following lemma which shows that the bracketing numbers of $\mathcal{F}$ and $\mathcal{G}$ are bounded by each other.

**Lemma C.3.** *Under Assumption C.2, for any $\omega > 0$, we have*

$$N_{[]}\Big(\omega, \mathcal{F}, \|\cdot\|_\infty\Big) \le N_{[]}\Big(\overline{\kappa}\omega, \mathcal{G}, \|\cdot\|_\infty\Big) \qquad and \qquad N_{[]}\Big(\kappa\omega, \mathcal{G}, \|\cdot\|_\infty\Big) \le N_{[]}\Big(\omega, \mathcal{F}, \|\cdot\|_\infty\Big).$$

*Proof.* For any $f, f' \in \mathcal{F}$, we assume they are parameterized by $g := g_f$ and $g' := g_{f'}$, respectively. Then, we have

$$\sup_{x,y} |f(x,y) - f'(x,y))| = \sup_{x} \big|\Phi\big(g(x)\big) - \Phi\big(g'(x)\big)\big| \le \overline{\kappa}^{-1} \sup_{x} |g(x) - g'(x)|$$

where the inequality is Lemma D.1. Hence, if we have a $\overline{\kappa}\omega$-bracket of $\mathcal{G}$ in the infinite norm, then we have an $\omega$-bracket of $\mathcal{F}$ in the infinite norm. This proves the first claim. For the second claim, by Lemma D.1, we have

$$\sup_{x} |g(x) - g'(x)| \le \kappa \sup_{x} \big|\Phi\big(g(x)\big) - \Phi\big(g'(x)\big)\big| = \kappa \sup_{x,y} |f(x,y) - f'(x,y)|.$$

This proves the second claim. $\qquad\square$

Then, Lemma C.1 leads to similar results with a version space constructed on $\mathcal{G}$.

**Corollary C.4.** *Under Assumption C.2, define*

$$\mathcal{V}^{\mathcal{G}} = \left\{ g \in \mathcal{G}\ :\ \sum_{i=1}^{n} \Big(\widehat{g}(x_i) - g(x_i)\Big)^2 \le \beta_{\mathcal{G}}(n) \right\}$$

*where $\beta_{\mathcal{G}}(n) := 98\kappa^2 \log(2N_{[]}(\overline{\kappa}(n|\mathcal{Y}|)^{-1}, \mathcal{G}, \|\cdot\|_\infty)/\delta)$ and we denote $\widehat{g} := g_{\widehat{f}}$ as the function that parameterizes the maximum likelihood estimator $\widehat{f}$. Then, the following holds*

*(1)* $g^\star \in \mathcal{V}^{\mathcal{G}}$ *with probability at least* $1 - \delta$

*(2) for any* $g, g' \in \mathcal{V}^{\mathcal{G}}$, *we have*

$$\sum_{i=1}^{n} \Big( g(x_i) - g'(x_i) \Big)^2 \leq 4\beta_{\mathcal{G}}(n).$$

*Proof of Corollary C.4.* To prove (i), we claim that for any $f \in \mathcal{V}^{\mathcal{F}}$ (defined in Lemma C.1), we have $g_f \in \mathcal{V}^{\mathcal{G}}$ as well. To see this, we note that

$$\sum_{i=1}^{n} \mathop{\mathbb{E}}_{x \sim \mathcal{D}_i} \Big( \widehat{g}(x) - g_f(x) \Big)^2 \leq \kappa^2 \sum_{i=1}^{n} \mathop{\mathbb{E}}_{x \sim \mathcal{D}_i} \big| \Phi(\widehat{g}(x)) - \Phi(g_f(x)) \big|^2$$

$$= \kappa^2 \sum_{i=1}^{n} \mathop{\mathbb{E}}_{x \sim \mathcal{D}_i} \big| \widehat{f}(x, 0) - f(x, 0) \big|^2 = \kappa^2 \sum_{i=1}^{n} \mathop{\mathbb{E}}_{x \sim \mathcal{D}_i} d_{\mathrm{TV}}^2 \Big( \widehat{f}(x, \cdot), f(x, \cdot) \Big)$$

where the inequality is Lemma D.1, and the last equality holds since we assume $\mathcal{Y} = \{0, 1\}$ is binary. Hence, for any $f \in \mathcal{V}^{\mathcal{F}}$, it holds that

$$\sum_{i=1}^{n} \mathop{\mathbb{E}}_{x \sim \mathcal{D}_i} \Big( \widehat{g}(x) - g_f(x) \Big)^2 \leq \kappa^2 \sum_{i=1}^{n} \mathop{\mathbb{E}}_{x \sim \mathcal{D}_i} d_{\mathrm{TV}}^2 \Big( \widehat{f}(x, \cdot), f(x, \cdot) \Big)$$

$$\leq \kappa^2 \beta_{\mathcal{F}}(n)$$

$$= \kappa^2 \cdot 98 \log(2N_{[]}((n|\mathcal{Y}|)^{-1}, \mathcal{F}, \| \cdot \|_\infty)/\delta)$$

By Lemma C.3, we have

$$\log(2N_{[]}((n|\mathcal{Y}|)^{-1}, \mathcal{F}, \| \cdot \|_\infty)/\delta) \leq \log(2N_{[]}(\overline{\kappa}(n|\mathcal{Y}|)^{-1}, \mathcal{G}, \| \cdot \|_\infty)/\delta).$$

Hence, we have

$$\sum_{i=1}^{n} \mathop{\mathbb{E}}_{x \sim \mathcal{D}_i} \Big( \widehat{g}(x) - g_f(x) \Big)^2 \leq \beta_{\mathcal{G}}(n)$$

which implies $g_f \in \mathcal{V}^{\mathcal{G}}$. Therefore, whenever $f^\star \in \mathcal{V}^{\mathcal{F}}$, we have $g_{f^\star} \in \mathcal{V}^{\mathcal{G}}$ as well. Since $f^\star \in \mathcal{V}^{\mathcal{F}}$ with probability at least $1 - \delta$, we have $g_{f^\star} \in \mathcal{V}^{\mathcal{G}}$ with probability at least $1 - \delta$. This completes the proof of (i).

Now we prove (2). For any $g, g' \in \mathcal{V}^{\mathcal{G}}$, we have

$$\sum_{i=1}^{n} \Big( g(x_i) - g'(x_i) \Big)^2 \leq 2 \sum_{i=1}^{n} \Big( \widehat{g}(x_i) - g(x_i) \Big)^2 + 2 \sum_{i=1}^{n} \Big( \widehat{g}(x_i) - g'(x_i) \Big)^2 \leq 4\beta_{\mathcal{G}}(n)$$

where the first inequality is by the fact that $(a + b)^2 \leq 2a^2 + 2b^2$ for any $a$ and $b$. $\qquad\square$

### C.1.2 ADAPTING RESULTS INTO BAYESIAN SETTING

Now we change the setting defined in Appendix C.1.1 into a Bayesian online setting. The reason for this adaptation is to make these results applicable to the Bayesian RL setting. We formally define the Bayesian online conditional probability estimation problem below.

Let $\mathcal{X}$ and $\mathcal{Y}$ be the instance space and the target space, respectively. Let $\mathcal{F} : (\mathcal{X} \times \mathcal{Y}) \to \mathbb{R}$ be a function class. The interaction proceeds for $T$ rounds. At each round $t \in [T]$, we observe an instance $x_t \in \mathcal{X}$ and need to outputs a function $f_t \in \mathcal{F}$ as our prediction. Then, the label $y_t \in \mathcal{Y}$ is revealed. We assume that $x_i \sim \mathcal{D}_i$ for some distribution $\mathcal{D}_i$ and $y_i \sim f^\star(x, \cdot)$. We assume the true conditional distribution $f^\star$ is sampled from some known prior distribution $\rho \in \Delta(\mathcal{F})$ at the beginning. Regarding the data generation process, we assume the data distribution $\mathcal{D}_i$ is history-dependent, i.e., $x_i$ can depend on the previous samples: $x_1, y_1, \ldots, x_{i-1}, y_{i-1}$ for any $i \in [n]$.

Denote $\mathcal{H}_t$ as the history up to round $t$, i.e., $\mathcal{H}_t = \{x_1, f_1, y_1, x_2, f_2, y_2, \ldots, x_t, f_t, y_t\}$. Define the maximum likelihood estimator of $f^\star$ on the dataset $\{(x_s, y_s)\}_{s=1}^{t-1}$ as $\widehat{f}_t$, i.e.,

$$\widehat{f}_t = \arg\max_{f \in \mathcal{F}} \sum_{s=1}^{t-1} \log f(x_s, y_s).$$

Similar to the previous section, we will also consider the presence of a function class $\mathcal{G}$ (Assumption C.2) in the analysis presented in this section. It's worth noting that although Assumption C.2 is initially introduced in the frequentist setting, when we mention that Assumption C.2 holds in this section, we are actually considering the Bayesian setting.

**Lemma C.5.** *It holds that*

$$\sum_{t=1}^{T} \mathop{\mathbb{E}}_{\mathcal{H}_{t-1}} \left[ \mathop{\mathbb{E}}_{f,f'} \left[ d_{\mathrm{TV}}\big(f(x_t), f'(x_t)\big) \,\Big|\, \mathcal{H}_{t-1} \right] \right] \le 6\sqrt{T\beta'_{\mathcal{F}}(T)} \cdot \dim_1\Big(\mathcal{F}, 1/T\Big) \cdot \log(T) \quad (15)$$

*and*

$$\sum_{t=1}^{T} \mathop{\mathbb{E}}_{\mathcal{H}_{t-1}} \left[ \mathop{\mathbb{E}}_{f,f'} \left[ d_{\mathrm{TV}}^2\big(f(x_t), f'(x_t)\big) \,\Big|\, \mathcal{H}_{t-1} \right] \right] \le 4\beta'_{\mathcal{F}}(T) + 2 \quad (16)$$

*where $\beta'_{\mathcal{F}}(t) := 98\log(2TN_{[]}((t|\mathcal{Y}|)^{-1}, \mathcal{F}, \|\cdot\|_{\infty}))$, and $f, f'$ are sampled from the posterior of $f^{\star}$ conditioning on $\mathcal{H}_{t-1}$ in the inner conditional expectation. Moreover, if $\mathcal{F}$ is parameterized by $\mathcal{G}$ (i.e., if Assumption C.2 holds), we have*

$$\sum_{t=1}^{T} \mathop{\mathbb{E}}_{\mathcal{H}_{t-1}} \left[ \mathop{\mathbb{E}}_{g,g'} \left[ \big|g(x_t) - g'(x_t)\big| \,\Big|\, \mathcal{H}_{t-1} \right] \right] \le 6\sqrt{T\beta'_{\mathcal{G}}(T)} \cdot \dim_1\Big(\mathcal{G}, 1/T\Big) \cdot \log(GT) \quad (17)$$

*and*

$$\sum_{t=1}^{T} \mathop{\mathbb{E}}_{\mathcal{H}_{t-1}} \left[ \mathop{\mathbb{E}}_{g,g'} \left[ \big(g(x_t) - g'(x_t)\big)^2 \,\Big|\, \mathcal{H}_{t-1} \right] \right] \le 4\beta'_{\mathcal{G}}(T) + 2 \quad (18)$$

*where $\beta'_{\mathcal{G}}(t) := 98\kappa^2 \log(2GTN_{[]}(\overline{\kappa}(t|\mathcal{Y}|)^{-1}, \mathcal{G}, \|\cdot\|_{\infty}))$, and $g, g'$ are sampled from the posterior of $g^{\star}$ conditioning on $\mathcal{H}_{t-1}$ in the inner conditional expectation.*

*Proof of Lemma C.5.* We will only prove (15) and (16) since the proof of (17) and (18) is almost identical.

**Proof of** (15). We define the version space at round $t$ for $t \in [T]$ as follows

$$\mathcal{V}_t^{\mathcal{F}} = \left\{ f \in \mathcal{F} \, : \, \sum_{s=1}^{t-1} d_{\mathrm{TV}}^2\left(\widehat{f}_t(x_s, \cdot), f(x_s, \cdot)\right) \le \beta_{\mathcal{F}}(t-1) \right\}$$

where $\widehat{f}_t$ is the maximum likelihood estimator on the dataset $\{(x_s, y_s)\}_{s=1}^{t-1}$. This construction aims to mimic the version space $\mathcal{V}^{\mathcal{F}}$ defined in Lemma C.1, which enable us to apply Lemma C.1.

We first split the left side of (15) into two cases based on whether $f$ and $f'$ belong to $\mathcal{V}_t^{\mathcal{F}}$:

$$\sum_{t=1}^{T} \mathop{\mathbb{E}}_{\mathcal{H}_{t-1}} \left[ \mathop{\mathbb{E}}_{f,f'} \left[ d_{\mathrm{TV}}\big(f(x_t), f'(x_t)\big) \,\Big|\, \mathcal{H}_{t-1} \right] \right]$$

$$= \sum_{t=1}^{T} \mathop{\mathbb{E}}_{\mathcal{H}_{t-1}} \left[ \mathop{\mathbb{E}}_{f,f'} \left[ \mathbb{1}\big\{f \in \mathcal{V}_t^{\mathcal{F}} \text{ and } f' \in \mathcal{V}_t^{\mathcal{F}}\big\} d_{\mathrm{TV}}\big(f(x_t), f'(x_t)\big) \,\Big|\, \mathcal{H}_{t-1} \right] \right.$$

$$\left. + \mathop{\mathbb{E}}_{f,f'} \left[ \mathbb{1}\big\{f \notin \mathcal{V}_t^{\mathcal{F}} \text{ or } f' \notin \mathcal{V}_t^{\mathcal{F}}\big\} d_{\mathrm{TV}}\big(f(x_t), f'(x_t)\big) \,\Big|\, \mathcal{H}_{t-1} \right] \right]$$

$$=: \mathsf{T}_1 + \mathsf{T}_2.$$

To bound $T_2$, since the total variation distance is upper bounded by 1, we have

$$
\begin{aligned}
T_2 &\leq \sum_{t=1}^{T} \underset{\mathcal{H}_{t-1}}{\mathbb{E}} \left[ \underset{f,f'}{\mathbb{E}} \left[ \mathbb{1}\left\{ f \notin \mathcal{V}_t^{\mathcal{F}} \text{ or } f' \notin \mathcal{V}_t^{\mathcal{F}} \right\} \Big| \mathcal{H}_{t-1} \right] \right] \\
&= \sum_{t=1}^{T} \underset{\mathcal{H}_{t-1}}{\mathbb{E}} \left[ \underset{f,f'}{\mathbb{E}} \left[ \mathbb{1}\left\{ f \notin \mathcal{V}_t^{\mathcal{F}} \text{ or } f' \notin \mathcal{V}_t^{\mathcal{F}} \right\} \Big| \mathcal{H}_{t-1} \right] \right] \\
&\leq \sum_{t=1}^{T} \underset{\mathcal{H}_{t-1}}{\mathbb{E}} \left[ 2 \underset{f^\star}{\mathbb{E}} \left[ \mathbb{1}\left\{ f^\star \notin \mathcal{V}_t^{\mathcal{F}} \right\} \Big| \mathcal{H}_{t-1} \right] \right] \\
&= 2 \sum_{t=1}^{T} \underset{\mathcal{H}_{t-1},f^\star}{\mathbb{E}} \left[ \mathbb{1}\left\{ f^\star \notin \mathcal{V}_t^{\mathcal{F}} \right\} \right] \\
&= 2 \sum_{t=1}^{T} \underset{f^\star}{\mathbb{E}} \left[ \underset{\mathcal{H}_{t-1}}{\mathbb{E}} \left[ \mathbb{1}\left\{ f^\star \notin \mathcal{V}_t^{\mathcal{F}} \right\} \Big| f^\star \right] \right] \\
&\leq 2T\delta
\end{aligned}
$$

where the second inequality holds by the union bound and the condition that $f$ and $f'$ have the same posterior distributions as $f^\star$ conditioning on $\mathcal{H}_{t-1}$. The last two equalities holds by the law of probability. The last inequality is Lemma C.1.

Now let's bound $T_1$. We first notice that we can replace the expectation with the supremum:

$$
T_1 \leq \sum_{t=1}^{T} \underset{\mathcal{H}_{t-1}}{\mathbb{E}} \left[ \sup_{f,f' \in \mathcal{V}_t^{\mathcal{F}}} d_{\mathrm{TV}}\big( f(x_t), f'(x_t) \big) \right].
$$

Applying Lemma D.12, we have

$$
T_1 \leq 4\sqrt{T\beta_{\mathcal{F}}(t-1)} \cdot \dim_1\left( \mathcal{F}, 1/T \right) \cdot \log T
$$

Combining the upper bounds of $T_1$ and $T_2$ and setting $\delta = 1/T$, we complete the proof of (15).

**Proof of** (16). We split the left side of (16) into two cases based on whether $f$ and $f'$ belong to $\mathcal{V}_t^{\mathcal{F}}$:

$$
\begin{aligned}
&\sum_{t=1}^{T} \underset{\mathcal{H}_{t-1}}{\mathbb{E}} \left[ \underset{f,f'}{\mathbb{E}} \left[ d_{\mathrm{TV}}^2\big( f(x_t), f'(x_t) \big) \Big| \mathcal{H}_{t-1} \right] \right] \\
&\leq \sum_{t=1}^{T} \underset{\mathcal{H}_{t-1}}{\mathbb{E}} \left[ \underset{f,f'}{\mathbb{E}} \left[ \mathbb{1}\left\{ f \in \mathcal{V}_t^{\mathcal{F}} \text{ and } f' \in \mathcal{V}_t^{\mathcal{F}} \right\} d_{\mathrm{TV}}^2\big( f(x_t), f'(x_t) \big) \Big| \mathcal{H}_{t-1} \right] \right] \\
&\quad + \sum_{t=1}^{T} \underset{\mathcal{H}_{t-1}}{\mathbb{E}} \left[ \underset{f,f'}{\mathbb{E}} \left[ \mathbb{1}\left\{ f \notin \mathcal{V}_t^{\mathcal{F}} \text{ or } f' \notin \mathcal{V}_t^{\mathcal{F}} \right\} d_{\mathrm{TV}}^2\big( f(x_t), f'(x_t) \big) \Big| \mathcal{H}_{t-1} \right] \right] \\
&=: T_3 + T_4.
\end{aligned}
$$

Following a similar argument as in the proof of (15) above, we have $T_4 \leq 2T\delta$. For $T_3$, by the definition o $\mathcal{V}_t^{\mathcal{F}}$, we directly have $T_3 \leq 4\beta_{\mathcal{F}}(T)$. Setting $\delta = 1/T$, we complete the proof of (16).

**Proof of** (17) **and** (18). An identical argument can be applied to prove (17) and (18) leveraging Corollary C.4. □

**Lemma C.6.** *Under Assumption C.2, it holds that*

$$
\sum_{t=1}^{T} \underset{\mathcal{H}_{t-1}}{\mathbb{E}} \left[ \mathbb{1}\left\{ \underset{g,g'}{\mathbb{E}} \left[ \big| g(x_t) - g'(x_t) \big| \Big| \mathcal{H}_{t-1} \right] > \epsilon \right\} \right]
$$

$$
\leq \min\left\{ \frac{9\sqrt{T\beta_{\mathcal{G}}'(T)}}{\epsilon} \cdot \dim_1\left( \mathcal{G}, \epsilon/2 \right), \frac{21\beta_{\mathcal{G}}'(T)}{\epsilon^2} \cdot \dim_2(\mathcal{G}, \epsilon/2) \right\}.
$$

*where* $\beta_{\mathcal{G}}'(t) := 98\kappa^2 \log(2GTN_{[]}(\overline{\kappa}(t|\mathcal{Y}|)^{-1}, \mathcal{G}, \|\cdot\|_\infty))$. *In the inner expectation,* $g, g'$ *are sampled from the posterior of* $g^\star$ *conditioning on* $\mathcal{H}_{t-1}$.

*Proof of Lemma C.6.* We define the version space at round $t$ for $t \in [T]$ as follows

$$\mathcal{V}_t^{\mathcal{G}} = \left\{ g \in \mathcal{G} \ : \ \sum_{s=1}^{t-1} \left( \widehat{g}_t(x_s) - g(x_s) \right)^2 \leq \beta_{\mathcal{G}}(t-1) \right\}$$

where recall that $\widehat{g}_t$ is defined as the function that parameterizes the maximum likelihood estimator on the dataset $\{(x_s, y_s)\}_{s=1}^{t-1}$ (i.e., $\widehat{g}_t = g_{\widehat{f}_t}$). This construction aims to mimic the version space $\mathcal{V}^{\mathcal{G}}$ defined in Corollary C.4, which enable us to apply Corollary C.4.

We first split the left side into two cases based on whether both $g$ and $g'$ belong to $\mathcal{V}_t^{\mathcal{G}}$:

$$\sum_{t=1}^{T} \mathbb{E}_{\mathcal{H}_{t-1}} \left[ \mathbb{1} \left\{ \mathbb{E}_{g,g'} \left[ \left| g(x_t) - g'(x_t) \right| \, \middle| \, \mathcal{H}_{t-1} \right] > \epsilon \right\} \right]$$

$$= \sum_{t=1}^{T} \mathbb{E}_{\mathcal{H}_{t-1}} \left[ \mathbb{1} \left\{ \mathbb{E}_{g,g'} \left[ \mathbb{1} \{ g \in \mathcal{V}_t^{\mathcal{G}} \text{ and } g' \in \mathcal{V}_t^{\mathcal{G}} \} | g(x_t) - g'(x_t) | \, \middle| \, \mathcal{H}_{t-1} \right] \right. \right.$$

$$\left. \left. + \mathbb{E}_{g,g'} \left[ \mathbb{1} \{ g \notin \mathcal{V}_t^{\mathcal{G}} \text{ or } g' \notin \mathcal{V}_t^{\mathcal{G}} \} | g(x_t) - g'(x_t) | \, \middle| \, \mathcal{H}_{t-1} \right] > \epsilon \right\} \right]$$

$$\leq \sum_{t=1}^{T} \mathbb{E}_{\mathcal{H}_{t-1}} \left[ \mathbb{1} \left\{ \mathbb{E}_{g,g'} \left[ \mathbb{1} \{ g \in \mathcal{V}_t^{\mathcal{G}} \text{ and } g' \in \mathcal{V}_t^{\mathcal{G}} \} | g(x_t) - g'(x_t) | \, \middle| \, \mathcal{H}_{t-1} \right] > \epsilon/2 \right\} \right]$$

$$+ \sum_{t=1}^{T} \mathbb{E}_{\mathcal{H}_{t-1}} \left[ \mathbb{1} \left\{ \mathbb{E}_{g,g'} \left[ \mathbb{1} \{ g \notin \mathcal{V}_t^{\mathcal{G}} \text{ or } g' \notin \mathcal{V}_t^{\mathcal{G}} \} | g(x_t) - g'(x_t) | \, \middle| \, \mathcal{H}_{t-1} \right] > \epsilon/2 \right\} \right]$$

$$=: \mathtt{T}_1 + \mathtt{T}_2.$$

Here the inequality is due to the fact that $\mathbb{1} \{ a + b > c \} \leq \mathbb{1} \{ a > c/2 \} + \mathbb{1} \{ b > c/2 \}$ for any $a, b, c$. To bound $\mathtt{T}_2$, recalling that $g(\cdot) \in [0, G]$, then we have

$$\mathtt{T}_2 \leq \sum_{t=1}^{T} \mathbb{E}_{\mathcal{H}_{t-1}} \left[ \mathbb{1} \left\{ \mathbb{E}_{g,g'} \left[ \mathbb{1} \{ g \notin \mathcal{V}_t^{\mathcal{G}} \text{ or } g' \notin \mathcal{V}_t^{\mathcal{G}} \} \cdot G \, \middle| \, \mathcal{H}_{t-1} \right] > \epsilon/2 \right\} \right]$$

$$\leq \frac{2G}{\epsilon} \sum_{t=1}^{T} \mathbb{E}_{\mathcal{H}_{t-1}} \left[ \mathbb{E}_{g,g'} \left[ \mathbb{1} \{ g \notin \mathcal{V}_t^{\mathcal{G}} \text{ or } g' \notin \mathcal{V}_t^{\mathcal{G}} \} \, \middle| \, \mathcal{H}_{t-1} \right] \right]$$

$$\leq \frac{2G}{\epsilon} \sum_{t=1}^{T} \mathbb{E}_{\mathcal{H}_{t-1}} \left[ 2 \mathbb{E}_{g^\star} \left[ \mathbb{1} \{ g^\star \notin \mathcal{V}_t^{\mathcal{G}} \} \, \middle| \, \mathcal{H}_{t-1} \right] \right]$$

$$= \frac{4G}{\epsilon} \sum_{t=1}^{T} \mathbb{E}_{g^\star, \mathcal{H}_{t-1}} \left[ \mathbb{1} \{ g^\star \notin \mathcal{V}_t^{\mathcal{G}} \} \right]$$

$$= \frac{4G}{\epsilon} \sum_{t=1}^{T} \mathbb{E}_{g^\star} \left[ \mathbb{E}_{\mathcal{H}_{t-1}} \left[ \mathbb{1} \{ g^\star \notin \mathcal{V}_t^{\mathcal{G}} \} \, \middle| \, g^\star \right] \right]$$

$$\leq \frac{4GT\delta}{\epsilon}.$$

where the second inequality is by the fact that $\mathbb{1} \{ a > b \} \leq a/b$ for any $a \geq 0$ and $b > 0$, the third inequality holds since $g$ and $g'$ are identically distributed as $g^\star$ conditioning on $\mathcal{H}_{t-1}$, the two equalities holds by the law of probability, and the last inequality is Corollary C.4.

Now let us bound $\mathtt{T}_1$. We first notice that we can replace the expectation with the supremum:

$$\mathtt{T}_1 \leq \sum_{t=1}^{T} \mathbb{E}_{\mathcal{H}_{t-1}} \left[ \mathbb{1} \left\{ \sup_{g,g' \in \mathcal{V}_t^{\mathcal{G}}} | g(x_t) - g'(x_t) | > \epsilon/2 \right\} \right]$$

Recall that, for any $g \in \mathcal{V}_t^{\mathcal{G}}$, we have

$$\sum_{s=1}^{t-1} \left( g(x_s) - \widehat{g}(x_s) \right)^2 \leq \beta_{\mathcal{G}}(t-1).$$

Hence, applying Lemma D.11, we get

$$
\begin{aligned}
\mathtt{T}_1 &\leq \min\left\{ \left( \frac{2\sqrt{T\beta_{\mathcal{G}}(t-1)}}{\epsilon/2} + 1 \right) \cdot \dim_1\left(\mathcal{G}, \epsilon/2\right), \left( \frac{4\beta_{\mathcal{G}}(t-1)}{\epsilon^2/4} + 1 \right) \dim_2(\mathcal{G}, \epsilon/2) \right\} \\
&\leq \min\left\{ \frac{5\sqrt{T\beta_{\mathcal{G}}(t-1)}}{\epsilon} \cdot \dim_1\left(\mathcal{G}, \epsilon/2\right), \frac{17\beta_{\mathcal{G}}(t-1)}{\epsilon^2} \cdot \dim_2(\mathcal{G}, \epsilon/2) \right\}
\end{aligned}
$$

Combining the upper bounds of $\mathtt{T}_1$ and $\mathtt{T}_2$, we obtain

$$
\begin{aligned}
&\sum_{t=1}^{T} \mathbb{E}\left[ \mathbb{1}\left\{ \mathbb{E}_{g,g'}\left[ \left| g(x_t) - g'(x_t) \right| \,\Big|\, \mathcal{H}_{t-1} \right] > \epsilon \right\} \right] \\
&\leq \frac{4GT\delta}{\epsilon} + \min\left\{ \frac{5\sqrt{T\beta_{\mathcal{G}}(t-1)}}{\epsilon} \cdot \dim_1\left(\mathcal{G}, \epsilon/2\right), \frac{17\beta_{\mathcal{G}}(t-1)}{\epsilon^2} \cdot \dim_2(\mathcal{G}, \epsilon/2) \right\}.
\end{aligned}
$$

Setting $\delta = 1/(TG)$, the parameter $\beta_{\mathcal{G}}(t-1)$ becomes $\beta'_{\mathcal{G}}(t-1)$, which is upper bounded by $\beta'_{\mathcal{G}}(T)$. Then, we finish the proof. $\qquad\square$

## C.2 Bounding Bayesian regret

Now we are ready to prove Theorem 5.4. We will prove the upper bound on the Bayesian regret in this section and prove the upper bound on the number of queries in the next section.

For the ease of notation, we will omit the dependence of the state-value function on the initial state $s_1$ and simply write $V \coloneqq V_1(s_1)$ for any state-value function $V$ throughout the proof.

We start by simplifying the Bayesian regret. Since $\pi_t^1 = \pi_{t-1}^0$, we have

$$
\begin{aligned}
\mathrm{BayesRegret}_T &= \mathbb{E}_{r^\star, P^\star}\left[ \sum_{t=1}^{T} \left( 2V^\star - V^{\pi_t^0} - V^{\pi_t^1} \right) \right] \\
&= \mathbb{E}_{r^\star, P^\star}\left[ \sum_{t=1}^{T} \left( 2V^\star - V^{\pi_t^0} - V^{\pi_{t-1}^0} \right) \right] \\
&= \mathbb{E}_{r^\star, P^\star}\left[ \sum_{t=1}^{T} \left( V^\star - V^{\pi_t^0} \right) \right] + \mathbb{E}_{r^\star, P^\star}\left[ \sum_{t=0}^{T-1} \left( V^\star - V^{\pi_t^0} \right) \right] \\
&\leq 2\, \mathbb{E}_{r^\star, P^\star}\left[ \sum_{t=0}^{T} \left( V^\star - V^{\pi_t^0} \right) \right].
\end{aligned}
$$

Hence, we only need to consider the regret incurred by $\pi_t^0$. We defined $V_{r,P}^\pi$ as the state-value function of policy $\pi$ with reward function $r$ and model $P$. Given that, we can express the Bayesian regret as

$$
\mathrm{BayesRegret}_T \leq 2\, \mathbb{E}_{r^\star, P^\star}\left[ \sum_{t=0}^{T} \left( V_{r^\star, P^\star}^{\pi^\star} - V_{r^\star, P^\star}^{\pi_t^0} \right) \right]
$$

We reformulate the expectation by first taking the expectation of the historical data up to round $t - 1$ and then taking the conditional expectation of $P^\star$ and $r^\star$. Concretely, we denote $\mathcal{H}_{t-1} = \{\tau_1^0, \tau_1^1, (o_1), \ldots, \tau_{t-1}^0, \tau_{t-1}^1, (o_{t-1})\}$ as the history up to round $t - 1$, and then we have

$$
\begin{aligned}
\mathrm{BayesRegret}_T &\leq 2\sum_{t=0}^{T} \mathbb{E}_{\mathcal{H}_{t-1}} \mathbb{E}_{r^\star, P^\star}\left[ V_{r^\star, P^\star}^{\pi^\star} - V_{r^\star, P^\star}^{\pi_t^0} \,\Big|\, \mathcal{H}_{t-1} \right] \\
&= 2\sum_{t=0}^{T} \mathbb{E}_{\mathcal{H}_{t-1}} \mathbb{E}_{r^\star, P^\star}\left[ V_{r_t, P_t}^{\pi_t^0} - V_{r^\star, P^\star}^{\pi_t^0} \,\Big|\, \mathcal{H}_{t-1} \right].
\end{aligned}
$$

For the equality above, we note that $(r_t, P_t)$ and $(r^\star, P^\star)$ are identically distributed given $\mathcal{H}_{t-1}$ and $\pi_t^0$ and $\pi^\star$ are the respective optimal policies of $(r_t, P_t)$ and $(r^\star, P^\star)$. Hence, $V_{r_t, P_t}^{\pi_t^0}$ and $V_{r^\star, P^\star}^{\pi^\star}$ are identically distributed given $\mathcal{H}_{t-1}$, which is the reason that we can do the replacement in the conditional expectation. Next, we proceed by decomposing the regret:

$$
\begin{aligned}
\text{BayesRegret}_T =& 2\sum_{t=0}^{T} \mathop{\mathbb{E}}_{\mathcal{H}_{t-1}} \mathop{\mathbb{E}}_{r^\star, P^\star} \left[ V_{r_t, P_t}^{\pi_t^0} - V_{r_t, P^\star}^{\pi_t^0} + V_{r_t, P^\star}^{\pi_t^0} - V_{r_t, P^\star}^{\pi_t^1} + V_{r^\star, P^\star}^{\pi_t^1} - V_{r^\star, P^\star}^{\pi_t^0} \,\Big|\, \mathcal{H}_{t-1} \right] \\
=& 2\sum_{t=0}^{T} \mathop{\mathbb{E}}_{\mathcal{H}_{t-1}} \mathop{\mathbb{E}}_{r^\star, P^\star} \left[ V_{r_t, P_t}^{\pi_t^0} - V_{r_t, P^\star}^{\pi_t^0} + V_{r_t, P^\star}^{\pi_t^0} - V_{r_t, P^\star}^{\pi_t^1} + V_{r^\star, P^\star}^{\pi_t^1} - V_{r^\star, P^\star}^{\pi_t^0} \,\Big|\, \mathcal{H}_{t-1} \right] \\
=& 2\sum_{t=0}^{T} \mathbb{E}\left[ V_{r_t, P_t}^{\pi_t^0} - V_{r_t, P^\star}^{\pi_t^0} + V_{r_t, P^\star}^{\pi_t^0} - V_{r_t, P^\star}^{\pi_t^1} + V_{r^\star, P^\star}^{\pi_t^1} - V_{r^\star, P^\star}^{\pi_t^0} \right] \\
=& 2\left( \underbrace{\sum_{t=0}^{T} \mathbb{E}\left[ V_{r_t, P_t}^{\pi_t^0} - V_{r_t, P^\star}^{\pi_t^0} \right]}_{=:\text{T}_{\text{model}}} + \underbrace{\sum_{t=0}^{T} \mathbb{E}\left[ V_{r_t, P^\star}^{\pi_t^0} - V_{r_t, P^\star}^{\pi_t^1} + V_{r^\star, P^\star}^{\pi_t^1} - V_{r^\star, P^\star}^{\pi_t^0} \right]}_{=:\text{T}_{\text{reward}}} \right)
\end{aligned}
\tag{19}
$$

Here we added and substracted $V_{r_t, P^\star}^{\pi_t^0}$ and $V_{r^\star, P^\star}^{\pi_t^1}$ in the first equality. For the second equality, we note that $\pi_t^1 = \pi_{t-1}^0$ is measurable with respect to $\mathcal{H}_{t-1}$ (i.e., $\mathbb{E}[\pi_t^1 \,|\, \mathcal{H}_{t-1}] = \pi_t^1$).[4] Hence, $V_{r^\star, P^\star}^{\pi_t^1}$ and $V_{r_t, P^\star}^{\pi_t^1}$ are identically distributed conditioning on $\mathcal{H}_{t-1}$.

Next, we will show that the two terms, $\text{T}_{\text{model}}$ and $\text{T}_{\text{reward}}$, arise from the estimation errors in the model and reward, respectively.

**Bounding $\text{T}_{\text{model}}$.** By simulation lemma (Lemma D.2) and the tower rule, we have

$$
\begin{aligned}
\text{T}_{\text{model}} \leq& H\, \mathbb{E}\left[ \sum_{t=0}^{T} \sum_{h=1}^{H} \mathop{\mathbb{E}}_{(s_h, a_h)\sim d_h^{\pi_t^0}} d_{\text{TV}}\Big( P_t(s_h, a_h), P^\star(s_h, a_h) \Big) \right] \\
=& H\, \mathbb{E}\left[ \sum_{t=0}^{T} \sum_{h=1}^{H} d_{\text{TV}}\Big( P_t(s_h, a_h), P^\star(s_h, a_h) \Big) \right] \\
=& H\, \mathbb{E}\left[ \sum_{t=0}^{T} \sum_{h=1}^{H} \mathop{\mathbb{E}}_{P_t, P^\star}\left[ d_{\text{TV}}\Big( P_t(s_h, a_h), P^\star(s_h, a_h) \Big) \,\Big|\, \mathcal{H}_{t-1} \right] \right] \\
=& H \sum_{h=1}^{H} \mathbb{E}\left[ \sum_{t=0}^{T} \mathop{\mathbb{E}}_{P_t, P^\star}\left[ d_{\text{TV}}\Big( P_t(s_h, a_h), P^\star(s_h, a_h) \Big) \,\Big|\, \mathcal{H}_{t-1} \right] \right] \\
\leq& O\left( H^2 \cdot \dim_1\Big(\mathcal{P}, 1/T\Big) \cdot \log(T) \cdot \sqrt{T \log(T N_{[]}((HT|\mathcal{S}|)^{-1}, \mathcal{P}, \|\cdot\|_\infty))} \right)
\end{aligned}
$$

where the last step is Lemma C.5.

**Bounding $\text{T}_{\text{reward}}$.** By the definition of state-value function, we have

$$
\begin{aligned}
\text{T}_{\text{reward}} =& \sum_{t=0}^{T} \mathbb{E}\left[ r_t(\tau_t^0) - r_t(\tau_t^1) + r^\star(\tau_t^1) - r^\star(\tau_t^0) \right] \\
=& \sum_{t=0}^{T} \mathbb{E}\left[ (1 - Z_t)\Big( r_t(\tau_t^0) - r_t(\tau_t^1) + r^\star(\tau_t^1) - r^\star(\tau_t^0) \Big) \right] \\
&+ \sum_{t=0}^{T} \mathbb{E}\left[ Z_t\Big( r_t(\tau_t^0) - r_t(\tau_t^1) + r^\star(\tau_t^1) - r^\star(\tau_t^0) \Big) \right].
\end{aligned}
$$

---

[4]Strictly speaking, $\pi_t^1$ is measurable with respect to the $\sigma$-algebra generated by $\mathcal{H}_t$.

For the first term, by the query condition, it holds that

$$\sum_{t=0}^{T} \mathbb{E}\left[(1-Z_t)\left(r_t(\tau_t^0) - r_t(\tau_t^1) + r^\star(\tau_t^1) - r^\star(\tau_t^0)\right)\right]$$

$$= \sum_{t=0}^{T} \mathbb{E}\left[\mathbb{1}\left\{\mathbb{E}\left[|r(\tau_t^0) - r(\tau_t^1) - (r'(\tau_t^0) - r'(\tau_t^1))|\,\big|\,\mathcal{H}_{t-1}, \tau_t^0, \tau_t^1\right] \leq \epsilon\right\}\right.$$
$$\left.\left(r_t(\tau_t^0) - r_t(\tau_t^1) + r^\star(\tau_t^1) - r^\star(\tau_t^0)\right)\right]$$

$$= \sum_{t=0}^{T} \mathop{\mathbb{E}}_{\mathcal{H}_{t-1}, \tau_t^0, \tau_t^1} \mathbb{E}\left[\mathbb{1}\left\{\mathbb{E}\left[|r(\tau_t^0) - r(\tau_t^1) - (r'(\tau_t^0) - r'(\tau_t^1))|\,\big|\,\mathcal{H}_{t-1}, \tau_t^0, \tau_t^1\right] \leq \epsilon\right\}\right.$$
$$\left.\left(r_t(\tau_t^0) - r_t(\tau_t^1) + r^\star(\tau_t^1) - r^\star(\tau_t^0)\right)\,\bigg|\,\mathcal{H}_{t-1}, \tau_t^0, \tau_t^1\right]$$

$$= \sum_{t=0}^{T} \mathop{\mathbb{E}}_{\mathcal{H}_{t-1}, \tau_t^0, \tau_t^1}\left[\mathbb{1}\left\{\mathbb{E}\left[|r(\tau_t^0) - r(\tau_t^1) - (r'(\tau_t^0) - r'(\tau_t^1))|\,\big|\,\mathcal{H}_{t-1}, \tau_t^0, \tau_t^1\right] \leq \epsilon\right\}\right.$$
$$\left.\mathbb{E}\left[\left(r_t(\tau_t^0) - r_t(\tau_t^1) + r^\star(\tau_t^1) - r^\star(\tau_t^0)\right)\,\bigg|\,\mathcal{H}_{t-1}, \tau_t^0, \tau_t^1\right]\right]$$

$$\leq T\epsilon.$$

Here the first equality is by definition. The second equality is by the law of probability. The third equality holds since the indicator is measurable with respect to $\mathcal{H}_{t-1}$, $\tau_t^0$, and $\tau_t^1$. The last inequality is by the indicator and the fact that $r_t$ and $r^\star$ have the same posterior conditioning on $\mathcal{H}_{t-1}$. Plugging this upper bound back, we have

$$\mathtt{T}_{\text{reward}} \leq T\epsilon + \sum_{t=0}^{T} \mathbb{E}\left[Z_t\left(r_t(\tau_t^0) - r_t(\tau_t^1) + r^\star(\tau_t^1) - r^\star(\tau_t^0)\right)\right]$$
$$= T\epsilon + \sum_{t=0}^{T} \mathop{\mathbb{E}}_{\mathcal{H}_{t-1}}\left[Z_t \mathop{\mathbb{E}}_{r_t, r^\star}\left[\left|r_t(\tau_t^0) - r_t(\tau_t^1) + r^\star(\tau_t^1) - r^\star(\tau_t^0)\right|\,\bigg|\,\mathcal{H}_{t-1}\right]\right].$$

In the inner expectation of the second term above, we notice that both $r_t$ and $r^\star$ are sampled from the posterior of $r^\star$ conditioning on $\mathcal{H}_{t-1}$. Thus, we can invoke the second statement in Lemma C.5 where the function $g$ corresponds to $r_t$ and the function $g'$ corresponds to $r^\star$. This gives

$$\mathtt{T}_{\text{reward}} \leq O\left(T\epsilon + \sqrt{T\kappa^2 \log\left(HTN_{[]}(\overline{\kappa}(2T)^{-1}, \widetilde{\mathcal{R}}, \|\cdot\|_\infty)\right)} \cdot \dim_1\left(\widetilde{\mathcal{R}}, 1/T\right) \cdot \log(HT)\right).$$

**Conclusion.** Given the upper bounds on $\mathtt{T}_{\text{model}}$ and $\mathtt{T}_{\text{reward}}$, ignoring logarithmic factors on $T$ and $H$, we conclude that

$$\text{BayesRegret}_T = \widetilde{O}\left(H^2 \cdot \dim_1\left(\mathcal{P}, 1/T\right) \cdot \sqrt{T\log(N_{[]}((HT|\mathcal{S}|)^{-1}, \mathcal{P}, \|\cdot\|_\infty))}\right.$$
$$\left. + T\epsilon + \kappa \cdot \dim_1\left(\widetilde{\mathcal{R}}, 1/T\right) \cdot \sqrt{T\log\left(N_{[]}(\overline{\kappa}(2T)^{-1}, \widetilde{\mathcal{R}}, \|\cdot\|_\infty)\right)}\right).$$

## C.3 BOUNDING NUMBER OF QUERIES

It holds that

$$
\begin{aligned}
\text{BayesQueries}_T &= \mathbb{E}\left[\sum_{t=0}^T Z_t\right] \\
&= \sum_{t=0}^T \mathbb{E}_{\mathcal{H}_{t-1}}\left[Z_t \mathbb{1}\left\{\mathbb{E}_{r,r'\sim\rho_{\text{r},t}}\left[\left(\left(r(\tau_t^0) - r(\tau_t^1)\right) - \left(r'(\tau_t^0) - r'(\tau_t^1)\right)\right)\right] > \epsilon\right\}\right].
\end{aligned}
$$

We notice that, in the inner expectaion, $r$ and $r'$ are sampled from the posterior of $r^\star$ conditioning on $\mathcal{H}_{t-1}$ (recalling the definition of $\rho_{\text{r},t}$). Thus, we can invoke Lemma C.6 where the function $g$ corresponds to $r$ and the function $g'$ corresponds to $r'$. This gives

$$
\text{BayesQueries}_T \leq \min\left\{\frac{9\sqrt{T\beta_{\mathcal{R}}}}{\epsilon} \cdot \dim_1\left(\widetilde{\mathcal{R}}, \epsilon/2\right), \frac{21\beta_{\mathcal{R}}}{\epsilon^2} \cdot \dim_2(\widetilde{\mathcal{R}}, \epsilon/2)\right\}.
$$

where the inequality is Lemma C.6 and $\beta_{\mathcal{R}} := 98\kappa^2 \log(2THN_{[]}(\overline{\kappa}(2T)^{-1}, \widetilde{\mathcal{R}}, \|\cdot\|_\infty))$.

## C.4 GENERALIZING THEOREM 5.4 THROUGH SEC

In this section, we show that the dependence on the eluder dimension in Theorem 5.4 can be generalized to the *Sequential Extrapolation Coefficient (SEC)* (Xie et al., 2022). It have been shown by Xie et al. (2022) that the SEC can be upper bounded by the eluder dimenion, the Bellman-eluder dimension (Jin et al., 2021), and the bilinear rank (Du et al., 2021). Thus, SEC is a more general measure of complexity.

We start by introducing the Bayesian Sequential Extrapolation Coefficient (Bayesian SEC) in the preference-based feedback, which is a variant of the original SEC to accommodate the Bayesian and preference-based learning setting. We define the Bayesian SEC for a model class $\mathcal{P}$ as

$$
\mathsf{BayesSEC_P}(\mathcal{P}) := \mathbb{E}_{r^\star, P^\star}\left[\sup_{\substack{P_1,\ldots,P_T\in\mathcal{P} \\ (\pi_1^0,\pi_1^1),\ldots,(\pi_T^0,\pi_T^1)}} \sum_{t=1}^T \frac{\left(\sum_{h=1}^H \mathbb{E}_{s,a\sim d_h^{\pi_t}}\left[d_{\text{TV}}\left(P_t(\cdot\,|\,s,a), P^\star(\cdot\,|\,s,a)\right)\right]\right)^2}{1\vee\sum_{i=1}^{t-1}\sum_{h=1}^H \mathbb{E}_{s,a\sim d_h^{\pi_i}}\left[d_{\text{TV}}^2\left(P_t(\cdot\,|\,s,a), P^\star(\cdot\,|\,s,a)\right)\right]}\right]
$$

and the Bayesian SEC for a reward function class $\mathcal{R}$ as

$$
\mathsf{BayesSEC_R}(\mathcal{R}) := \mathbb{E}_{r^\star, P^\star}\left[\sup_{\substack{r_1,\ldots,r_T\in\mathcal{R} \\ (\pi_1^0,\pi_1^1),\ldots,(\pi_T^0,\pi_T^1)}} \sum_{t=1}^T \frac{\mathbb{E}_{\tau_t^0\sim\pi_t^0,\tau_t^1\sim\pi_t^1}\left[\left(r_t(\tau_t^0) - r_t(\tau_t^1) + r^\star(\tau_t^1) - r^\star(\tau_t^0)\right)^2\right]}{1\vee\sum_{i=1}^{t-1}\mathbb{E}_{\tau_i^0\sim\pi_i^0,\tau_i^1\sim\pi_i^1}\left[\left(r_t(\tau_i^0) - r_t(\tau_i^1) + r^\star(\tau_i^1) - r^\star(\tau_i^0)\right)^2\right]}\right]
$$

We note that the Bayesian SEC can be easily reduced to the frequentist SEC by specifying the prior on $r^\star$ and $P^\star$ to be the Dirac delta function. We note that our definition of the reward function SEC involves squaring within the expectation in the numerator rather than externally, potentially making it larger.

Now, we are ready to state the generalization of Theorem 5.4.

**Theorem C.7.** *PbTS (Algorithm 2) guarantees that*

$$
\text{BayesRegret}_T = \widetilde{O}\left(T\epsilon + H\sqrt{\mathsf{BayesSEC_P}(\mathcal{P})\cdot TH\cdot\iota_{\mathcal{P}}} + \sqrt{\mathsf{BayesSEC_R}(\mathcal{R})\cdot T\cdot\kappa^2\cdot\iota_{\mathcal{R}}}\right),
$$

$$
\text{BayesQueries}_T = \widetilde{O}\left(\frac{\kappa^2\cdot\mathsf{BayesSEC_R}(\mathcal{R})\cdot\iota_{\mathcal{R}}}{\epsilon^2}\right).
$$

*where we denote $\iota_{\mathcal{P}} := \log(N_{[]}((HT|\mathcal{S}|)^{-1}, \mathcal{P}, \|\cdot\|_\infty))$ and $\iota_{\mathcal{R}} := \log(N_{[]}(\overline{\kappa}(2T)^{-1}, \widetilde{\mathcal{R}}, \|\cdot\|_\infty))$.*

The proofs are provided in the following sections (Appendices C.4.1 and C.4.2).

### C.4.1 BOUNDING BAYESIAN REGRET VIA SEC

We first prove the upper bound on the Bayesian regret. Following (19), it suffices to separately bound $\mathsf{T}_{\text{model}}$ and $\mathsf{T}_{\text{reward}}$ via SEC. We start with $\mathsf{T}_{\text{model}}$.

**Bounding** $\mathtt{T}_{\mathrm{model}}$**.**    By simulation lemma (Lemma D.2), we have

$$\mathtt{T}_{\mathrm{model}} \leq H \, \mathbb{E} \left[ \sum_{t=0}^{T} \sum_{h=1}^{H} \mathop{\mathbb{E}}_{(s_h, a_h) \sim d_h^{\pi_t^0}} d_{\mathrm{TV}} \Big( P_t(s_h, a_h), P^{\star}(s_h, a_h) \Big) \right]$$

where $d_h^{\pi}(s, a)$ denotes the probability of $\pi$ reaching $(s, a)$ at time step $h$. By multiplying and dividing by the same term, we obtain the following.

$$\mathtt{T}_{\mathrm{model}} \leq H \, \mathbb{E} \left[ \sum_{t=0}^{T} \frac{\sum_{h=1}^{H} \mathbb{E}_{(s_h, a_h) \sim d_h^{\pi_t^0}} d_{\mathrm{TV}} \Big( P_t(s_h, a_h), P^{\star}(s_h, a_h) \Big)}{\sqrt{1 \vee \sum_{i=0}^{t-1} \sum_{h=1}^{H} \mathbb{E}_{(s_h, a_h) \sim d_h^{\pi_i^0}} d_{\mathrm{TV}}^2 \Big( P_t(s_h, a_h), P^{\star}(s_h, a_h) \Big)}} \right. $$

$$\left. \times \sqrt{1 \vee \sum_{i=0}^{t-1} \sum_{h=1}^{H} \mathop{\mathbb{E}}_{(s_h, a_h) \sim d_h^{\pi_i^0}} d_{\mathrm{TV}}^2 \Big( P_t(s_h, a_h), P^{\star}(s_h, a_h) \Big)} \right]$$

$$\leq H \underbrace{\sqrt{\mathbb{E} \left[ \sum_{t=0}^{T} \frac{\Big( \sum_{h=1}^{H} \mathbb{E}_{(s_h, a_h) \sim d_h^{\pi_t^0}} d_{\mathrm{TV}} \big( P_t(s_h, a_h), P^{\star}(s_h, a_h) \big) \Big)^2}{1 \vee \sum_{i=0}^{t-1} \sum_{h=1}^{H} \mathbb{E}_{(s_h, a_h) \sim d_h^{\pi_i^0}} d_{\mathrm{TV}}^2 \big( P_t(s_h, a_h), P^{\star}(s_h, a_h) \big)} \right]}}_{(\mathrm{i})}$$

$$\times \underbrace{\sqrt{\sum_{t=0}^{T} \mathbb{E} \left[ 1 \vee \sum_{i=0}^{t-1} \sum_{h=1}^{H} \mathop{\mathbb{E}}_{(s_h, a_h) \sim d_h^{\pi_i^0}} d_{\mathrm{TV}}^2 \Big( P_t(s_h, a_h), P^{\star}(s_h, a_h) \Big) \right]}}_{(\mathrm{ii})}$$

Here we note that $a \vee b := \max(a, b) \leq a + b$ for $a, b \geq 0$. The last step above is the Cauchy-Schwarz inequality. We observe that term (i) is exactly bounded by the Bayesian SEC of models. For term (ii), we can invoke (16) in Lemma C.5 since $P_t$ has an identical posterior distribution as $P^{\star}$ given the historical data up to round $t - 1$. This gives (ii) $\leq \widetilde{O}(\sqrt{TH \cdot \iota_{\mathcal{P}}})$. Plugging these back, we obtain

$$\mathtt{T}_{\mathrm{model}} \leq \widetilde{O} \left( H \sqrt{\mathsf{BayesSEC}_{\mathsf{P}}(\mathcal{P}) \cdot TH \cdot \iota_{\mathcal{P}}} \right)$$

**Bounding** $\mathtt{T}_{\mathrm{reward}}$**.**    The reward part can be bounded similarly, except that we need to additionally consider the query conditions. To begin with, by the definition of state-value function, we have

$$\mathtt{T}_{\mathrm{reward}} = \sum_{t=0}^{T} \mathbb{E} \left[ r_t(\tau_t^0) - r_t(\tau_t^1) + r^{\star}(\tau_t^1) - r^{\star}(\tau_t^0) \right]$$

$$= \underbrace{\sum_{t=0}^{T} \mathbb{E} \left[ (1 - Z_t) \Big( r_t(\tau_t^0) - r_t(\tau_t^1) + r^{\star}(\tau_t^1) - r^{\star}(\tau_t^0) \Big) \right]}_{(\mathrm{i})} + \underbrace{\sum_{t=0}^{T} \mathbb{E} \left[ Z_t \Big( r_t(\tau_t^0) - r_t(\tau_t^1) + r^{\star}(\tau_t^1) - r^{\star}(\tau_t^0) \Big) \right]}_{(\mathrm{ii})}.$$

Following the same argument as in the proof of Appendix C.2, we can bound (i) as (i) $\leq T\epsilon$. Now we proceed to bound (ii), which is the regret incurred when making queries. By multiplying and

dividing by the same term, we obtain the following.

$$
\text{(ii)} = \sum_{t=0}^{T} \mathop{\mathbb{E}}_{\pi_t^0, \pi_t^1, r_t, Z_t} \left[ \frac{\mathbb{E}\left[Z_t\left(r_t(\tau_t^0) - r_t(\tau_t^1) + r^\star(\tau_t^1) - r^\star(\tau_t^0)\right) \,\big|\, \pi_t^0, \pi_t^1, r_t, Z_t\right]}{\sqrt{1 \vee \sum_{i=0}^{t-1} \mathbb{E}\left[Z_t\left(r_t(\tau_i^0) - r_t(\tau_i^1) + r^\star(\tau_i^1) - r^\star(\tau_i^0)\right)^2 \,\Big|\, \pi_t^0, \pi_t^1, r_t, Z_t\right]}} \right.
$$

$$
\left. \times \sqrt{1 \vee \sum_{i=0}^{t-1} \mathop{\mathbb{E}}_{\tau_t^0, \tau_t^1}\left[Z_t\left(r_t(\tau_i^0) - r_t(\tau_i^1) + r^\star(\tau_i^1) - r^\star(\tau_i^0)\right)^2 \,\Big|\, \pi_t^0, \pi_t^1, r_t, Z_t\right]} \right]
$$

$$
\leq \underbrace{\sqrt{\sum_{t=0}^{T} \mathop{\mathbb{E}}_{\pi_t^0, \pi_t^1, r_t, Z_t}\left[ \frac{\left(\mathbb{E}\left[Z_t\left(r_t(\tau_t^0) - r_t(\tau_t^1) + r^\star(\tau_t^1) - r^\star(\tau_t^0)\right) \,\big|\, \pi_t^0, \pi_t^1, r_t, Z_t\right]\right)^2}{1 \vee \sum_{i=0}^{t-1} \mathbb{E}\left[Z_t\left(r_t(\tau_i^0) - r_t(\tau_i^1) + r^\star(\tau_i^1) - r^\star(\tau_i^0)\right)^2 \,\Big|\, \pi_t^0, \pi_t^1, r_t, Z_t\right]} \right]}}_{\text{(iii)}}
$$

$$
\times \underbrace{\sqrt{\sum_{t=0}^{T}\left(1 \vee \sum_{i=0}^{t-1}\mathbb{E}\left[Z_t\left(r_t(\tau_i^0) - r_t(\tau_i^1) + r^\star(\tau_i^1) - r^\star(\tau_i^0)\right)^2\right]\right)}}_{\text{(iv)}}\Bigg)
$$

where the last inequality is Cauchy-Schwarz inequality. We observe that term (iii) is exactly bounded by the Bayesian SEC of reward functions via Jensen's inequality. For term (iv), we invoke (18) in Lemma C.5 and get (iv) $\leq \widetilde{O}(\sqrt{\kappa^2 T \cdot \iota_{\mathcal{R}}})$. Now plugging these upper bounds back, we get

$$
\mathsf{T}_{\text{reward}} \leq T\epsilon + \widetilde{O}\left(\sqrt{\mathsf{BayesSEC}_{\mathsf{R}}(\mathcal{R}) \cdot \kappa^2 T \cdot \iota_{\mathcal{R}}}\right).
$$

**Conclusion.** Given the upper bounds on $\mathsf{T}_{\text{model}}$ and $\mathsf{T}_{\text{reward}}$, we conclude that

$$
\mathsf{BayesRegret}_T \leq \widetilde{O}\left(H\sqrt{\mathsf{BayesSEC}_{\mathsf{P}}(\mathcal{P}) \cdot TH \cdot \iota_{\mathcal{P}}} + T\epsilon + \sqrt{\mathsf{BayesSEC}_{\mathsf{R}}(\mathcal{R}) \cdot \kappa^2 T \cdot \iota_{\mathcal{R}}}\right).
$$

### C.4.2 BOUNDING NUMBER OF QUERIES VIA SEC

By the definition of $Z_t$, we have

$$
\mathsf{BayesQueries}_T = \mathbb{E}\left[\sum_{t=0}^{T} Z_t\right] = \mathbb{E}\left[\sum_{t=0}^{T} Z_t^2\right]
$$

$$
= \mathbb{E}\left[\sum_{t=0}^{T} Z_t \mathbb{1}\left\{\mathop{\mathbb{E}}_{r,r'\sim\rho_{\mathrm{r},t}}\left[\left|(r(\tau_t^0) - r(\tau_t^1)) - (r'(\tau_t^0) - r'(\tau_t^1))\right|\right] > \epsilon\right\}\right]
$$

$$
= \mathbb{E}\left[\sum_{t=0}^{T} Z_t \mathbb{1}\left\{\mathop{\mathbb{E}}_{r,r'\sim\rho_{\mathrm{r},t}}\left[\left|(r(\tau_t^0) - r(\tau_t^1)) - (r'(\tau_t^0) - r'(\tau_t^1))\right|^2\right] > \epsilon^2\right\}\right]
$$

$$
\leq \frac{1}{\epsilon^2} \mathbb{E}\left[\sum_{t=0}^{T} Z_t \mathop{\mathbb{E}}_{r,r'\sim\rho_{\mathrm{r},t}}\left[\left|(r(\tau_t^0) - r(\tau_t^1)) - (r'(\tau_t^0) - r'(\tau_t^1))\right|^2\right]\right]
$$

$$
= \frac{1}{\epsilon^2} \mathbb{E}\left[\sum_{t=0}^{T} Z_t \mathop{\mathbb{E}}_{r,r'}\left[\left|(r_t(\tau_t^0) - r_t(\tau_t^1)) - (r^\star(\tau_t^0) - r^\star(\tau_t^1))\right|^2 \,\Big|\, \mathcal{H}_{t-1}\right]\right].
$$

Here the last step is by the definition of $\rho_{r,t}$. Now we swap the order of expectation and get $\text{BayesQueries}_T$

$$
\leq \frac{1}{\epsilon^2} \mathbb{E}\left[\sum_{t=0}^{T} \mathop{\mathbb{E}}_{\tau_t^0, \tau_t^1}\left[Z_t\big((r_t(\tau_t^0) - r_t(\tau_t^1)) - (r^\star(\tau_t^0) - r^\star(\tau_t^1))\big)^2 \,\Big|\, \pi_t^0, \pi_t^1, r^\star, r_t, Z_t\right]\right]
$$

$$
= \frac{1}{\epsilon^2} \mathbb{E}\left[\sum_{t=0}^{T} \frac{\mathbb{E}_{\tau_t^0, \tau_t^1}\left[Z_t\big((r_t(\tau_t^0) - r_t(\tau_t^1)) - (r^\star(\tau_t^0) - r^\star(\tau_t^1))\big)^2 \,\Big|\, \pi_t^0, \pi_t^1, r^\star, r_t, Z_t\right]}{1 \vee \sum_{i=0}^{t-1} \mathbb{E}_{\tau_i^0, \tau_i^1}\left[Z_t\big((r_t(\tau_i^0) - r_t(\tau_i^1)) - (r^\star(\tau_i^0) - r^\star(\tau_i^1))\big)^2 \,\Big|\, \pi_i^0, \pi_i^1, r^\star, r_t, Z_t\right]}\right.
$$
$$
\left. \times \left(1 \vee \sum_{i=0}^{t-1} \mathop{\mathbb{E}}_{\tau_i^0, \tau_i^1}\left[Z_t\big((r_t(\tau_i^0) - r_t(\tau_i^1)) - (r^\star(\tau_i^0) - r^\star(\tau_i^1))\big)^2 \,\Big|\, \pi_i^0, \pi_i^1, r^\star, r_t, Z_t\right]\right)\right]
$$

$$
\leq \frac{1}{\epsilon^2} \mathbb{E}\left[\sum_{t=0}^{T} \frac{\mathbb{E}_{\tau_t^0, \tau_t^1}\left[Z_t\big((r_t(\tau_t^0) - r_t(\tau_t^1)) - (r^\star(\tau_t^0) - r^\star(\tau_t^1))\big)^2 \,\Big|\, \pi_t^0, \pi_t^1, r^\star, r_t, Z_t\right]}{1 \vee \sum_{i=0}^{t-1} \mathbb{E}_{\tau_i^0, \tau_i^1}\left[Z_t\big((r_t(\tau_i^0) - r_t(\tau_i^1)) - (r^\star(\tau_i^0) - r^\star(\tau_i^1))\big)^2 \,\Big|\, \pi_i^0, \pi_i^1, r^\star, r_t, Z_t\right]}\right]
$$
$$
\times \widetilde{O}\left(\kappa^2 \cdot \iota_{\mathcal{R}}\right)
$$

where in the last step we applied Jensen's inequality to the first multiplicative term and applied (18) in Lemma C.5 to the second term. We observe that the first term can be bounded by SEC, and thus we obtain

$$
\text{BayesQueries}_T \leq \widetilde{O}\left(\frac{\kappa^2 \cdot \text{BayesSEC}_R(\mathcal{R}) \cdot \iota_{\mathcal{R}}}{\epsilon^2}\right).
$$

## D  TECHNICAL LEMMAS

**Lemma D.1.** *Under Assumption 3.1, we have $\overline{\kappa}(\Phi(a) - \Phi(b)) \leq a - b \leq \kappa(\Phi(a) - \Phi(b))$ for any $a, b \in [0, H]$.*

*Proof of Lemma D.1.* We note that $\Phi(a) - \Phi(b) = \int_b^a \Phi'(x)\,\mathrm{d}x$ and

$$
\kappa^{-1}(a - b) \leq (a - b)\inf_{x \in [a,b]} \Phi'(x) \leq \int_b^a \Phi'(x)\,\mathrm{d}x \leq (a - b)\sup_{x \in [a,b]} \Phi'(x) \leq \overline{\kappa}^{-1}(a - b).
$$

$\square$

**Lemma D.2** (Simulation lemma)**.** *For any models $P, \widehat{P}$ and any policy $\pi$, we have*

$$
\left|V_{P,1}^\pi(s_1) - V_{\widehat{P},1}^\pi(s_1)\right| \leq \sum_{h=1}^{H} \mathop{\mathbb{E}}_{(s_h, a_h) \sim d_{P,h}^\pi} \left|\mathop{\mathbb{E}}_{s' \sim P(s_h, a_h)} V_{\widehat{P}, h+1}^\pi(s') - \mathop{\mathbb{E}}_{s' \sim \widehat{P}(s_h, a_h)} V_{\widehat{P}, h+1}^\pi(s')\right|
$$
$$
\leq H \sum_{h=1}^{H} \mathop{\mathbb{E}}_{(s_h, a_h) \sim d_{P,h}^\pi} d_{\mathrm{TV}}\left(P(\cdot \,|\, s_h, a_h), \widehat{P}(\cdot \,|\, s_h, a_h)\right)
$$

*where $d_{P,h}^\pi(s, a)$ is the probability of $\pi$ reaching $(s, a)$ at time step $h$ given model $P$.*

**Lemma D.3** (Gaussian concentration)**.** *Let $\epsilon \sim \mathcal{N}(0, c\Sigma^{-1})$ for $c \in \mathbb{R}^+$ and $\Sigma$ a positive definite matrix. Then, for any $\delta > 0$, we have*

1. $\Pr\left(\|\epsilon\|_\Sigma > \sqrt{2cd\log(2d/\delta)}\right) \leq \delta.$

2. $\mathbb{E}\left[\|\epsilon\|_\Sigma\right] \leq \sqrt{cd}.$

*Proof of Lemma D.3.* The proof of the first inequality is provided in Abeille & Lazaric (2017, Appendix A). We provide a proof here for completeness. Let $\eta \sim \mathcal{N}(0, I_d)$ where $I_d$ denotes the $d$-dimensional identity matrix. Fix an arbitrary $\alpha$. We have

$$
\Pr\left(\|\eta\|_2 > \alpha\sqrt{d}\right) \leq \Pr\left(\exists i : |\eta_i| > \alpha\right) \leq d\Pr\left(|\eta_1| > \alpha\right) \leq d \cdot 2e^{-\alpha^2/2}
$$

where the second inequality is the union bound, and the last inequality is the standard concentration inequality for one-dimensional Gaussian random variable. We choose $\alpha = \sqrt{2\log(2d/\delta)}$ and get

$$\Pr\left(\|\eta\|_2 > \sqrt{2d\log(2d/\delta)}\right) \le \delta.$$

Let $B$ denote the square root of $\Sigma$, i.e., $BB = \Sigma$. Then, we have

$$\|\epsilon\|_\Sigma = \sqrt{\epsilon^\top \Sigma \epsilon} = \sqrt{\left(\sqrt{c}B^{-1}\eta\right)^\top BB \left(\sqrt{c}B^{-1}\eta\right)} = \sqrt{c}\|\eta\|_2 \tag{20}$$

Hence, it holds that

$$\Pr\left(\|\epsilon\|_\Sigma > \sqrt{2cd\log(2d/\delta)}\right) = \Pr\left(\|\eta\|_2 > \sqrt{2d\log(2d/\delta)}\right) \le \delta.$$

This proves the first inequality. To prove the second inequality, we note that, by (20),

$$\mathbb{E}\left[\|\epsilon\|_\Sigma\right] = \sqrt{c}\,\mathbb{E}\left[\|\eta\|_2\right] \le \sqrt{c}\sqrt{\mathbb{E}\left[\eta^\top\eta\right]} = \sqrt{cd}.$$

$\square$

The following inequalities are well-known, and we use the version in Zhu & Nowak (2022).

**Lemma D.4.** *(Zhu & Nowak, 2022) Let $\{X_t\}_{t\le T}$ be a sequence of positive valued random variables adapted to a filtration $\mathfrak{F}_t$, and let $\mathbb{E}_t[\cdot] := \mathbb{E}[\cdot \mid \mathfrak{F}_{t-1}]$. If $X_t \le B$ almost surely, then with probability at least $1-\delta$, the following holds:*

$$\sum_{t=1}^T X_t \le \frac{3}{2}\sum_{t=1}^T \mathbb{E}_t[X_t] + 4B\log(1/\delta),$$

$$\sum_{t=1}^T \mathbb{E}_t[X_t] \le 2\sum_{t=1}^T X_t + 8B\log(1/\delta).$$

**Lemma D.5** (Self-normalized process). *(Abbasi-Yadkori et al., 2011) Let $\{x_i\}_{i=1}^\infty$ be a real valued stochastic process sequence over the filtration $\{\mathcal{F}_i\}_{i=1}^\infty$. Let $x_i$ be conditionally $B$-subgaussian given $\mathcal{F}_{i-1}$. Let $\{\phi_i\}_{i=1}^\infty$ with $\phi_i \in \mathcal{F}_{i-1}$ be a stochastic process in $\mathbb{R}^d$ with each $\|\phi_i\| \le L_\phi$. Define $\Sigma_i = \lambda I + \sum_{j=1}^{i-1}\phi_i\phi_i^\top$. Then for any $\delta > 0$ and all $i \ge 0$, with probability at least $1-\delta$*

$$\left\|\sum_{i=1}^{k-1}\phi_i x_i\right\|_{\Sigma_k^{-1}}^2 \le 2B^2\log\left(\frac{\det(\Sigma_i)^{1/2}\det(\lambda I)^{-1/2}}{\delta}\right) \le 2B^2\left(d\log\left(\frac{\lambda + kL_\phi^2}{\lambda}\right) + \log(1/\delta)\right)$$

**Lemma D.6** (Elliptical potential lemma). *(Abbasi-Yadkori et al., 2011) Following the setting of Lemma D.5, we have*

$$\sum_{i=1}^k \min\left\{1, \|\phi_i\|_{\Sigma_i^{-1}}^2\right\} \le 2d\log\left(\frac{\lambda + kL_\phi^2}{\lambda}\right)$$

**Lemma D.7** (Sum of features). *(Jin et al., 2020) Following the setting of Lemma D.5, we have*

$$\sum_{i=1}^{k-1}\|\phi_i\|_{\Sigma_k^{-1}}^2 \le d$$

The following lemma converts the TV distance between Gaussian distributions into the distance between their means.

**Lemma D.8.** *Assume $p_1 = \mathcal{N}(\mu_1, \sigma^2 I)$ and $p_2 = \mathcal{N}(\mu_2, \sigma^2 I)$ are two Gaussian distributions over $\mathbb{R}^d$. Assume $\|\mu_1 - \mu_2\|_2 \le m$. Then, we have*

$$\|\mu_1 - \mu_2\|_2 \cdot \underbrace{\min\left\{\sqrt{\frac{1}{2e\pi\sigma^2}}, \frac{1}{2m}\right\}}_{=:C_1} \le d_{\mathrm{TV}}(p_1, p_2) \le \|\mu_1 - \mu_2\|_2 \cdot \underbrace{\sqrt{\frac{1}{2\pi\sigma^2}}}_{=:C_2}$$

*Proof.* Without loss of generality, we can rotate and translate the $\mathbb{R}^d$ space and assume that $p_1 = \mathcal{N}(0, \sigma^2 I)$ and $p_2 = \mathcal{N}(\alpha \cdot e_1, \sigma^2 I)$ where $\alpha = \|\mu_1 - \mu_2\|_2$ and $e_1 = [1, 0, \ldots, 0] \in \mathbb{R}^d$. It is clear that the total variation distance is preserved. Then, by definition,

$$d_{\mathrm{TV}}(p_1, p_2) = \frac{1}{2} \int_{x \in \mathbb{R}^d} |p_1(x) - p_2(x)| \; \mathrm{d}x = \int_{x \cdot e_1 \leq \alpha/2} p_1(x) - p_2(x) \; \mathrm{d}x$$

where the last step holds since $p_1(x) - p_2(x) \geq 0$ if and only if $x \cdot e_1 \leq \alpha/2$. Note that the above formulation is independent of any dimension except for first dimension, so we can marginize other dimensions and reduce into a one-dimensional problem:

$$\begin{aligned} d_{\mathrm{TV}}(p_1, p_2) &= \int_{x \leq \alpha/2} \mathcal{N}(x \,|\, 0, \sigma^2) - \mathcal{N}(x \,|\, \alpha, \sigma^2) \; \mathrm{d}x \\ &= \Big( F(\alpha/(2\sigma)) - F(-\alpha/(2\sigma)) \Big) \\ &= \int_{-\alpha/(2\sigma)}^{\alpha/(2\sigma)} \mathcal{N}(x \,|\, 0, 1) \; \mathrm{d}x \end{aligned}$$

where we denote F as the CDF of standard Gaussian $\mathcal{N}(0, 1)$ in the middle step. Below we show that the above integral is upper bounded by $C_2 \alpha$ and lower bounded by $C_1 \alpha$.

**Upper bound.** For an upper bound, we note that for any $v > 0$, we have

$$\int_{-v}^{v} \mathcal{N}(x \,|\, 0, 1) \; \mathrm{d}x \leq 2v \cdot \max_x \mathcal{N}(x \,|\, 0, 1) \leq v \sqrt{\frac{2}{\pi}}.$$

Inserting $v = \alpha/(2\sigma)$, we obtain the desired upper bound.

**Lower bound.** For a lower bound, we note that for any $0 < v \leq m/(2\sigma)$, if $v < 1$, we have

$$\int_{-v}^{v} \mathcal{N}(x \,|\, 0, 1) \; \mathrm{d}x \geq 2v \max_{x \in [-v, v]} \mathcal{N}(x \,|\, 0, 1) \geq 2v \mathcal{N}(1 \,|\, 0, 1) = 2v \cdot \frac{e^{-1/2}}{\sqrt{2\pi}} = v \cdot \sqrt{\frac{2}{e\pi}}.$$

When $v \geq 1$, we have

$$\int_{-v}^{v} \mathcal{N}(x \,|\, 0, 1) \; \mathrm{d}x \geq \int_{-\sigma}^{\sigma} \mathcal{N}(x \,|\, 0, 1) \; \mathrm{d}x \approx 0.6827 \geq v \cdot \frac{1}{2v} \geq v \cdot \frac{\sigma}{m}.$$

Combining the two cases together, we have

$$\int_{-v}^{v} \mathcal{N}(x \,|\, 0, 1) \; \mathrm{d}x \geq v \cdot \min \left\{ \sqrt{\frac{2}{e\pi}}, \frac{\sigma}{m} \right\}.$$

Inserting $v = \alpha/(2\sigma)$, we obtain the desired lower bound. $\square$

**Lemma D.9.** *Let $x \sim \mathcal{N}(0, I_d)$ and A be a positive semi-definite matrix. Then, we have*

$$\mathbb{E}_x \|x\|_A \geq \sqrt{\frac{2 \operatorname{tr}(A)}{\pi}}.$$

*Proof.* Let $P$ denote the orthogonal matrix diagonalizing $A$, i.e., $A = P \Lambda P^\top$ where $\Lambda = \operatorname{diag}(\lambda_1, \ldots, \lambda_d)$. Then, we have $\mathbb{E}_x \|x\|_A = \mathbb{E}_x \sqrt{x^\top A x} = \mathbb{E}_y \sqrt{y^\top \Lambda y}$ where $y = P^\top x \sim \mathcal{N}(0, I_d)$. Now we consider the following optimization problem:

$$\min_{M \in \mathcal{M}} f(M) := \mathbb{E}_y \sqrt{y^\top M y} \quad \text{where} \quad \mathcal{M} := \{M \succeq 0 : M \text{ is diagonal and } \operatorname{tr}(M) = \operatorname{tr}(\Lambda)\}.$$

Then, we have $\Lambda \in \mathcal{M}$ and $\mathbb{E}_x \|x\|_A = f(\Lambda)$. Thus, the solution to the above optimization problem is a lower bound of $\mathbb{E}_x \|x\|_A$. Since $f(M)$ is concave in $M$, it must attain the minimum on the bounary of $\mathcal{M}$. By symmetry of entries in the diagonal, we know that the minimizer must have $\operatorname{tr}(\Lambda)$ at one entry and 0 at all other entries. Hence, the minimum value is

$$\mathbb{E}_y \sqrt{\operatorname{tr}(\Lambda) y_1^2} = \sqrt{\operatorname{tr}(\Lambda)} \, \mathbb{E}_y |y_1| = \sqrt{\frac{2 \operatorname{tr}(\Lambda)}{\pi}}.$$

We finish the proof by noticing that $\operatorname{tr}(\Lambda) = \operatorname{tr}(A)$. $\square$

The following lemma was originally established by Agarwal et al. (2020) and later adapted by Wu et al. (2023a) to accommodate infinitely large function classes.

**Lemma D.10** (Maximum likelihood estimation). *(Wu et al., 2023a, Lemma C.3) Consider a sequential conditional probability estimation problem. Let $\mathcal{X}$ and $\mathcal{Y}$ be the instance space and the target space, respectively. Let $\mathcal{F} : (\mathcal{X} \times \mathcal{Y}) \to \mathbb{R}$ denote the function class. We are given a dataset $D := \{(x_i, y_i)\}_{i=1}^{n}$ where $x_i \sim \mathcal{D}_i$ and $y_i \sim f^\star(x, \cdot)$. We assume $f^\star \in \mathcal{F}$. For the data generating process, we assume the data distribution $\mathcal{D}_i$ is history-dependent, i.e., $x_i$ can depend on the previous samples: $x_1, y_1, \ldots, x_{i-1}, y_{i-1}$ for any $i \in [n]$. We fix $\delta \in (0, 1)$. Let $\widehat{f}$ denote the maximum likelihood estimator,*

$$\widehat{f} = \arg\max_{f \in \mathcal{F}} \sum_{i=1}^{n} \log f(x_i, y_i).$$

*Then, we have*

$$\sum_{i=1}^{n} \mathbb{E}_{x \sim \mathcal{D}_i} d_{\mathrm{TV}}^2\left(\widehat{f}(x, \cdot), f^\star(x, \cdot)\right) \leq 10 \log\left(N_{[]}\left((n|\mathcal{Y}|)^{-1}, \mathcal{F}, \|\cdot\|_\infty\right)/\delta\right) \tag{21}$$

*with probability at least $1 - \delta$. Here $|\mathcal{Y}|$ denotes the total measure of space $\mathcal{Y}$, i.e., $|\mathcal{Y}| := \int_{\mathcal{Y}} 1 \, \mathrm{d}y$.*

### D.1 ELUDER DIMENSION

The following lemmas are adapted from Russo & Van Roy (2013) and Liu et al. (2022a).

**Lemma D.11.** *Following the notation of Definition 5.2, define*

$$\mathcal{F}_t = \left\{ f \in \mathcal{F} \; : \; \sum_{s=1}^{t-1} d^2\big(f(x_s), \widehat{f}_t(x_s)\big) \leq \beta \right\}$$

*where $\widehat{f}_t \in \mathcal{F}$ is an arbitrary function. Then, we have*

$$\sum_{t=1}^{T} \mathbb{1}\left\{ \sup_{f,f' \in \mathcal{F}_t} d\left(f(x_t), f'(x_t)\right) \geq \omega \right\} \leq \left(\frac{2\sqrt{T\beta}}{\omega} + 1\right) \dim_1(\mathcal{F}, \omega), \tag{22}$$

$$\sum_{t=1}^{T} \mathbb{1}\left\{ \sup_{f,f' \in \mathcal{F}_t} d\left(f(x_t), f'(x_t)\right) \geq \omega \right\} \leq \left(\frac{4\beta}{\omega^2} + 1\right) \dim_2(\mathcal{F}, \omega) \tag{23}$$

*for any constant $\omega > 0$.*

*Proof of Lemma D.11.* We first prove (22) and then (23).

**Proof of (22).** We begin by showing that if

$$\sup_{f,f' \in \mathcal{F}_t} d\left(f(x_t), f'(x_t)\right) \geq \omega$$

for some $t \in [T]$, then $x_t$ is $\omega$-dependent on at most $2\beta/\omega$ disjoint subsequence of its predecessors. To see this, we note that, if $x_t$ is $\omega$-dependent on a subsequence $(x_{i_1}, x_{i_2}, \ldots, x_{i_n})$ of its predecessors, we must have

$$\sum_{s=1}^{n} d\big(f(x_{i_s}) - f'(x_{i_s})\big) > \omega.$$

Hence, if $x_t$ is $\omega$-dependent on $l$ disjoint subsequences, we have

$$\sum_{s=1}^{t-1} d\big(f(x_s), f'(x_s)\big) > l\omega. \tag{24}$$

For the left-hand side, we also have

$$
\begin{aligned}
\sum_{s=1}^{t-1} d\big(f(x_s), f'(x_s)\big) &\leq \sum_{s=1}^{t-1} d\big(f(x_s), \widehat{f}_t(x_s)\big) + \sum_{s=1}^{t-1} d\big(\widehat{f}_t(x_s), f'(x_s)\big) \\
&\leq \sqrt{T}\sqrt{\sum_{s=1}^{t-1} d^2\big(f(x_s), \widehat{f}_t(x_s)\big)} + \sqrt{T}\sqrt{\sum_{s=1}^{t-1} d^2\big(\widehat{f}_t(x_s), f'(x_s)\big)} \\
&\leq 2\sqrt{T\beta}
\end{aligned}
\tag{25}
$$

where the first inequality is the triangle inequality, and the second inequality holds by the definition of $\mathcal{F}_t$. Combining (24) and (25), we get that $l \leq 2\sqrt{T\beta}/\omega$.

Next, we show that for any sequence $(x'_1, \ldots, x'_\tau)$, there is at least one element that is $\omega$-dependent on at least $\tau/d - 1$ disjoint subsequence of its predecessors, where $d := \dim_1(\mathcal{F}, \omega)$. To show this, let $m$ be the integer satisfying $md + 1 \leq \tau \leq md + d$. We will construct $m$ disjoint subsequences, $B_1, \ldots, B_m$. At the beginning, let $B_i = (x'_i)$ for $i \in [m]$. If $x'_{m+1}$ is $\omega$-dependent on each subsequence $B_1, \ldots, B_m$, then we are done. Otherwise, we select a subsequence $B_i$ which $x'_{m+1}$ is $\omega$-independent of and append $x'_{m+1}$ to $B_i$. We repeat this process for all elements with indices $j > m + 1$ until either $x'_j$ is $\omega$-dependent on each subsequence or $j = \tau$. For the latter, we have $\sum_{i=1}^{m} |B_i| \geq md$, and since each element of a subsequence $B_i$ is $\omega$-independent of its predecesors, we must have $|B_i| = d$ for all $i$. Then, $x_\tau$ must be $\omega$-dependent on each subsequence by the definition of eluder dimension.

Finally, let's set the sequence $(x'_1, \ldots x'_\tau)$ to be the subsequence of $(x_1, \ldots, x_T)$ consisting of elements $x_t$ for which $d\big(f(x_t), f'(x_t)\big) > \omega$. As we have established, we have

1. each $x'_i$ is $\ell_1$-norm $\omega$-dependent on at most $2\sqrt{T\beta}/\omega$ disjoint subsequences, and

2. some $x'_i$ is $\ell_1$-norm $\omega$-dependent on at least $\tau/d - 1$ disjoint subsequences.

Therefore, we must have $\tau/d - 1 \leq 2\sqrt{T\beta}/\omega$, implying that $\tau \leq (2\sqrt{T\beta}/\omega + 1)d$.

**Proof of** (23). The proof is quite similar to the proof of (22). For $\ell_2$-norm, following the same argument, we can show the following:

1. each $x'_i$ is $\ell_2$-norm $\omega$-dependent on at most $4\beta/\omega^2$ disjoint subsequences, and

2. some $x'_i$ is $\ell_2$-norm $\omega$-dependent on at least $\tau/d - 1$ disjoint subsequences.

Therefore, we must have $\tau/d - 1 \leq 4\beta/\omega^2$, implying that $\tau \leq (4\beta/\omega^2 + 1)d$. $\qquad\square$

**Lemma D.12.** *Following the setting of Lemma D.11, assume $d(\cdot, \cdot)$ is upper bounded by $D \leq \beta$. Then, we have*

$$
\sum_{t=1}^{T} \sup_{f, f' \in \mathcal{F}_t} d\big(f(x_t), f'(x_t)\big) \leq 4\sqrt{T\beta} \cdot \dim_1(\mathcal{F}, 1/T) \cdot \log(DT)
$$

*Proof of Lemma D.12.* For notational simplicity, we denote

$$
w_t := \sup_{f, f' \in \mathcal{F}_t} d\big(f(x_t), f'(x_t)\big)
$$

Now we fix a constant $\Delta > 0$, and then we have

$$
\begin{aligned}
\sum_{t=1}^{T} w_t &= \sum_{t=1}^{T} \int_0^D \mathbb{1}\{w_t \geq \delta\} \, \mathrm{d}\delta \\
&= \int_0^D \sum_{t=1}^{T} \mathbb{1}\{w_t \geq \delta\} \, \mathrm{d}\delta \\
&\leq \Delta T + \int_\Delta^D \sum_{t=1}^{T} \mathbb{1}\{w_t \geq \delta\} \, \mathrm{d}\delta \\
&\leq \Delta T + \int_\Delta^D \left( \frac{2\sqrt{T\beta}}{\delta} + 1 \right) \dim_1(\mathcal{F}, \delta) \, \mathrm{d}\delta \\
&\leq \Delta T + \int_\Delta^D \frac{3\sqrt{T\beta}}{\delta} \cdot \dim_1(\mathcal{F}, \delta) \, \mathrm{d}\delta \\
&\leq \Delta T + 3\sqrt{T\beta} \cdot \dim_1(\mathcal{F}, \Delta) \cdot \log(D/\Delta)
\end{aligned}
$$

where the second inequality is Lemma D.11, and the last inequality uses the fact that $\dim_1(\mathcal{F}, \delta)$ is non-increasing in $\delta$. We can conclude the proof by setting $\Delta = 1/T$. $\qquad\square$

