# OpenReview forum: "Making RL with Preference-based Feedback Efficient via Randomization"
_ICLR.cc/2024/Conference — ICLR 2024 poster_

### Official Review · Reviewer_V7cf · 2023-10-26

**Soundness:** 3 good
**Presentation:** 3 good
**Contribution:** 3 good
**Rating:** 6
**Confidence:** 3

**Summary:**

This paper considers Reinforcement Learning with preference-based feedback. Two algorithms are proposed with corresponding bounds on the (Bayesian) regret and (Bayesian) query complexity. The novelty lies at the application of randomization into the algorithm design in the preference-based feedback setup, making the algorithm more computationally efficient. It shows that there is a trade-off between the regret and the query times, i.e., more queries lead to less regret. The relationship is quantitatively characterized in the main results in terms of the parameter $\beta$.

**Strengths:**

- For MDPs with linear structure, it proposes an algorithm that utilizes randomizations to facilitate the learning process under the context of preference feedback.
- When the MDPs are not linear, it introduces a Thompson sampling based algorithm with accompanying Bayesian regret bound and query complexity bound.
- For the proposed algorithms, regret bounds and query complexities are derived, which quantitatively characterizes the tradeoff between them. This provides insights on how to tune the threshold $\epsilon$ in the algorithm in order to achieve certain performance, which I  appreciate a lot.
- By injecting randomness to the algorithm, it improves the computation efficiency of the RL algorithm for the preference-based feedback problem.

**Weaknesses:**

- It would be great if the authors can illustrate the empirical performances of the proposed algorithms. Algorithm 2 seems to be computationally intensive. Additionally, it would be convincing if we can observe the tradeoff between the regret and query times in the experiments.

Minors:
- While most notations are well explained, some notations are used without being properly defined, e.g., $\hat{\theta}_{P,t,h}$. This may hinder the readers in other relevant field from reading this paper. Hope the authors can further improve the paper presentation.

**Questions:**

- In terms of the regret bound for algorithm 1 (Theorem 4.2), can the authors comment on the dependence on $d$? Compared to the work by Zanette et al. (2020), it seems the power has increased a lot.

---

> ### Author Response · Authors · 2023-11-17
>
> Thank you for your valuable feedback.
>
> Regarding your concern on empirical performance: we acknowledge that investigating the empirical performance of the proposed algorithm would be interesting. However, our focus is on the theoretical side of RL with preference feedback with the goal of advancing our understanding of how to do RL with preference feedback with strong theoretical guarantees. There are many existing works on theoretical RLHF that don't have experiments as well (e.g., [1,2,3]). Nevertheless, that didn't prevent them from being recognized for their theoretical contribution. Our work was built on top of them, but compared to theirs, this work significantly improved the computation complexity (e.g., from exponential complexity to a polynomial complexity) and query complexity (e.g., from a linear query complexity to a sublinear query complexity).
>
> Regarding the notational issue, we will ensure to include more explanations for the notations in the next version of our paper. Here $\hat\theta_{P,t,h}$ (line 7, alg 1) is defined as the least squares estimate of the parameter of the state-action value function at the next step $h+1$.
>
> The dependence of Theorem 4.2 on $d$: we believe that the dependence on $d$ can be reduced through a more refined algorithmic design. In particular, implementing the same truncation technique (the $\omega(s,a)$ thing below Eq. (1)) could probably improve the dependence on $d$. However, this would make the algorithm super complicated and make the analysis tedious. The primary goal of this paper is to introduce a randomized approach to achieve efficiency; further improvement of this bound could be an intriguing direction for future research. We will ensure to include this remark in the next version of our paper.
>
> [1] Chen, Xiaoyu, et al. "Human-in-the-loop: Provably efficient preference-based reinforcement learning with general function approximation." ICML, 2022
>
> [2] Saha, Aadirupa, Aldo Pacchiano, and Jonathan Lee. "Dueling RL: Reinforcement Learning with Trajectory Preferences." AISTATS, 2023
>
> [3] Wang, Yuanhao, Qinghua Liu, and Chi Jin. "Is RLHF More Difficult than Standard RL?." NeurIPS, 2023

---

### Official Review · Reviewer_eu8c · 2023-10-29

**Soundness:** 3 good
**Presentation:** 3 good
**Contribution:** 3 good
**Rating:** 6
**Confidence:** 2

**Summary:**

This paper considers learning linear MDP (Assp 4.1) with preferential feedback. The unobservability of rewards makes learning significantly harder than standard RL. The author proposes PR-LSVI (Alg 1). PR-LSVI combines the MLE and Gaussian exploration to balance the exploration versus exploitation tradeoff. The algorithm only queries the feedback when the two trajectories can have a large gap (2). Theorem 4.2 bounds regret and query complexity. The paper also introduced a posterior sampling algorithm (Alg 2) and derived its Bayesian regret and query complexity. The paper is somewhat more detailed than it should be but readable in general. I could not follow the details of the paper but found no issues.

Minor comments:
* Alg 1 Line 15: Quey -> Query

**Strengths:**

* Learning of linear MDP under limited feedback with optimal regret in terms of $T$ ($\tilde{O}(\sqrt{T})$)
* Conceptually simple algorithm (Alg 1 and 2), which is computationally tractable and amount of query is controllable.

**Weaknesses:**

* Large volume: 53 pages is a bit large for conference proceedings. Not to mention, the shorter, the better s.t. the same results.
* Suboptimal dependence on parameters except for $T$, such as $H$, which makes the exploration larger than what is required (Alg 1).
* No experimental/simulation results.
* Disjointness of two results: Two algorithms are not connected together, and they have different objectives (frequentist or Bayesian).

**Questions:**

* Could you elaborate with the connection of the paper by Wang et al. (2023)?

* Could you elaborate on $\omega$ after (1)? To my understanding, this controls the rewards in subsequent rounds. Essentially, the exploration and exploitation tradeoff is resolved by demonstrating that the exploration is enough (and decaying at the same rate as uncertainty), and describing the mechanism would help the reader.

* One of the closest papers to this is Zanette et al 2019 which combines randomized exploration with least squares. Obviously, this paper is new in the sense that it deals with preferential noise. Other than that, are there technical novelty that should be highlighted in this paper?

**Details Of Ethics Concerns:**

The algorithm is theoretical and has no human subjective, no crowdsourcing, etc.

---

> ### Author Response · Authors · 2023-11-17
>
> Thank you for your valuable feedback. Please find our response to your questions below.
>
> 1. **elaborate with the connection of the paper by Wang et al. (2023)**
>
> They also consider learning from preference-based feedback. They first proposed a reduction framework (their Alg. 1) that can be computationally efficient and then proposed a complete algorithm (their Alg. 2). However, for both algorithms, they proposed to compares the current greedy policy with a fixed comparator, thus only achieving a PAC bound. Instead, we propose to draw two trajectories from a combination of new and older policies. This better balances exploration and exploitation and leads to a sublinear regret guarantee.
>
> They also explored learning from general preferences and demonstrated its connection to game theory (their Section 4). This setting is beyond the scope of our study, and whether a computationally efficient algorithm exists in that setting remaines a question.
>
> In addition, it's important to note that we also focus on query complexity --- we achieve a near-optimal balance between regret and query complexity while they don't focus on the query complexity.
>
> We will appropriately incorporate this part into the next version of our paper to make the connection clear.
>
> 2. **elaborate on $\omega$ after (1)**
>
> This truncation trick is from Zanette et al. (2020) and is used to control the abnormally high value estimates. Without such truncation, the value functions $\overline Q\_{t,h}$ and $\overline V\_{t,h}$ could be exponentially large on state-action pairs where $\phi(s,a)^\top \Sigma\_{t,h} \phi(s,a)$ is small. Moreover, the usual "value clipping" trick (i.e., simply constraining the value function within the range of $[0,H-h+1]$ by clipping) cannot work here since it introduces bias to the random walk analysis.
>
> Specifically, by introducing $\omega$, we are truncating according to the value of $\|\phi(s,a)\|\_{\Sigma\_{t-1,h}^{-1}}$. When the second and the third case of $\omega$ hold (i.e., $\|\phi(s,a)\|\_{\Sigma\_{t-1,h}^{-1}} > \alpha\_L$), the uncertainty in the direction of $\phi(s,a)$ could be very large, and thus the value of $\phi(s,a)^\top\overline\theta\_{P,t,h}$ could be abnormally large. In this case, we have to truncate it. On the other hand, if the first case holds (i.e., $\|\phi(s,a)\|\_{\Sigma\_{t-1,h}^{-1}} \leq \alpha\_L$), we know that the uncertainty in the direction of $\phi(s,a)$ is small and thus the estimate $\phi(s,a)^\top\overline\theta\_{P,t,h}$ is reasonable, so we don't need to truncate it in this case.
>
> In analysis, we showed that, although this truncation (the second and the third case) leads to biased estimate, it will not happen for too many times (Lemma B.14). Hence, for most of the time, the first case will be active and we don't do any truncation.
>
> We will appropriately incorporate this explanation into the next version of our paper.

---

> ### Author Response · Authors · 2023-11-17
>
> 3. **technical novelty that should be highlighted in this paper?**
>
> We highlight some new technical components of Algorithm 1:
>
> - Since the feedback is trajectory-wise, we need to design random noise that preserves the state-action-wise format (so that it can be used in DP) but captures the trajectory-wise uncertainty. The random noise used in original RLSVI did not meet its requirements, and we addressed this issue by maintaining the covariance matrix using trajectory-wise feature differences.
>
> - We assumed that the preference feedback is generated from some probabilistic model, characterized by a general link function $\Phi$. This setting is more general and necessitates the use of a novel technique. Specifically, we use to learn a reward model and use MLE generalization bound to capture the uncertainty in the learned reward model.
>
> - We design a new regret decomposition technique for regret analysis to accommodate preference-based feedback. Particularly, we decompose regret into a form that characterizes the *reward difference* between $\pi\_t^0$ and $\pi\_t^1$:
> 	$$
> 	\text{Regret}\_T \lesssim
> 		\sum\_{t=1}^T (  \overline V\_t - \widetilde V\_t  ) - (V^{\pi\_t^0} - V^{\pi\_t^1}  )
> 	$$
> 	where $\overline V\_t$ is an estimate of $V^{\pi^0\_t}$, and $\widetilde V\_t := \mathbb{E}\_{\tau\sim\pi\_t^1}[\sum\_{h=1}^H\phi(s\_h,a\_h)^\top\overline{\theta}\_{r,t}]$ is an estimate of $V^{\pi^1\_t}$ under the real transition and the learned reward model. This is different from RLSVI for standard RL, and is necessary since we cannot in general guarantee the learned reward model will be accurate in a state-action-wise manner under the preference-based feedback.
>
> - Our algorithms have a new randomized active learning procedure for reducing the number of queries, and our analysis achieves a near-optimal tradeoff between regret and query complexity;
>
> - In every round $t$, we propose to draw a pair of trajectories where one is from the current greedy policy $\pi^0\_t$ while the other is from the greedy policy $\pi^0\_{t-1}$ at the previous round. This approach makes sure that $\pi^1\_{t}$ is conditionally independent of the random Gaussian noises introduced at round $t$. This trick is the key to proving optimism (with a constant probability) when learning from preference-based feedback, i.e., $\overline V\_t - \widetilde V\_t  \geq V^\star - V^{\pi^1\_t}$ (recall $\widetilde V\_t$ is an estimate of $V^{\pi^1\_t}$ using the current reward $\overline{\theta}\_{\text{r},t}$). However, this was not an issue in the original RLSVI algorithm for standard RL.
>
> Furthermore, our second algorithm (PbTS) extends beyond the linear function case, which is also new.
>
> 4. **No experimental/simulation results.**
>
> We acknowledge that investigating the empirical performance of the proposed algorithm would be interesting. However, our focus is on the theoretical side of RL with preference feedback with the goal of advancing our understandings of how to do RL with preference feedback with strong theoretical guarantees . There are many existing works on theoretical RLHF that doesn't have experiments as well (e.g., [1,2,3]). Nevertheless, that didn't prevent them from being recognized for their theoretical contribution. Our work built on top of them, but compared to theirs, this work significantly improved the computation complexity (e.g., from exponential complexity to a polynomial complexity) and query complexity (e.g., from a linear query complexity to a sublinear query complexity).
>
> [1] Chen, Xiaoyu, et al. "Human-in-the-loop: Provably efficient preference-based reinforcement learning with general function approximation." ICML, 2022
>
> [2] Saha, Aadirupa, Aldo Pacchiano, and Jonathan Lee. "Dueling RL: Reinforcement Learning with Trajectory Preferences." AISTATS, 2023..
>
> [3] Wang, Yuanhao, Qinghua Liu, and Chi Jin. "Is RLHF More Difficult than Standard RL?." NeurIPS, 2023

---

> > ### Comment · Reviewer_eu8c · 2023-11-22
> > **Thank you for the clarifications**
> >
> > Dear authors,
> >
> > Thank you for your clarifications. I am inclined to keep the current rating (WA).

---

### Official Review · Reviewer_8PN9 · 2023-10-30

**Soundness:** 3 good
**Presentation:** 2 fair
**Contribution:** 2 fair
**Rating:** 5
**Confidence:** 4

**Summary:**

This paper studied RL with preference feedback and provided regret guarantee for linear MDP. Some extensions to nonlinear case using eluder dimension is also studied.

**Strengths:**

An interesting problem and solid theory.

**Weaknesses:**

I have some major concerns:

1. The theoretical contributions are incremental. The proposed algorithm is fairly standard (RLSVI) and almost identical as previous work. The proof technique is very routine and it is hard for me to see novelty here. There are a large body of theoretical works based on linear MDP and using model-free algorithms. The author didn't have a concrete comparison with them. Extension to nonlinear function approximation using eluder dimension is fairly well-known as well. People know for a while the only known function class for smaller eluder dimension is linear and generalized linear. There are many more general complexity measures for RL: Bellman-Eluder, Decision-Estimation Coefficient, bilinear class. The appendix is very long. I feel it might not be good to prove everything from scratch. There are so many prior works and I might suggest the authors to query existing lemma as much as possible to respect prior works.

2. This work is far away from the practical RLHF use case. There is no experiment at all and the algorithm seems to be practically feasible. For example, it is unclear how to implement LLM fine-tuning through RLHF using algorithm 2. It is also unclear why we care about cumulative regret in the context of RLHF for LLM alignment.

**Questions:**

See above.

---

> ### Author Response · Authors · 2023-11-17
>
> Thank you for your valuable feedback. Please find our response to your concerns below.
>
> Regarding the theoretical contributions, we argue that our work makes two big advancements in our understanding of how to do RL with preference feedback with provable guarantees. First, we emphasize that this work overcomed the computational challenges in **RL with preference-based feedback** (i.e., theoretical RLHF) -- prior work does not even have polynomial running time for tabular MDPs while our algorithm achieves poly running time for linear MDP (which generalizes tabular MDP). We proposed the very first algorithm that achieves sublinear worst-case regret and computational efficiency simultaneously in this field. We discussed the challenges in achieving computational efficiency in Appendix A and also provided comparisons with related works on RL with preference feedback. It explains how this work contributes to advancing this field. Secondly, the algorithm integrates an active learning procedure and attains a near-optimal tradeoff between the regret and the query complexity, which is another big contribution compared to all prior theory work on RLHF.
>
>
> Below we address your specific concerns.
>
> **1. "The proposed algorithm is fairly standard (RLSVI) and almost identical as previous work."**
>
> We do not think our first algorithm is identical to RLSVI, as we have briefly highlighted some key differences at the bottom of page 6. While we are inspired by RLSVI of injecting random noise, all other components of our method are distinct. We provide a more detailed explanation of these differences here:
>
> - Since the feedback is trajectory-wise, we need to design random noise that preserves the state-action-wise format (so that it can be used in DP) but captures the trajectory-wise uncertainty. The random noise used in the original RLSVI did not meet its requirements, and we addressed this issue by maintaining the covariance matrix using trajectory-wise feature differences;
>
> - We assumed that the preference feedback is generated from some probabilistic model, characterized by a general link function $\Phi$. This setting is more general and necessitates the use of a novel technique. Specifically, we use to learn a reward model and use MLE generalization bound to capture the uncertainty in the learned reward model.
>
> - We design a new regret decomposition technique for regret analysis to accommodate preference-based feedback. Particularly, we decompose regret into a form that characterizes the *reward difference* between $\pi\_t^0$ and $\pi\_t^1$:
> 	$$
> 	\text{Regret}\_T \lesssim
> 		\sum\_{t=1}^T (  \overline V\_t - \widetilde V\_t  ) - (V^{\pi\_t^0} - V^{\pi\_t^1}  )
> 	$$
> 	where $\overline V\_t$ is an estimate of $V^{\pi^0\_t}$, and $\widetilde V\_t := \mathbb{E}\_{\tau\sim\pi\_t^1}[\sum\_{h=1}^H\phi(s\_h,a\_h)^\top\overline{\theta}\_{r,t}]$ is an estimate of $V^{\pi^1\_t}$ under the real transition and the learned reward model. This is different from RLSVI for standard RL, and it is necessary since we cannot in general guarantee the learned reward model will be accurate in a state-action-wise manner under the preference-based feedback.
>
> - Our algorithms have a new randomized active learning procedure for reducing the number of queries, and our analysis achieves a near-optimal tradeoff between regret and query complexity. Note the prior RLSVI work does not even consider minimizing query complexity!
>
> - In every round $t$, we propose to draw a pair of trajectories where one is from the current greedy policy $\pi^0\_t$ while the other is from the greedy policy $\pi^0\_{t-1}$ at the previous round. This approach makes sure that $\pi^1\_{t}$ is conditionally independent of the random Gaussian noises introduced at round $t$. This trick is the key to proving optimism (with a constant probability) when learning from preference-based feedback, i.e., $\overline{V}\_t - \widetilde V\_t  \geq V^\star - V^{\pi^1\_t}$ (recall $\widetilde V\_t$ is an estimate of $V^{\pi^1\_t}$ using the current reward $\overline{\theta}\_{\text{r},t}$). However, this was not an issue in the original RLSVI algorithm for standard RL.
>
> Considering all these points, we do not think the algorithm is identical to RLSVI. Our approach gets the idea of injecting random noise from RLSVI, but the analysis beyond that point diverges. Furthermore, our second algorithm (PbTS) extends beyond the linear function case, which also distinguishes us from RLSVI.
>
> We would also like to point out that if using a simple idea like injecting random noise can already address key limitations of prior theoretical RLHF works (e.g., making their computation and queries efficient), this indeed should be considered as a good contribution -- when simple and existing technique makes the algorithms work better, we should simply use them instead of inventing new wheels.

---

> ### Author Response · Authors · 2023-11-17
>
> **2. "There are a large body of theoretical works based on linear MDP and using model-free algorithms. The author didn't have a concrete comparison with them."**
>
> We are happy to compare to other related theoretical works. However, we emphasize that the setting and the goal of this work are different from those works on linear MDP for standard RL (i.e., learning from a reward signal).
>
> In this paper, our primary focus is on **achieving computational efficiency when learning from preference-based feedback**, rather than proposing a new algorithm for linear MDPs. We acknowledge the existence of numerous works on linear MDP for standard RL. However, given that computational efficiency is not a concern in standard RL, the significance of comparing our method with most of them is unclear. Instead, we mostly compared our algorithm with those that also learns from preference-based feedback, which can be found in the related work section as well as Appendix A.
>
> However, it is possible that we may have missed some related work that should be compared with. Please let us know if we have missed any.
>
> **3. "Extension to nonlinear function approximation using eluder dimension is fairly well-known as well. People know for a while the only known function class for smaller eluder dimension is linear and generalized linear. There are many more general complexity measures for RL: Bellman-Eluder, Decision-Estimation Coefficient, bilinear class"**
>
> We really appreciate your suggestion. We have worked out the theoretical guarantee of our second algorithm using the Sequential Exptrapolation Coefficient (SEC) [1]. It has been shown in [1] that SEC is more general because it subsumes eluder dimension, Bellman-eluder dimension, and bilinear rank. We have revised and uploaded the draft to include this additional theoretical findings. Please find it towards the end of page 9 (highlighed in red) and detailed in Appendix C.4. We briefly introduce it below.
>
> We start by definition a notion of Bayesian SEC, which extend the original frequentist SEC to the Bayesian setting. The concrete definition can be found in page 49. We note that Bayesian SEC can be easily reduced to the frequentist SEC by specifying the prior distribution to be the Dirac delta function. Then we defined $\mathtt{BayesSEC\_P}$ and $\mathtt{BayesSEC\_R}$ as the Bayesian SEC of the model class and reward class, respectively.
>
> We showed that PbTS (algorithm 2), without any modification, guarantees the following:
> $$
> \begin{align*}
>     \text{BayesRegret}\_T =
>     \widetilde O\bigg(&
> 	T\epsilon
> 	+ H \sqrt{\mathtt{BayesSEC\_P}(\mathcal{P})\cdot TH \cdot \iota\_\mathcal{P}}
> 	+ \sqrt{\mathtt{BayesSEC\_R}(\mathcal{R})\cdot T\cdot\kappa^2\cdot\iota\_\mathcal{R}}
>     \bigg),\\
>     \text{BayesQueries}\_T =
>     \widetilde O\bigg( &
> 		\frac{\kappa^2\cdot\mathtt{BayesSEC\_R}(\mathcal{R})\cdot\iota\_\mathcal{R}}{\epsilon^2}
> 		\bigg).
> \end{align*}
> $$
> where we denote $\iota\_\mathcal{P} := \log(N\_{[]}((HT|\mathcal{S}|)^{-1},\mathcal{P},\|\cdot\|\_\infty))$ and $\iota\_{\mathcal R} := \log(N\_{[]}(\overline{\kappa}(2 T)^{-1},\widetilde{\mathcal R},\|\cdot\|\_\infty))$.
>
> This new result indicates that our algorithm not only works under small eluder dimension (as we initially proposed in the paper) but also works under small SEC, which is more general than eluder dimension. We will ensure to keep this new finding in the next version of our paper.
>
> [1] Xie, Tengyang, et al. "The role of coverage in online reinforcement learning." ICLR, 2023

---

> ### Author Response · Authors · 2023-11-17
>
> **4. "The appendix is very long. I feel it might not be good to prove everything from scratch.""**
>
> Although the high-level idea of injecting random noise is similar, the proof is different. This is mainly due to the preference-based feedback and the active learning procedure. Hence, we can't invoke many previous lemmas. As emphasized in this responses earlier, preference feedback and active queries introduce significant challenges (especially if you want to achieve a near-optimal balance between them). Consequently, we have to develop new techniques to address these issues effectively.
>
> Analogously, we can consider the comparison between standard bandits and dueling bandits. Dueling bandit is a variant of bandits that only receives preference feedback. The introduction of preference feedback presents substantial challenges to existing works, making the direct application of existing lemmas of standard bandits infeasible. To illustrate this, we can consider a comparison between [2] and [3]. The former examines standard bandits, while the latter studies dueling bandits. Despite both relying on an online regression oracle, their analytical approaches are fundamentally different.
>
> In addition, we also proposed a new randomized active learning procedure that achieves the optimal tradeoff in $T$, which is different from most of the prior active learning and selective sampling literature already (e.g., [4]). This alone already requires developing new techniques and more careful analysis.
>
> We emphasize that we do not intend to make the appendix long. We tried our best to reuse existing results whenever it is appropriate. But as we said, due to the different settings (preference feedback and query complexity minimization), we do need to develop new analysis.
>
> [2] Foster, Dylan, and Alexander Rakhlin. "Beyond ucb: Optimal and efficient contextual bandits with regression oracles." ICML, 2020
>
> [3] Sekhari, Ayush, et al. "Contextual bandits and imitation learning via preference-based active queries." NeurIPS, 2023
>
> [4] Dekel, Ofer, Claudio Gentile, and Karthik Sridharan. "Selective sampling and active learning from single and multiple teachers." The Journal of Machine Learning Research 13.1 (2012): 2655-2697.
>
>
>
> **5. "There is no experiment at all and the algorithm seems to be practically feasible. For example, it is unclear how to implement LLM fine-tuning through RLHF using algorithm 2. It is also unclear why we care about cumulative regret in the context of RLHF for LLM alignment."**
>
> We acknowledge that investigating the empirical performance of the proposed algorithm would be interesting. However, our focus is on the theoretical side of RLHF. Note that fine-tuning LLM is not the only application of sequential decision-making from preference feedback (e.g., the dueling bandit literature existed a long time ago), and this is not an LLM paper either. We emphasize that our contribution is on the theoretical side of RL with preference feedback, and as we pointed out already, our work does make a big advancement compared to all prior theoretical work on RL with preference feedback.
>
> There are also many existing works on theoretical RLHF that don't have experiments as well (e.g., [5,6,8]). However, that didn't prevent them from being recognized for their new theoretical contribution. Our work built on top of them, and significantly improved these prior works in terms of computational complexity and query complexity.
>
>
> Regarding the applicability of regret: regret is a common metric for evaluating algorithms and is also considered in many other theoretical RLHF works such as [5,6,7] and in the literature of dueling bandits [9,10,11]. LLM is not the only application for RLHF. For instance, the very early RLHF work from OpenAI indeed considered continuous control as an application [12].
>
> [5] Chen, Xiaoyu, et al. "Human-in-the-loop: Provably efficient preference-based reinforcement learning with general function approximation." ICML, 2022
>
> [6] Saha, Aadirupa, Aldo Pacchiano, and Jonathan Lee. "Dueling RL: Reinforcement Learning with Trajectory Preferences." AISTATS, 2023.
>
> [7] Novoseller, Ellen, et al. "Dueling posterior sampling for preference-based reinforcement learning." UAI, 2020.
>
> [8] Wang, Yuanhao, Qinghua Liu, and Chi Jin. "Is RLHF More Difficult than Standard RL?." NeurIPS, 2023
>
> [9] Yue, Yisong, and Thorsten Joachims. "Beat the mean bandit." Proceedings of the 28th international conference on machine learning ICML, 2011.
>
> [10] Yue, Yisong, et al. "The k-armed dueling bandits problem." Journal of Computer and System Sciences 78.5 (2012): 1538-1556.
>
> [11] Dudík, Miroslav, et al. "Contextual dueling bandits." COLT, 2015
>
> [12] Christiano, Paul F., et al. "Deep reinforcement learning from human preferences." NeurIPS, 2017.

---

> ### Author Response · Authors · 2023-11-21
>
> Dear Reviewer,
>
> Thank you sincerely for your valuable feedback on our work!  As we are approaching the end of the rebuttal session, we hope to respectfully inquire if you have any additional suggestions or concerns. We are readily available to provide further clarification.
>
> Best Regards,
>
> The Authors

---

### Official Review · Reviewer_ynpG · 2023-11-01

**Soundness:** 4 excellent
**Presentation:** 3 good
**Contribution:** 3 good
**Rating:** 8
**Confidence:** 3

**Summary:**

This paper introduces a novel RLHF algorithm capable of learning from preference-based feedback while maintaining efficiency across statistical complexity, computational complexity, and query complexity. The algorithm is demonstrated to strike a near-optimal balance between regret bound and query complexity. Additionally, the authors extend these results to encompass more general nonlinear function approximation through the development of a model-based Thompson sampling method, accompanied by a Bayesian regret analysis.

**Strengths:**

1. This paper makes good theoretical contributions. Despite prior works achieving sublinear worst-case regret for RL with preference-based feedback, these existing algorithms are often computationally infeasible, even in simplified models like tabular MDP. In the context of linear MDP, this paper marks the first RL algorithm that simultaneously achieves sublinear worst-case regret and computational efficiency when handling preference-based feedback.

2. While the primary contribution is theoretical, the research provides valuable practical insights. Notably, the proposed algorithms suggest drawing one trajectory from the latest policy and another from an older policy, rather than two from the same policy, for regret minimization. This innovation departs from conventional practices and enhances the practicality of RLHF.

3. Despite its theoretical nature, the paper is exceptionally well-written and easily comprehensible.

**Weaknesses:**

see Questions

**Questions:**

1. Sekhari et al. (2023a) have previously explored contextual bandits and imitation learning via preference-based active queries, demonstrating efficiency in statistical, computational, and query complexity. While this current paper addresses RL with linear MDP and nonlinear MDP, it would be beneficial to elucidate the specific technical differentiators that set it apart from Sekhari et al. (2023a).

2. The paper proposes a new RLHF algorithm that relies on various input parameters such as $\sigma_r, \sigma_P, \epsilon, \alpha_L, \alpha_U$. The authors provide rates for these parameters in the main theorem, some of which depend on unknown true parameters. To establish practical utility, it is crucial to use numerical simulations to justify the newly proposed algorithm.

3. It is essential to support the theoretical findings with numerical evidence that demonstrates the superiority of the proposed approach over existing benchmark RLHF methods. Numerical comparisons would enhance the practical relevance and acceptance of this paper.

---

> ### Author Response · Authors · 2023-11-17
>
> Thank you for your valuable feedback. Please find our response to your questions below.
>
> 1. **Comparision to Sekhari et al. (2023a).**
>
> We elaborate on the difference below.
>
> (1) Different settings.
>
> Although Sekhari et al. (2023a) also explored learning from preference-based feedback, their setting differs from ours. Their paper has two parts --- contextual bandits and imitation learning. For contextual bandits, they assume that preference feedback is sampled from the rewards difference of arms. This is a special case of ours since Markov decision processes generalize bandits. Since bandit problems have horizon length equal to 1, they do not have to overcome the challenge of exploration over long horizons.
>
> For imitation learning, they assume that the preference feedback is generated from the differences in the state-action value functions of an expert's policy. This is also different from ours since we consider preference feedback to be drawn from the actual cumulative reward. Furthermore, imitation learning setting usually does not require exploration either. Indeed they reduce the imitation learning problem into a series of bandit problems.
>
> To summarize, a typical difference is that they don't need to do deep exploration. However, our setting and feedback modality are different from their, and thus necessitate new technique. For example, since they receive feedback directly from the expert's state-action value function, they can use the performance difference lemma to reduce the performance gap to the learning error. However, this technique may not be applicable in our case, as we don't have any information from an expert.
>
> (2) Different active learning procedures.
>
> The query condition of Sekhari et al. (2023a) replies an explicit version space construction, which could potentially hinder the computational efficiency. In contract, our query conditions are based on variance-style uncertainty quantification of the preference induced by the randomness of the reward model. We query for preference feedback only when the uncertainty is large. This query condition is more practical since approximately computing the uncertainty can be easily done using polynomially many i.i.d. random reward models drawn from the reward model distribution.
>
> In summary, our work explores a different setting from theirs, which requires the development of new techniques. Additionally, our active learning procedure not only differs from their approach but also offers improved computation efficiency. We will ensure to include comprehensive comparison in the revised version of our paper.
>
> 2. **Some of the input parameters depend on unknown true parameters /  support the theoretical findings with numerical evidence**
>
> We think it is not uncommon for theoretical work to assume knowledge of certain parameters, such as the norm upper bound of the true paramter (e.g.,  $B$ in our paper). We also point out we don't need to use the exact values of these parameters in the algorithm, using some coarse upper bounds for these unknown parameters is enough. Also in theory, we think standard techinique such as double trick can be used to overcome this issue. We also agree that the empirical study of our algorithms would be beneficial, and we leave that for future research.

---

> > ### Comment · Reviewer_ynpG · 2023-11-22
> > **Thanks for the feedback**
> >
> > I appreciate the authors' efforts in addressing my feedback. However, as my comment regarding the numerical analysis remains unaddressed, I have chosen not to further increase the rating.

---

### Meta-Review · Area_Chair_gkf1 · 2023-12-06

**Metareview:**

The paper studied RL with preference-based feedback and proposed a new algorithm by randomization. The main contribution is on the theory side, where the authors analyzed the regret bound and query complexity of the algorithm in linear MDP and beyond. Reviewers appreciate the important problem setting and technical contribution. There are some concerns about lack of experiments. However, considering reviewers' recognition of the algorithmic and theoretical contribution, I would recommend accepting this theory paper.

**Justification For Why Not Higher Score:**

There are some concerns about lack of experiments.

**Justification For Why Not Lower Score:**

All reviewers recognized the algorithmic and theoretical contribution.

---

### Decision · Program_Chairs · 2024-01-16

Accept (poster)